**Subject Category:**
Biology (whole organism)

evolution/developmental biology

ontogeny, crista circumfenestralis, macrostomy, heterochrony, paedomorphosis, snake evolution

**Author for correspondence:**
Catherine R. C. Strong
e-mail: crstrong@ualberta.ca

†Present address: Department of Organismic and Evolutionary Biology, Museum of Comparative Zoology, Harvard University, Cambridge, MA 02138, USA.

# Cranial ontogeny of *Thamnophis radix* (Serpentes: Colubroidea) with a re-evaluation of current paradigms of snake skull evolution

Catherine R. C. Strong[1], Tiago R. Simões[1,†], Michael W. Caldwell[1,2] and Michael R. Doschak[3]

[1]Department of Biological Sciences, and [2]Department of Earth and Atmospheric Sciences, University of Alberta, Edmonton, Alberta, Canada T6G 2E9
[3]Faculty of Pharmacy and Pharmaceutical Sciences, University of Alberta, Edmonton, Alberta, Canada T6G 2E1

CRCS, 0000-0002-6080-9245

Accurate knowledge of skeletal ontogeny in extant organisms is crucial in understanding important morpho-functional systems and in enabling inferences of the ontogenetic stage of fossil specimens. However, detailed knowledge of skeletal ontogeny is lacking for most squamates, including snakes. Very few studies have discussed postnatal development in snakes, with none incorporating data from all three major ontogenetic stages—embryonic, juvenile and adult. Here, we provide the first analysis encompassing these three ontogenetic stages for any squamate, using the first complete micro-computed tomography (micro-CT)-based segmentations of any non-adult snake, based on fresh specimens of *Thamnophis radix*. The most significant ontogenetic changes involve the feeding apparatus, with major elongation of the tooth-bearing elements and jaw suspensorium causing a posterior shift in the jaw articulation. This shift enables macrostomy (large-gaped feeding in snakes) and occurs in *T. radix* via a different developmental trajectory than in most other macrostomatans, indicating that the evolution of macrostomy is more complex than previously thought. The braincase of *T. radix* is also evolutionarily unique among derived snakes in lacking a crista circumfenestralis, a phenomenon considered herein to represent paedomorphic retention of the embryonic condition. We thus present numerous important challenges to current paradigms regarding snake cranial evolution.

# 1. Introduction

Though many studies have examined embryonic cranial development in snakes (e.g. [1–3]), comparatively few have analysed the postnatal development of the snake skull (e.g. [4–6]). Of these few studies on snake postnatal cranial ontogeny, many have focused on changes in gape size and diet, thus analysing only the jaw bones (e.g. [7,8]). Furthermore, to our knowledge, no study thus far has provided a detailed description of changes in cranial osteology encompassing both pre- and postnatal ontogeny in any squamate species. However, studies of postnatal ontogeny can be crucial in revealing new information about organismal biology and evolution [8]. For example, macrostomy—or large-gaped feeding—is a fundamental and anatomically complex feature of the body plan of most extant snakes and has been hypothesized as one of the major factors enabling the diversification of this group [8,9]. Initially considered to have evolved only once [10], recent analyses of the postnatal cranial development of various major snake groups have revealed the macrostomatan condition to be achieved via different developmental pathways in different groups of snakes, suggesting that this feature is homoplastic [6]. This conclusion is supported by recent phylogenies showing macrostomatan snakes to be non-monophyletic (e.g. [11]).

A thorough understanding of ontogenetic patterns is essential not only in revealing new information about the biology of extant snakes but also in informing our interpretations of fossil snake material [5]. Discerning between individual and interspecific variation can be difficult for many fossil species, especially when fossil specimens are composed of disarticulated bones, as is often the case with fossil snakes. Changes to osteology throughout growth are a major source of this individual variation; as such, understanding major patterns of snake skeletal ontogeny can help inform our interpretations of fossil snakes, especially regarding species delimitations. Similarly, knowledge of which features are more prone to change during ontogeny versus which features remain stable can help determine characters' suitability for phylogenetic analyses [6]. Ideally, the characters included in morphological character matrices should remain consistent throughout growth, as variation resulting from ontogeny can potentially dilute the phylogenetic signal of the character data.

With the advent of micro-computed tomography (micro-CT), studies of postnatal ontogeny can also serve as highly detailed re-examinations of the general anatomy of the organism in question. This ability to re-analyse specimens in fine detail can, in turn, provide new information and interpretations pertaining to organismal evolution, such as recognition of cases of heterochrony via paedomorphosis or peramorphosis. Such re-analyses of extant organismal development have increasingly been employed as a method of examining evolutionary trends in these organisms (e.g. [9]), as they enable detailed observations not afforded by the fossil record alone.

In the light of these advantages of studying postnatal ontogeny, the current study addresses the aforementioned limitations in the literature via an analysis of the cranial osteology of *Thamnophis radix*, the Plains Garter Snake. A member of the Colubroidea, the genus *Thamnophis* is recognized as one of the most diverse and ubiquitous genera of colubroid snakes in North America [12]. Despite this diversity, however, only a few studies have been performed regarding general ontogeny or cranial development in this genus (e.g. [13,14]), with none close to the level of detail provided by micro-CT imaging.

We thus provide the first detailed anatomical description of the changes in the cranial ontogeny of this snake taxon, performed using fully segmented micro-CT scans of embryonic, juvenile and adult *T. radix* skulls (figures 1–5). This represents both the first instance of a complete skull segmentation for any non-adult snake and the first comprehensive summary of cranial ontogeny fully spanning embryo to juvenile to adult stages in any squamate species. This analysis also reveals unique cranial features of *T. radix* that challenge current views regarding the evolution of both macrostomy—a feeding strategy involving large gape, unique to certain snakes—and of the crista circumfenestralis (CCF), a component of the snake braincase historically associated with an advanced phylogenetic position.

# 2. Material and methods

Eight alcohol-preserved *T. radix* specimens—three embryos (UAMZ R620), two juveniles (UAMZ R950) and three adults (UAMZ R636)—were obtained from the University of Alberta Museum of Zoology (UAMZ) collections. Specific information regarding the parentage of the embryonic and juvenile snakes was unavailable. However, given that all of the embryos were accessioned under the same specimen number, it is likely that they are from the same mother. The same line of reasoning also

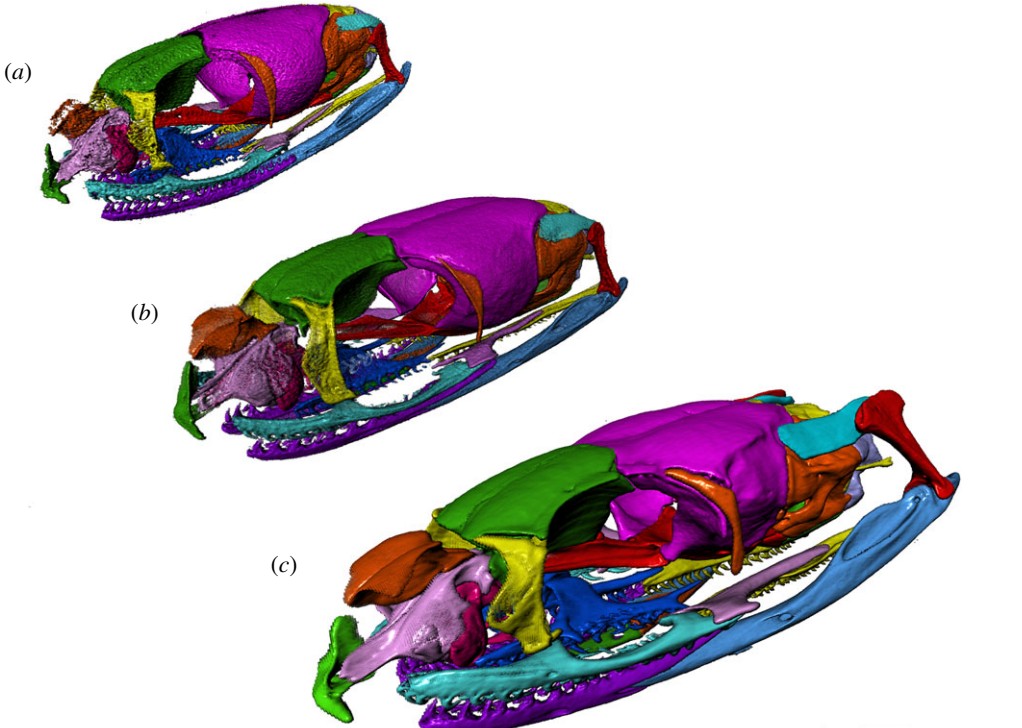

**Figure 1.** Left anterolateral view of *T. radix* skull throughout ontogeny: (*a*) embryo; (*b*) juvenile; (*c*) adult. Note: full length of scale bar equals 2 mm; black and grey portions of scale bar each equal 1 mm. Surface mesh (STL) files for each individually segmented bone in each stage are available in the electronic supplementary material given in [15].

applies to the two juvenile snakes. The embryonic snakes were stillborn in captivity, and the degree of ossification and overall size of the individuals (skull length = 9.7 mm) indicate that they were close to birth. The juveniles are slightly larger than the embryos (skull length = 11.6 mm) and probably represent an early postnatal stage of development. The adults are much larger than the juveniles (skull length = 17.1 mm), being similar in size to adults of closely related species (e.g. *Thamnophis elegans*) (T.R.S., personal observation, 2018). Our sample size was limited to eight individuals, as the only other embryonic or juvenile *Thamnophis* specimens in the UAMZ collections were from other species and therefore would have introduced interspecific variation as a confounding factor in this ontogenetic analysis. This sample size is consistent with that used in other micro-CT-related studies of snake morphology (e.g. see [6,16]).

All specimens were scanned using a Bruker-SkyScan 1076 micro-CT scanner (Bruker-SkyScan, Kontich, Belgium) at the Pharmaceutical Orthopaedic Research Lab (University of Alberta). Samples were scanned at 9 µm resolution, and the reconstructed image contrast was optimized for each skull sample by adjusting the cathode ray tube voltage to permit approximately 22–25% X-ray transmission to the detector for the densest parts of each sample projection. That corresponded to a tube voltage/current of 74 kV/129 µA for the adult skull and 48 kV/200 µA for both the juvenile and embryo skull samples, with low energy X-rays removed in all samples using a 0.5-mm aluminium filter. Three scan projections were averaged per step, through the 180° of rotation at 0.5° step increments with exposure times of 1178 ms for the adult skull and 589 ms for both the juvenile and embryo skull samples. The two-dimensional raw image projections were reconstructed using a modified Feldkamp back-projection algorithm, with the cross-section to image conversion values set to 0.0–0.101 for all samples, using bundled vendor software (NRecon, v. 1.4.4, Skyscan NV, Belgium). The resulting image files were visualized in Dragonfly 1.0 (Object Research Systems), with the Threshold tool being used to digitally remove the surrounding soft tissue from the skull. The highest-quality scan for each ontogenetic stage was chosen for complete segmentation of the skull bones.

The segmented skulls were then qualitatively compared to determine ontogenetic trends among the three stages, with further comparisons to the non-segmented skulls to ensure consistency and reliability in our observations of developmental differences. We also examined a non-accessioned skeletonized specimen of *T. sirtalis parietalis* to compare external morphology of the otico-occipital region.

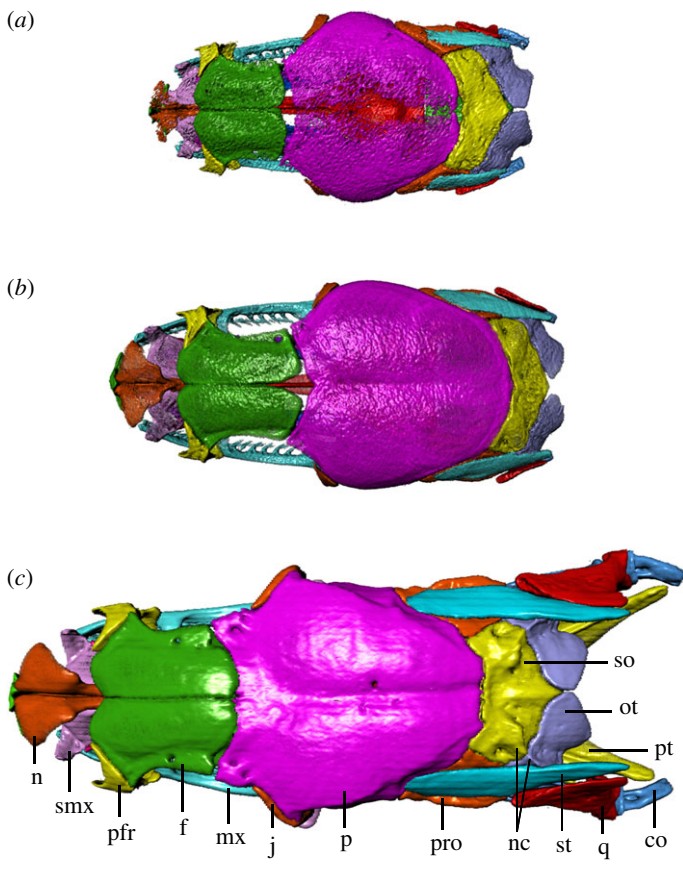

**Figure 2.** Dorsal view of *T. radix* skull throughout ontogeny: (*a*) embryo; (*b*) juvenile; (*c*) adult. Anterior is to the left. Bones are labelled on the adult skull; coloration of each bone is consistent throughout all figures. co, compound bone; f, frontal; j, jugal; mx, maxilla; n, nasal; nc, nuchal crest; ot, otoccipital; p, parietal; pfr, prefrontal; pro, prootic; pt, pterygoid; q, quadrate; smx, septomaxilla; so, supraoccipital; st, supratemporal. Scale bar as explained in figure 1. Surface mesh (STL) files for each individually segmented bone in each stage are available in the electronic supplementary material given in [15].

## 3. Results

We herein present a full description of the cranial ontogeny of each bone in the *T. radix* skull, with the bones grouped according to the general skull region. A summary of the most significant ontogenetic changes to the *T. radix* skull is presented in table 1.

### 3.1. Snout

The snout region is the most poorly ossified region of the embryonic skull, with the nasal being the least ossified snout element. Many of the changes undergone in this region are thus associated with improved ossification. The overall snout and its constituent elements also display a general change in shape throughout ontogeny, going from globular in the embryo and juvenile stages to more depressed in the adult (figure 3*a*–*c*).

Throughout ontogeny, changes to the vomer, premaxilla and septomaxilla result in increased integration of the premaxilla with the rest of the snout (figures 3*a*–*c*, 4*a*–*c* and 5*a*–*c*). The premaxillary processes of the embryonic vomer and the anterior processes of the septomaxilla are short, while the nasal process of the premaxilla is dorsoventrally and anteroposteriorly short. As such, the space between the nasal process and vomerine processes of the premaxilla, which will eventually accommodate the septomaxilla and vomer, is shallow and articulates only with the septomaxilla (figures 3*a*, 4*a* and 5*a*). In the juvenile, the vomer's premaxillary processes become elongated relative to the rest of the vomer, now extending to the vomerine processes of the premaxilla (figures 3*b*, 4*b* and 5*b*). From the juvenile to adult stages, the premaxilla's nasal process expands both dorsoventrally and anteroposteriorly, thus making the space between the nasal and vomerine processes narrower dorsoventrally and deeper anteroposteriorly. Extension of the anterior processes of the septomaxilla and premaxillary processes of the vomer relative to the rest of these bones

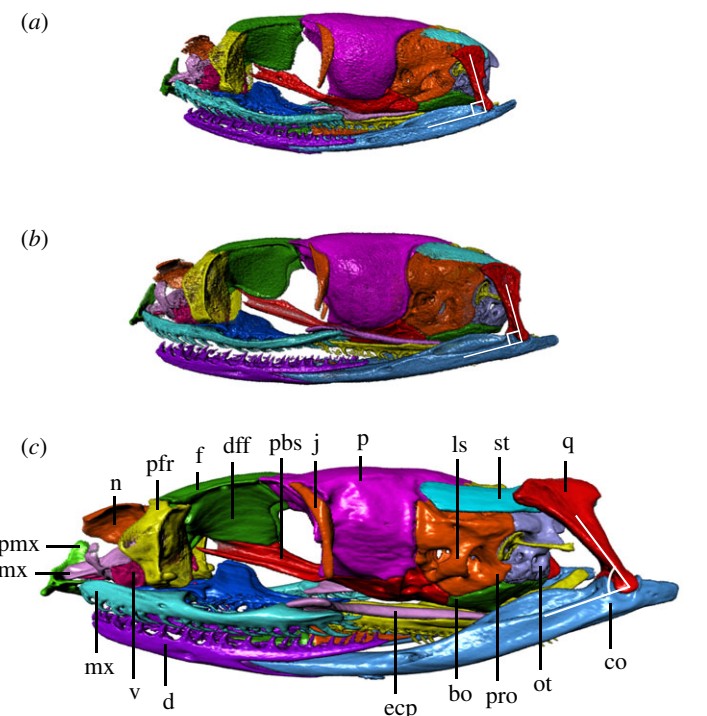

**Figure 3.** Left lateral view of *T. radix* skull throughout ontogeny: (*a*) embryo; (*b*) juvenile; (*c*) adult. Anterior is to the left. Note the posterior rotation of the quadrate from the juvenile to adult stages. bo, basioccipital; co, compound bone; d, dentary; ecp, ectopterygoid; dff, descending flange of the frontal; f, frontal; j, jugal; ls, laterosphenoid; mx, maxilla; n, nasal; ot, otoccipital; p, parietal; pbs, parabasisphenoid; pfr, prefrontal; pmx, premaxilla; pro, prootic; q, quadrate; smx, septomaxilla; st, supratemporal; v, vomer. Scale bar as explained in figure 1. Surface mesh (STL) files for each individually segmented bone in each stage are available in the electronic supplementary material given in [15].

creates stronger contact—and thus improved integration—between the premaxilla and the rest of the snout (figures 3*c*, 4*c* and 5*c*).

The embryonic nasal consists mainly of the vertical laminae, as the dorsal laminae are present only as a very weakly ossified surface (figure 2*a*). The juvenile nasal (figure 2*b*) shows improved ossification, with the dorsal laminae, in particular, showing increased mineralization, thus approaching the adult form. The adult nasal is characterized by complete ossification (figure 2*c*).

## 3.2. Skull roof

Changes to the skull roof throughout ontogeny are generally the result of improved ossification throughout development (figure 2*a–c*).

Contact between the frontal and parietal is completely absent in the embryo. The juvenile skull roof has minor contact between these bones, though a central fontanelle persists. This fontanelle is closed by the adult stage. A second fontanelle is present in the embryo between the parietal and supraoccipital; this fontanelle closes by the juvenile stage.

The paired parietals and frontals each remain unfused in the embryo. The parietals are unfused along almost the entire longitudinal axis, with a large unossified region at the centre of the parietal roof, while the frontals are weakly sutured anteriorly. By the juvenile stage, the parietal roof is fused and bears a slight sagittal sulcus that deepens in the adult. The adult parietal also has a prominent crest running dorsolaterally from the posterior margin of the orbit to the posteromedial margin of the parietal. The posterior margin itself is smoothly lobate in the embryo and juvenile but becomes more angled in the adult. The frontals are not completely sutured to each other until the adult stage.

## 3.3. Orbit

The orbit is large relative to the rest of the skull in the embryonic stage, gradually decreasing in relative size and becoming dorsoventrally flattened through the juvenile and adult stages (figure 3*a–c*). The optic foramen also decreases in relative size throughout ontogeny, due to a posterior elongation of the descending flange of the frontal, medial expansion of the anterior margin of the parietal and an overall flattening of the skull.

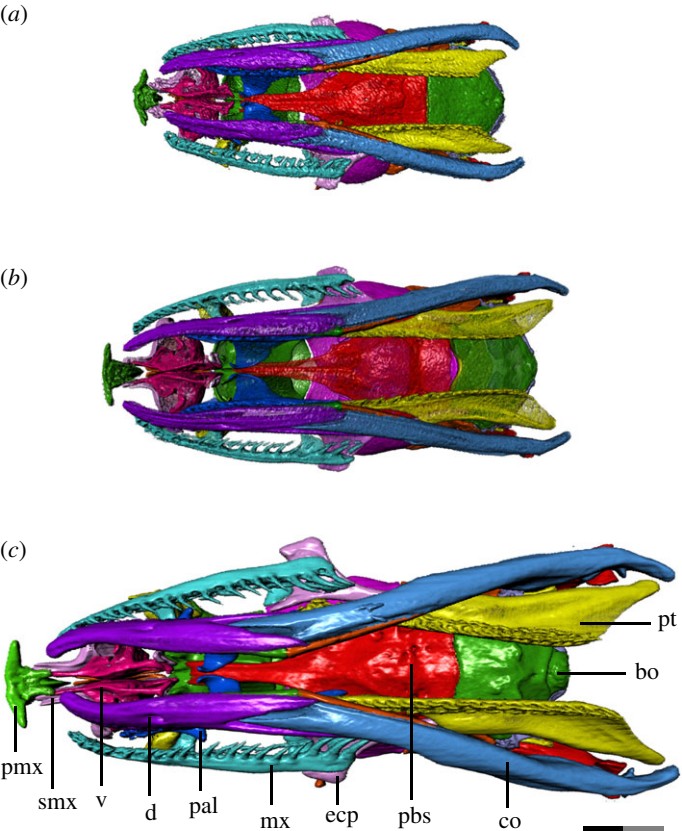

**Figure 4.** Ventral view of *T. radix* skull throughout ontogeny: (*a*) embryo; (*b*) juvenile; (*c*) adult. Anterior is to the left. bo, basioccipital; co, compound bone; d, dentary; ecp, ectopterygoid; mx, maxilla; pal, palatine; pbs, parabasisphenoid; pmx, premaxilla; pt, pterygoid; smx, septomaxilla; v, vomer. Scale bar as explained in figure 1. Surface mesh (STL) files for each individually segmented bone in each stage are available in the electronic supplementary material given in [15].

The major discrete changes to the orbit involve the descending flange of the frontal and the parasphenoid process of the parabasisphenoid. The embryo displays very limited contact between these elements, with the paired ventral ridges of the frontal that articulate with the parasphenoid process in the adult being only weakly developed (figure 3*a*). These ridges become increasingly prominent through the juvenile and adult stages, in conjunction with an increase in contact between the descending flange of the frontal and the parasphenoid process (figure 3*b,c*). This increased contact is also accompanied by the development of a tubercle on the dorsal margin of the parasphenoid process where it contacts the posteroventral corner of the descending flange of the frontal (figure 3*a–c*).

The descending flange of the frontal also bears an emargination on its posterior margin that deepens throughout development. The parasphenoid process develops increasingly prominent crests along its dorsal and ventral margins throughout ontogeny.

The frontal also undergoes changes to its articulating surfaces with other bones. The paired flanges—ventral to the ethmoid foramina—that articulate with the septomaxilla are more prominent in the adult than in the embryo and juvenile stages (figure 1). The anterodorsal wings and notch that articulate with the prefrontals are similarly only weakly present in the embryo; minor elaboration of this anterodorsal notch in the juvenile precedes a strong deepening in the adult (figures 2*a–c* and 3*a–c*). A corresponding thickening of the dorsal lappet of the prefrontal results in a strong interlocking contact between the frontal and prefrontals.

The parietal undergoes a similar elaboration of the anterolateral wings that accommodate the jugal (figures 2*a–c* and 3*a–c*).

## 3.4. Palate

The ectopterygoid is progressively lengthened throughout ontogeny, with its posterior terminus moving from below the parabasisphenoid–basioccipital suture in the embryo, to slightly beyond this suture in the juvenile, to well posterior to this suture in the adult.

The pterygoid experiences a similar elongation of its posterior flange (quadrate process *sensu* [17]). In the embryo, the pterygoid's posterior terminus is at the level of the occipital condyle (figures 4*a*

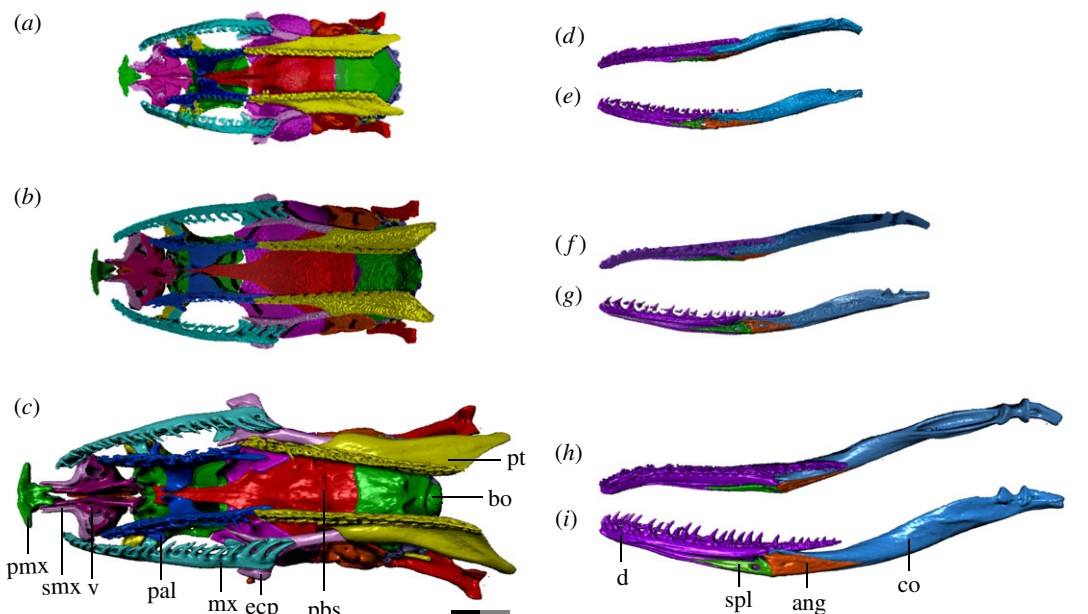

**Figure 5.** Ventral view of *T. radix* skull throughout ontogeny with the mandibles removed (*a*–*c*) and displayed separately (*d*–*i*). (*a*) Embryo; (*b*) juvenile; (*c*) adult; (*d*) embryonic right mandible in dorsal and medial (*e*) views; (*f*) juvenile right mandible in dorsal and medial (*g*) views; (*h*) adult right mandible in dorsal and medial (*i*) views. Anterior is to the left. ang, angular; bo, basioccipital; co, compound bone; d, dentary; ecp, ectopterygoid; mx, maxilla; pal, palatine; pbs, parabasisphenoid; pmx, premaxilla; pt, pterygoid; smx, septomaxilla; spl, splenial; v, vomer. Note: scale bar as explained in figure 1. Surface mesh (STL) files for each individually segmented bone in each stage are available in the electronic supplementary material given in [15].

and 5*a*), while in the juvenile, it is slightly posterior to the condyle (figures 4*b* and 5*b*). The pterygoid tooth row terminates anteriorly to the condyle in both of these stages. The pterygoid experiences a major elongation in the adult, with the tooth row terminating well posterior to the occipital condyle, and the quadrate process terminating even further posterior to that (figures 4*c* and 5*c*). The quadrate process is also dorsally recurved, with its posterior terminus rising to the level of the occipital condyle in the adult, rather than being inferior to it as in previous growth stages (figure 3*a*–*c*).

The pterygoid bears 24 tooth positions in the embryo, though the limited state of development makes it difficult to distinguish between functional and replacement teeth. The pterygoid bears 23 tooth positions (with an average of 10 ankylosed teeth) in the juvenile and 26 tooth positions (with 18 ankylosed teeth) in the adult. The teeth start as quite small and squat in the embryo, becoming slightly longer in the juvenile, and finally much longer and more recurved in the adult (figures 3*a*–*c* and 4*a*–*c*).

By contrast, the palatine does not undergo a similar dramatic change in shape or increase in length. The palatine tooth row bears 15 tooth positions in the embryo, though again the poor state of development makes it difficult to distinguish between replacement teeth, functional teeth in the process of being replaced and ankylosed functional teeth. The palatine bears 16 tooth positions (with 12 ankylosed teeth) in the juvenile and 16 tooth positions (average of 12 ankylosed teeth) in the adult. These teeth undergo an increase in length and curvature similar to that noted in the pterygoid teeth.

## 3.5. Braincase

The braincase undergoes a general dorsoventral flattening throughout ontogeny, from an initially globular appearance in the embryo to a dorsoventrally compressed and anteroposteriorly elongated form in the adult (figure 3*a*–*c*). This dorsoventral compression is most pronounced in the anterior half of the braincase, as the posterior half maintains a relatively constant depth throughout ontogeny.

The foramina of the otic capsule undergo several changes throughout ontogeny. In the embryonic otoccipital (figure 6*a*), the jugular foramen—posteroventral to the fenestra ovalis and posterior to the lateral aperture of the recessus scalae tympani (LARST; fenestra rotunda of some authors, e.g. [18])—is small, poorly defined and undivided internally. The crista tuberalis—separating the jugular foramen from the LARST—is present as a narrow shaft with a small tubercle at its base and is not fused ventrally to the crista interfenestralis, causing the LARST to be open laterally. Neither of these cristae contacts the basioccipital ventrally. The LARST itself is larger than the jugular foramen.

**Table 1.** Summary of the most significant osteological trends throughout the cranial ontogeny of *T. radix* (see text for details). LARST, lateral aperture of the recessus scalae tympani.

| bone | embryo | juvenile | adult |
|---|---|---|---|
| premaxilla | — minimal contact with rest of snout | — as in embryo | — stronger integration with rest of snout |
| nasal | — dorsal laminae present only as a very weakly ossified surface | — increased ossification of dorsal laminae | — completely ossified |
| frontal | — no contact with parietals<br>— posterior third unsutured | — minor contact with parietal, though fontanelle persists | — complete contact with parietal<br>— frontals completely sutured |
| parietal | — central parietal roof unossified<br>— parietal—supraoccipital fontanelle present | — parietal roof ossified<br>— closure of parietal—supraoccipital fontanelle | — prominent muscle attachment crests |
| pterygoid | — posterior terminus below occipital condyle | — posterior terminus slightly beyond occipital condyle | — major posterior elongation of tooth row and quadrate process |
| otoccipital | — jugular foramen small, internally undivided<br>— crista tuberalis unfused ventrally to crista interfenestralis, LARST opens laterally | — jugular foramen more defined, still undivided<br>— crista tuberalis expanded, but still unfused to crista interfenestralis | — jugular foramen large, subdivided, with well-defined jugular recess<br>— crista tuberalis fused to crista interfenestralis |
| quadrate | — anterior to occipital condyle<br>— only slightly off-vertical | — dorsoventrally elongated<br>— slight posterior shift | — further elongation<br>— posterior to occiput<br>— ventral terminus rotated strongly posteriorly |
| supratemporal | — posterior terminus downcurved and anterior to occiput | — no downward curvature<br>— slight posterior elongation | — posterior terminus well posterior to occiput |
| compound bone | — terminates slightly posteroventral to occipital condyle<br>— mandibular condyle anterior to occipital condyle | — slight posterior elongation<br>— mandibular condyle closer to level of occipital condyle | — strong posterior elongation and posterodorsal deflection<br>— mandibular condyle well posterior to occiput |
| dentary | — teeth short, stout and minimally recurved | — teeth longer and narrower than in embryo | — teeth long, narrow and strongly recurved |

In the juvenile otoccipital (figure 6b), the jugular foramen is now larger than the LARST, with a more defined rim than in the embryo, though remains undivided. The crista tuberalis is expanded internally into a wall dividing the jugular foramen from the LARST (rather than just a shaft as in the embryo), though remains unfused to the crista interfenestralis, with these cristae still lacking ventral contact with the basioccipital. The tubercle at the base of the crista tuberalis remains small, as in the embryo.

In the adult otoccipital (figure 6c), the jugular foramen is further enlarged, with a distinct ridge around its rim defining the jugular recess. The jugular foramen also develops internal struts subdividing it into three

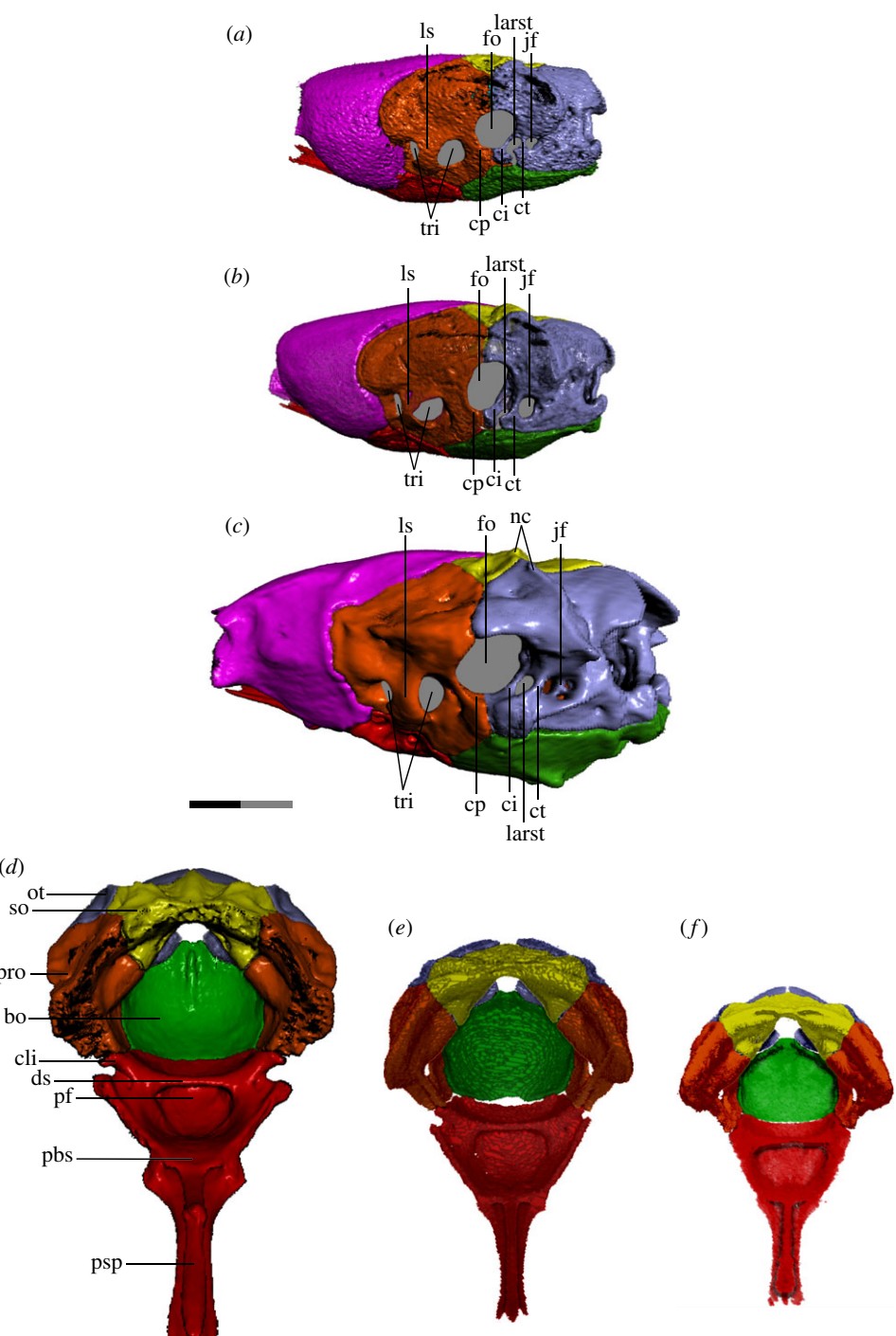

**Figure 6.** Braincase of *T. radix* in the left posterolateral view (*a–c*) and anterodorsal view (*d–f*). Typically, colubroids (derived snakes) possess a CCF in which the cristae prootica, interfenestralis and tuberalis are elaborated into a shelf that partially covers the stapedial footplate/fenestra ovalis; however, in *T. radix*, this feature is conspicuously absent. (*a,f*) Embryo; (*b,e*) juvenile; (*c,d*) adult. In both *a* and *b*, note the lack of fusion between the cristae interfenestralis and tuberalis, leaving the LARST open laterally. bo, basioccipital; cli, clinoid process; ci, crista interfenestralis; cp, crista prootica; ct, crista tuberalis; ds, dorsum sellae; fo, fenestra ovalis; jf, jugular foramen; larst, lateral aperture of the recessus scalae tympani, ls, laterosphenoid; nc, nuchal crest; ot, otoccipital; pbs, parabasisphenoid; pf, pituitary fossa; pro, prootic; psp, parasphenoid process of the parabasisphenoid; so, supraoccipital; tri, trigeminal openings. Note: scale bar as explained in figure 1. Surface mesh (STL) files for each individually segmented bone in each stage are available in the electronic supplementary material given in [15].

openings internally, associated with the glossopharyngeal, vagus, accessory and hypoglossal nerves [19,20]. The crista tuberalis is now fused ventrally to the crista interfenestralis, with the tubercle at its base becoming significantly enlarged and more bulbous compared to previous stages. The LARST is thus fully enclosed.

The crista prootica has minimal prominence in all three stages. The CCF—a structure produced by elaboration of the cristae prootica, tuberalis and interfenestralis to cover the stapedial footplate—is absent throughout ontogeny, leaving the stapedial footplate and LARST exposed in lateral view (figures 3a–c and 6a–c) (see [21] for illustrations of the CCF in a range of snake taxa).

Other changes to the otoccipital include an increase in dorsal proximity between the paired otoccipitals (figure 2a–c), as well as a progressive enlargement of the posteroventral tubercles that contribute to the occipital condyle. The nuchal crest that runs anteromedial–posterolateral above the fenestra ovalis is minimally developed in the embryo, resulting in a globular process above each fenestra ovalis; as the nuchal crest progressively increases in prominence, it eventually becomes sharp and triangular in the adult otoccipital (figures 2c and 6a–c).

Major changes to the prootic include increased prominence of muscle crests throughout development. For example, the dorsal surface of the prootic is smooth and rounded in the embryo, but progressively develops a flattened ledge and distinct lateral ridge in association with its increased contact with the supratemporal (figures 2a–c and 3a–c). Similarly, the prootic also experiences a progressive thickening and increase in definition of the crests and subdivisions associated with the semicircular canals.

The laterosphenoid ossification—subdividing the trigeminal foramen into anterior and posterior openings for the maxillary and mandibular branches of the trigeminal nerve, respectively—is fully ossified in all three ontogenetic stages (figures 3a–c and 6a–c). The anterior border of the trigeminal foramen is narrow in the embryo and juvenile, but increases in thickness in the adult.

The supraoccipital changes from roughly boomerang-shaped in dorsal view in the embryo and juvenile to more transversely compressed in the adult (figure 2a–c). Rather than being smoothly curved as in previous stages, the anterior margin of the adult supraoccipital is straight, with the anterior corners being angled anterolaterally, while the posterior margin tapers to a narrow point. This change in shape is accompanied by a decrease in how far the supraoccipital extends transversely across the skull roof, as well as a narrowing of the large ventral notch of the supraoccipital, between the otic capsules. The supraoccipital also progressively develops a more pronounced nuchal crest that runs anteromedial–posterolateral and continues onto the otoccipital. The supraoccipital's internal contact with the otoccipital and prootic is weak in the embryo and juvenile braincases compared to the adult condition. Within the adult otic capsule, internal contact between the supraoccipital and otoccipital is almost complete, while internal contact with the prootic is stronger than in the juvenile but still incomplete.

The stapes undergoes a progressive increase in the length of its shaft (figure 3a–c).

Both the basioccipital and parabasisphenoid become progressively more dorsally concave throughout development, with their lateral apices going from thin and only slightly elevated in the embryo and juvenile, to thickened and rising higher along the lateral surface of the braincase in the adult (figure 3a–c).

In the parabasisphenoid, this increase in concavity is accompanied by an elaboration of the dorsum sellae (figure 6). Present only as a simple shelf overhanging a broad and shallow central basin and pituitary fossa in the embryo (figure 6f), the dorsum sellae develops anterolaterally projecting clinoid processes and an accompanying notch in the juvenile (figure 6e). This notch contributes significantly to the foramen through which nerve innervating the constrictor internus dorsalis complex (cid-nerve) [22] exits the braincase, a foramen which in the embryo is formed only by a slight posteroventral notch in the parietal. With this thickening and elaboration of the dorsum sellae, the pituitary fossa becomes deeper and narrower. These changes continue in the adult, with the clinoid processes extending further laterally—causing the notch to deepen and thus the exit for the cid-nerve to be enlarged—while the pituitary fossa becomes further deepened and narrowed (figure 6d).

The junction between the parasphenoid process and the body of the parabasisphenoid also becomes elaborated, with minor lateral projections in the embryo and juvenile becoming enlarged into prominent triangular projections in the adult (figure 6d–f). These projections contribute to an increase in sutural contact with the anteroventral parietal.

The basioccipital's occipital condyle becomes more pronounced throughout ontogeny, becoming progressively larger and more bulbous and developing a deep ventral groove separating it from the main body of the basioccipital (figure 4a–c). The anterior margin of the embryonic basioccipital is straight and only barely contacts the parabasisphenoid (figure 6f); this contact is increased in the juvenile, though the basioccipital develops a central notch on its anterior margin that creates a gap between this margin and the parabasisphenoid (figure 6e). In the adult braincase, the basioccipital and parabasisphenoid are in complete contact (figure 6d).

The lateral and ventral surfaces of the basioccipital and parabasisphenoid—initially smooth in the embryo—undergo a progressive elaboration of their muscle crests. For example, slight ventral transverse ridges that appear on the juvenile basioccipital are elaborated into prominent triangular

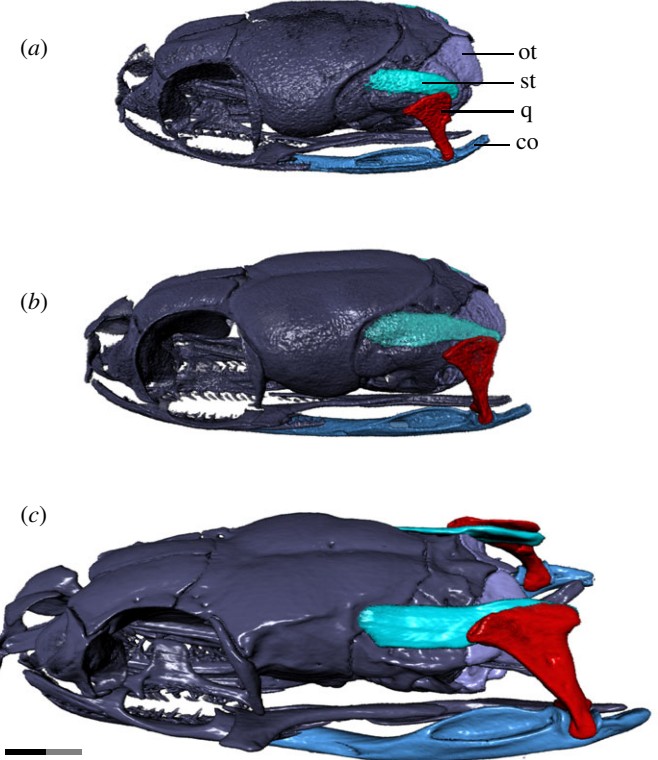

**Figure 7.** Left laterodorsal view of the jaw suspensorium and related skull elements of *T. radix* throughout ontogeny. (*a*) Embryo; (*b*) juvenile; (*c*) adult. Note the strong posterior elongation of the supratemporal and compound throughout development, as well as the elongation and posterior rotation of the quadrate; this pathway of development has so far been observed only in *T. radix* and one other, unrelated genus within the Caenophidia, indicating independent evolution of this pattern. co, compound bone; ot, otoccipital; q, quadrate; st, supratemporal. Anterior is to the left. Note: scale bar as explained in figure 1. Surface mesh (STL) files for each individually segmented bone in each stage are available in the electronic supplementary material given in [15].

protrusions on either side of the ventral longitudinal midline in the adult (figure 4*a*–*c*). A slight dorsal longitudinal crest on the juvenile occipital condyle is extended into a prominent crest running from the centre to the posterior margin of the dorsal basioccipital (figure 6*d*–*f*).

## 3.6. Suspensorium

The embryonic quadrate is squat and only slightly off-vertical (figures 3*a* and 7*a*). Throughout ontogeny, the quadrate becomes dorsoventrally elongated and undergoes significant posterior rotation of the ventral terminus between the juvenile and adult stages (figures 3*a*–*c* and 7*a*–*c*). The cephalic condyle of the quadrate becomes longitudinally expanded and shifts from articulating equally with the prootic anteriorly and the supratemporal posteriorly in the embryo, to articulating with mainly the supratemporal in the juvenile, to finally articulating only with the supratemporal in the adult (figures 3*a*–*c* and 7*a*–*c*). This shift in articulation is associated with an overall posterior shift in the location of the quadrate relative to the skull; the quadrate progressively moves from lateral to the basioccipital and anterior to the level of the occipital condyle in the embryo to fully posterior to the occipital condyle in the adult, with the majority of this shift occurring between the juvenile and adult stages. The quadrate also undergoes a progressive deepening of its mandibular condyle, as well as a progressive development of a sharp adductor crest on its anterolateral margin.

The overall shape of the supratemporal changes throughout development (figures 3*a*–*c* and 7*a*–*c*): this bone is downcurved posteriorly in the embryo; in the juvenile, it is straight though still angled slightly posteroventrally, while the adult supratemporal is horizontal in orientation. The supratemporal experiences significant posterior elongation relative to the skull throughout development, with the posterior terminus located just anterior to and at the level of the occipital condyle in the embryo and juvenile, respectively, while being well posterior to the occipital condyle in the adult. The anterior terminus is also progressively elongated, progressively shifting its articulation from just the prootic in the embryo to the prootic and parietal in the adult.

## 3.7. Jaws

The lower jaws become progressively lengthened relative to the skull throughout ontogeny, due largely to a posterodorsal elongation of the compound bone. The embryonic compound bone terminates just posteroventral to the occipital condyle, with the mandibular condyle anteroventral to the occipital condyle (figures 3a and 7a). The juvenile compound bone is slightly elongated, terminating slightly farther beyond the occipital condyle (figures 3b and 7b). In the adult, both the mandibular condyle and the posterior terminus of the compound bone are well posterior to the occipital condyle (figures 3c and 7c). This lengthening is accompanied by a strong dorsal deflection of the posterior half of the compound bone; while the compound bone is slightly upturned posteriorly in the embryo and juvenile—though is still entirely ventral to the level of the occipital condyle—a pronounced dorsal deflection in the adult stage results in the posterior terminus—from the mandibular condyle posteriorly—being above this level.

Throughout ontogeny, the mandibular condyle itself elaborates from a simple depression on the posterodorsal surface of the embryonic compound bone to a saddle-shaped condyle bordered by narrow anterior and posterior ascending processes in the juvenile (figure 3a,b). These ascending processes are expanded transversely in the adult (figure 3c), such that they are greater in width than the shaft of the compound bone itself. The compound bone also undergoes a slight deepening and increase in the dorsal enclosure of the mandibular fossa, as well as an increase in ossification of the anterior terminus and an associated increase in definition of the Meckelian groove medially exposed at this terminus (figure 5d–i).

The Meckelian groove of the dentary becomes deeper throughout ontogeny. While this groove extends to the dentary's anterior terminus in the embryo and juvenile stages, it has increased medial enclosure in the adult, thus terminating at around the level of the fifth dentary tooth. The posterior bifurcation of the dentary deepens with growth, reaching progressively farther back along the compound bone and becoming thickened in the adult stage (figure 3a–c). The anterior termini of the dentary become progressively medially inflected throughout ontogeny (figure 4a–c).

The dentary bears 25 tooth positions (average of 18 ankylosed teeth) in the embryo, 25 tooth positions (average of 15 ankylosed teeth) in the juvenile and 28 tooth positions (average of 19 ankylosed teeth) in the adult. The embryonic teeth are initially short and squat, though become increasingly longer, narrower and more recurved throughout ontogeny (figure 3a–c).

The anterior termini of the maxilla undergo a similar medial inflection throughout ontogeny, though not as pronounced as in the dentary (figure 4a–c). The maxilla is initially dorsoventrally narrow in the embryo, with a thin and hook-like anterior palatine process. This process is broadened in the juvenile, becoming more shelf-like, while the overall maxilla becomes taller and thicker in the adult, especially near the anterior and posterior termini (figure 3a–c).

The maxilla bears 23 tooth positions (average of 18 ankylosed teeth) in the embryo, 23 tooth positions (average of 17 ankylosed teeth) in the juvenile and 24 tooth positions (average of 18 ankylosed teeth) in the adult. Similar to the dentary teeth, the maxillary teeth undergo an increase in length throughout ontogeny, though the maxilla is unique in having the posterior teeth become elongated and thickened relative to the anterior teeth (figure 3a–c). The curvature of the maxillary teeth is similar throughout growth.

The angular and splenial (figure 5d–i) undergo minor changes throughout development, mainly associated with increased ossification. Both bones experience a progressive deepening of the lateral longitudinal depressions present to accommodate Meckel's canal and are slightly dorsally expanded and anteroposteriorly elongated throughout development.

## 3.8. Other

Both the embryonic and juvenile skulls contain small, roughly ovoidal concretions within the otic capsule, surrounded by the prootic, otooccipital and supraoccipital bones. These concretions are likely statolithic masses, calcareous crystals that accrete within the saccule of the inner ear [2,23].

# 4. Discussion

## 4.1. Ontogenetic development of macrostomy

The most dramatic changes to the *T. radix* skull throughout ontogeny are associated with modifications to the tooth-bearing elements and suspensorium (figure 7a–c). Lengthening of the compound bone and

supratemporal relative to the skull, backward rotation of the quadrate, and a posterior shift in the position of the quadrate relative to the skull result in an overall posterior shift of the jaw joint that ultimately increases the maximum gape of the jaws [6,8,10]. These changes are essential in enabling the feeding mechanics of the adult snake. *T. radix* feeds using macrostomy, a condition in which prey with a high cross-sectional area are ingested whole, with essentially no food processing or size reduction occurring in the mouth [8]. As individuals get larger throughout development, they require more energy and thus larger prey items; as such, an ontogenetic shift in diet occurs between the juvenile and adult stages, and the gape of the jaws must increase in order to accommodate increasingly larger prey [4,8]. Specifically, *T. radix* juveniles eat small prey items such as annelids and small anurans, whereas adults consume a range of prey, including larger organisms such as small mammals [24]. The aforementioned increase in gape and lengthening of the jaw apparatus occurs in conjunction with a general increase in length and recurvature of the teeth throughout ontogeny, thus improving the snake's ability to grasp onto progressively larger and stronger prey. The quadrate ramus of the pterygoid also increases in length posteriorly such that the pterygoid tooth row extends past the occipital condyle in adults (figures 2*a*–*c* and 4*a*–*c*). Similar posterior shifts in the jaw joint have been noted in other macrostomatans (e.g. [6,8,25]), indicating that this ontogenetic change is highly conserved among snakes that employ this feeding strategy.

The specific pattern of jaw joint development present in *T. radix* provides new information regarding the evolution of macrostomy. Though macrostomy has previously been used to define a monophyletic clade of derived snakes (e.g. [10]), recent phylogenies (e.g. [11,26]) indicate that, rather than being a strictly synapomorphic condition, macrostomy likely evolved independently on multiple occasions in snakes (figure 8). This hypothesis of homoplasy is supported by the fact that two of the major groups that compose the 'macrostomatans'—booids (boas and pythons [10]) and caenophidians (acrochordids and colubroids [10])—achieve macrostomy via different ontogenetic pathways [6]. In the first pathway (figure 8*a*), the jaw joint shifts posteriorly due to posterior elongation of the supratemporal. The quadrate, though also elongated, is laterally displaced at the mandibular condyle but remains in the same transverse plane (i.e. perpendicular to the skull in lateral view) [6]. In the second pathway (figure 8*b*), the supratemporal is not posteriorly elongated throughout ontogeny; instead, the jaw articulation shifts posteriorly due solely to ventral rotation and lengthening of the quadrate [6]. These pathways occur in booids and caenophidians, respectively [6].

Interestingly, while *T. radix* is a member of the Caenophidia, it does not follow the typical caenophidian mode of jaw development as proposed by Palci *et al.* [6]. Instead, this taxon exhibits a combination of the two ontogenetic patterns, with both a relative posterior elongation of the supratemporal and a relative lengthening and rotation of the quadrate causing posterior displacement of the jaw articulation (figures 7*a*–*c* and 8*c*). This possible third developmental pathway has been briefly noted in a few other caenophidians, such as *Homalopsis buccata* [6]. However, this genus is not closely related to *Thamnophis* within the Caenophidia [27] (figure 8). The presence of a similar ontogenetic pattern between *Thamnophis* and *Homalopsis* therefore suggests additional homoplasy in the evolution of the macrostomatan condition, causing an even greater deviation from the original interpretation of macrostomy as having a single evolutionary origin. This recognition of greater-than-anticipated osteological variation adds to recent descriptions of greater-than-anticipated variation in soft tissues associated with macrostomy, thus reinforcing novel realizations of the complex evolutionary history of this morpho-functional system [9].

Other major ontogenetic changes to the skull of *T. radix* are associated largely with increases in ossification, development of muscle crests and integration between different skull elements. These changes—such as the closure of fontanelles, increased the prominence of muscle crests and more strongly developed articulating surfaces—have been noted in other taxa, both snake and otherwise (e.g. [2,6,29]), indicating that these are highly conserved patterns of development among vertebrates. Changes to the general shape of the *T. radix* skull—such as an overall dorsoventral flattening of the braincase—are also consistent with conservative ontogenetic patterns noted in various snake taxa (e.g. [2,6]).

## 4.2. Suspensorium

Previous studies have observed a lack of contact between the quadrate and otic capsule in embryonic snakes, with the quadrate only ever articulating dorsally with the supratemporal (e.g. [30]). However, the new data reported here on the suspensorium (see §3.6) indicate that, through the observed ontogenetic stages, the suspension of the lower jaw shifts from a prootic-supratemporal articulation of the quadrate to a supratemporal-only articulation (figures 3 and 7). This quadrate-prootic contact—which is most prominent in the embryo—thus revises our current knowledge of this aspect of braincase development and indicates heretofore unrecognized variation in the ontogeny of these structures. Furthermore, given this observed

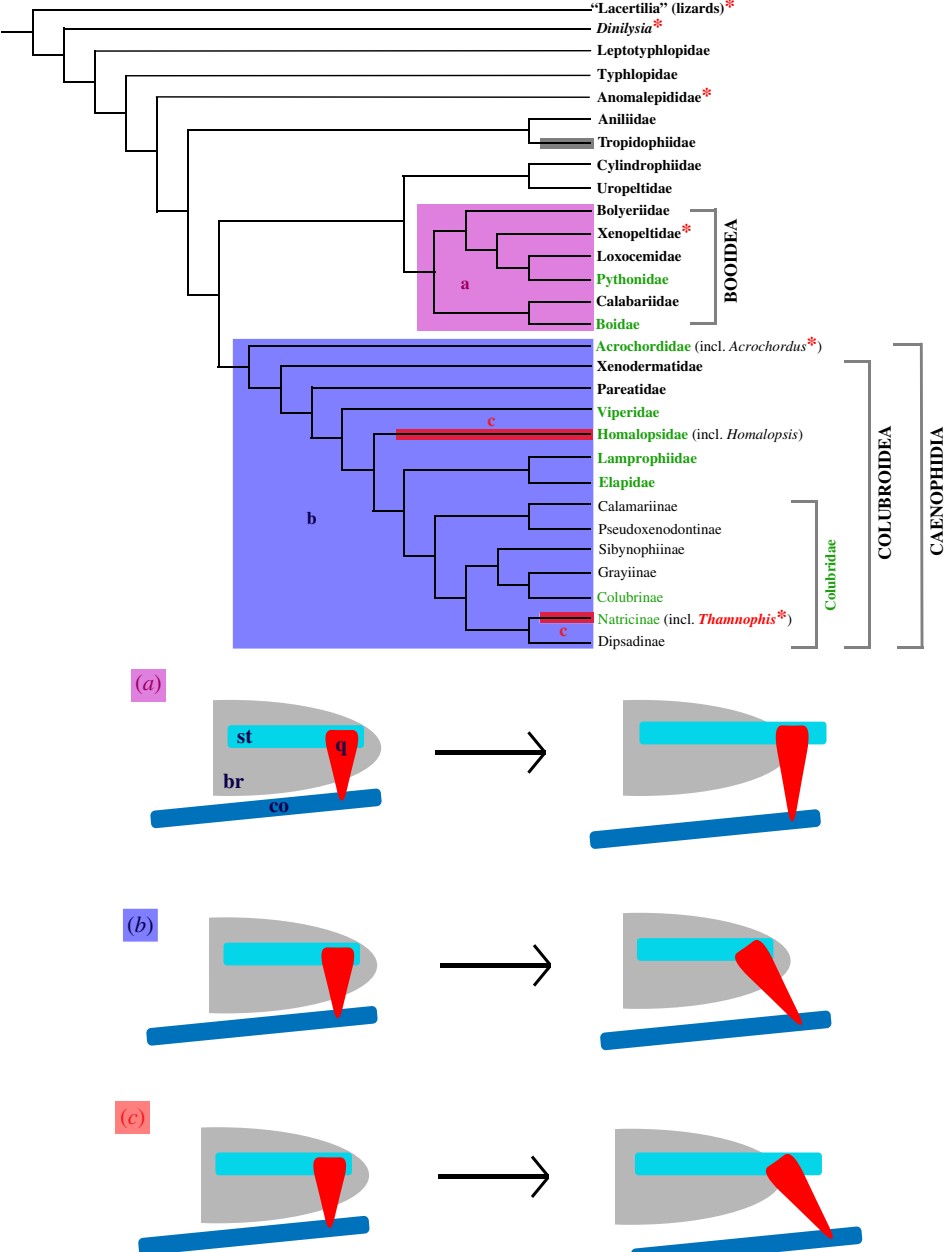

**Figure 8.** Phylogeny of Serpentes, highlighting the distribution of presence of the CCF and the hypothesized evolution of macrostomy (modified from the evolutionary pathways proposed by Palci *et al.* [6]). Groups marked with an asterisk (*) contain taxa lacking a CCF; note that *Thamnophis* is the only colubroid recognized as lacking this feature. The shaded boxes indicate taxa exhibiting macrostomy. These taxa do not form a monophyletic group, suggesting independent evolution of this condition; this interpretation of homoplasy is supported by the fact that different groups develop macrostomy via different developmental pathways. Each pathway is represented by a different colour of box and is depicted schematically in the corresponding diagram below the phylogeny. The grey box for Tropidophiidae indicates a lack of data regarding the development of macrostomy in this clade. Macrostomatan family and subfamily names in green represent groups for which representatives have been directly observed, either by us or in other studies [6]. (*a*) Booids (boas, pythons and relatives) achieve macrostomy via elongation of the supratemporal and quadrate, with the quadrate becoming ventrolaterally deflected but remaining in the same transverse plane. (*b*) Caenophidians (acrochordids and colubroids) increase gape size via elongation and rotation of the quadrate, while the supratemporal is proportionally unchanged. (*c*) *Thamnophis* and *Homalopsis*— both caenophidians—achieve macrostomy via both rotation of the quadrate and elongation of the supratemporal and quadrate. This pathway may be more widespread than previously recognized, but has currently only been observed in these two genera. br, braincase; co, compound bone; q, quadrate; st, supratemporal. Phylogenetic relationships mainly from [11], with phylogeny of Colubridae from [27] and placement of *Dinilysia* from [21] and [28]. Distribution of the CCF is as presented by Palci & Caldwell [21], with incorporation of new data regarding *T. radix*. Hypothesized pathways of macrostomy development are as discussed by Palci *et al.* [6].

quadrate-prootic contact, it is likely that, in earlier developmental stages of *T. radix* not yet observed by us, the quadrate articulates only with the endochondral braincase; this makes sense as the chondrocranium forms before any dermatocranial elements such as the supratemporal, thus the quadrate could only articulate with the chondrocranial prootic-otooccipital prior to development of the supratemporal through ontogeny. The eventual loss of the prootic articulation in *T. radix* is clearly linked to positive allometric growth of the supratemporal; as the supratemporal elongates posteriorly, it carries the quadrate away from any contact with the prootic.

In evolutionary terms, linking ontogenetic patterns to the interpretation of adult stages in other snakes makes it possible to consider the articulation of the quadrate to the chondrocranium in other extant snakes, most importantly here, scolecophidians. In almost all known scolecophidians, the supratemporal is lost; however, even when it is present, this splint-like 'supratemporal' does not articulate with the quadrate and articulates only with the chondrocranium (e.g. [31,32]). If the plesiomorphic snake condition for the development of the quadrate is as observed in *T. radix*, with the quadrate changing from a prootic-supratemporal articulation in embryos and juveniles to a supratemporal-only articulation in adults—rather than as described by Kamal & Hammouda [33], in which the quadrate does not contact the chondrocranium—this condition in scolecophidians can be interpreted as heterochronic. More specifically, the data present in *T. radix* embryos and juveniles would reveal the scolecophidian skull to be paedomorphic, as adult scolecophidians retain the putative plesiomorphic embryonic condition in which the quadrate articulates with only the chondrocranium. This is an important observation, especially when coupled with the observation of Kamal [30] regarding the posterior rotation of the mandibular condyle of the quadrate in snake ontogeny; i.e. the mandibular condyle is oriented anteriorly in early embryonic snakes and rotates posteriorly through ontogeny to a vertical position in more basal forms, to a posterior position in highly derived forms [33]. Coupled with the posterior elongation of the supratemporal through ontogeny, this means that the quadrate can be displaced from the prootic, displaced past the basioccipital condyle, and have the mandibular condyle pushed even further posteriorly (see [34]).

## 4.3. Crista circumfenestralis

The CCF is a bony crest—formed by the cristae prootica, interfenestralis and tuberalis—that surrounds the fenestra ovalis and lateral aperture of the recessus scalae tympani (LARST) and, in its extreme form, forms a bony dome covering the stapedial footplate and fenestra ovalis, creating the juxtastapedial recess [19,21]. Certain treatments of the CCF consider this structure an unequivocal synapomorphy present in all snakes (e.g. [19,35]). However, a recent analysis of the CCF concluded that, while this structure is either partially or completely present in most snakes, it is absent in the basal snake *Dinilysia* as well as in extant taxa such as *Anomalepis*, *Xenopeltis* and *Acrochordus* [21] (figure 8). Of particular note among these different studies is the distribution of this character within the Colubroidea, the most deeply nested major clade of snakes. Despite their varying conclusions, none of the aforementioned analyses have disputed the idea that the CCF is universally present, without exception, in all colubroids. As such, the presence of this feature has historically been strongly associated with a derived phylogenetic position within snakes.

However, despite its deeply nested position within the Colubroidea (e.g. [11,27,28,36]), *T. radix* is highly unusual in lacking the CCF, a phenomenon heretofore unrecognized among this clade. Instead, the associated cristae remain unelaborated and unexpanded, leaving the stapedial footplate, fenestra ovalis and LARST completely exposed in lateral view (figures 3*a–c* and 6*a–c*). Regarding the morphology of this structure, *T. radix* therefore resembles the plesiomorphic character state seen in lizards and the fossil *Dinilysia*, rather than the derived condition universally present in other colubroids (figure 8).

In light of this discovery, three scenarios exist to explain the character distribution of the CCF within the Colubroidea. In the first scenario, *T. radix* lacks a CCF because it is not, in fact, a colubroid; instead, this taxon diverges basally to the Colubroidea, thus preserving the correlation of the presence of the CCF with advanced phylogenetic placement. In the second scenario, *T. radix* remains within the Colubroidea and is thus the only colubroid to lack a CCF. Although we did not (nor did any other previous study) assess this character at a species level across all species of colubroids (a seemingly impossible task at the moment), the current consensus in the literature is that presence of the CCF in adults is widespread among colubroids (e.g. [19,21,35]). In order to achieve this adult condition, the organism must necessarily pass through an earlier ontogenetic stage in which the CCF is absent or undeveloped, as is characteristic of any endochondral ossification. As such, it is safe to infer that, for colubroids, the most frequent condition is to lack a CCF in the embryonic (and possibly also juvenile) stage and to possess a CCF in the adult stage. We therefore suggest that, in the phylogenetic context of colubroids, the absence of a CCF in adults of *T. radix* (figures 3 and 6) can be seen as the derived condition, thus representing a paedomorphic pattern when

compared to other colubroids, i.e. retention of a plesiomorphic embryonic or juvenile feature in an adult individual. In this scenario, the assumption that derived snakes possess a CCF still generally holds true, although there is now an exception to this pattern of character distribution. In the third and final scenario, *T. radix* is not the only colubroid to lack the CCF; other colubroid taxa also lack this feature and just have yet to be noted. In this scenario, rather than having the CCF be uniformly present within the most derived clade of snakes, the distribution of this character state is now variable; the CCF is therefore no longer a uniform marker of phylogenetically derived status.

Determining which of these scenarios holds true has strong implications for our current understanding of snake braincase evolution. Each scenario holds different consequences not only for whether or not the CCF can still be considered a widely distributed condition among Colubroidea, but also for the overall evolutionary plasticity of this particular feature. Regarding scenario 1, given that the genus *Thamnophis* has consistently been recovered as a colubroid, including in recent squamate phylogenetic analyses (e.g. [28]), this explanation for character distribution of the CCF can, therefore, be quickly rejected. However, scenarios 2 and 3 are more difficult to discern.

Our ability to determine whether or not *T. radix* is unique among colubroids in lacking a CCF depends entirely on morphological assessment, or re-assessment, of other colubroid taxa. For example, if a well-known snake taxon such as *Thamnophis* has only now been recognized as lacking a CCF, it is therefore strongly possible that previous studies have simply assumed the CCF to be present or have not examined this character in their descriptions of other extant snake taxa. In this case, scenario 3 would prevail. However, it is also possible that assessment of other snake taxa for this character may reveal that *Thamnophis* is indeed unique among the Colubroidea in lacking a CCF, thus fulfilling scenario 2.

A well-studied example of the absence of the CCF occurs in *Acrochordus*, the sister group to the Colubroidea (figure 8). This taxon is therefore useful for comparison to *Thamnophis* and for assessment of the aforementioned scenarios regarding character distribution of the CCF. Initial analyses of the CCF in *Acrochordus* preliminarily discussed its absence as an autapomorphy caused by paedomorphosis, but ultimately concluded that this absence instead represents a continuation of the plesiomorphic character state found in lizards [37]. The absence of the CCF in *Acrochordus* was therefore considered consistent with a basal placement of *Acrochordus* among snakes, similar to the first scenario proposed herein for *Thamnophis* [37]. However, this basal placement of *Acrochordus* was later rejected, due largely to the strong phylogenetic evidence placing *Acrochordus* at a far more deeply nested position as the sister to Colubroidea [38]; again, this rejection parallels our rejection of scenario 1 in *Thamnophis* (i.e. lack of a CCF does not inherently indicate basal phylogenetic status). This later analysis re-characterized the otico-occipital region of the skull of *Acrochordus*—focusing on the persistent lack of division of the metotic fissure—as paedomorphic, making *Acrochordus* unique among snakes in retaining this embryonic morphology—similar to our scenario 2—though it lacked ontogenetic data supporting this conclusion [19,38].

This lack of data was addressed by a more recent analysis of cranial development in a partial developmental series of *Acrochordus* embryos [19]. Somewhat confusingly, this analysis recognized that the embryonic individuals exhibit the same condition of the CCF as the adults, but concluded that the ear region in *Acrochordus*—including the lack of a CCF—is neither plesiomorphic nor paedomorphic, but instead represents a secondarily derived, autapomorphic condition [19]. However, this conclusion erroneously implies that autapomorphy and paedomorphosis are mutually exclusive. Autapomorphy refers to a derived feature present in a single taxon and is, therefore, an observation of character distribution. Paedomorphosis refers to the presence or retention of an embryonic or juvenile trait of an ancestral form in the adult of a descendant form and thus refers to a process through which different morphological states arise. Therefore, rather than being mutually exclusive, paedomorphosis and autapomorphy, in fact, complement each other regarding the CCF. In the case of *Thamnophis*, this taxon is currently the only colubroid recognized to lack the CCF and is thus autapomorphic relative to other colubroids based on our current understanding of the distribution of this character. From our data, the persistent absence of the CCF throughout all three ontogenetic stages (a condition shared with numerous non-ophidian lizards and with other more phylogenetically basal snakes such as *Acrochordus* [19]) demonstrates this condition in *Thamnophis* to be a case of paedomorphosis. Essentially, autapomorphy describes the unique distribution of this character relative to other colubroids, while paedomorphosis provides a process through which this distribution arises. This same combination of pattern and process also applies to *Acrochordus*: at the time of the aforementioned study, *Acrochordus* was considered unique among snakes in lacking a CCF, rendering this condition autapomorphic. Furthermore, as recognized by the authors of that study, the adult condition matches the plesiomorphic snake embryonic condition, indicating paedomorphosis as the process giving rise to this condition.

It should be noted that, while *Acrochordus* was initially considered to be the only snake lacking the CCF (e.g. [19,37,38]), this absence has since been noted in other snake taxa [21] (figure 8). Returning to our scenarios regarding the distribution of the CCF and expanding these scenarios to apply to the overall distribution of the CCF among all snakes, this represents a rejection of scenario 2 and confirmation of scenario 3. In other words, we can reject the absence of the CCF in *Acrochordus* as a unique occurrence among snakes and instead recognize that this absence is more widespread than previously thought. Narrowing down to our original scope in discussing the distribution of the CCF specifically among colubroids, this recognition of greater-than-expected plasticity in the evolution of the CCF suggests that scenario 3 is more likely than scenario 2; i.e. based on its varied distribution among snakes as a whole, we expect the distribution of the CCF to also vary among colubroids as per scenario 3, rather than being entirely unique to *Thamnophis*.

As stated previously, this prediction can only be confirmed or rejected via re-examination of this feature in other colubroids. Preliminary comparison to *T. sirtalis parietalis* reveals this taxon to also lack a CCF (C.R.C.S., personal observation, 2018), suggesting that this absence may be widespread within the genus *Thamnophis* as a whole. Since our study focuses specifically on the ontogeny of *T. radix*, our sampling efforts were in turn focused on this species. However, denser sampling of other colubroid species—including other species of *Thamnophis*—is a key avenue of future research to further investigate the possibilities we have raised herein. Regardless of whichever scenario ultimately applies, the fact that *T. radix*—an otherwise derived snake—lacks a CCF—a presumed uniformly advanced character—brings into question our current assumptions of character distributions among snakes and forces a re-examination of current paradigms regarding this group.

# 5. Conclusion

We herein present the first in-depth analysis of the skeletal ontogeny of *T. radix*. This represents the first study to perform a full micro-CT segmentation of all skull elements of any non-adult snake, as well as the first ontogenetic analysis of any squamate to encapsulate all three major ontogenetic stages—i.e. embryo, juvenile and adult—within a single study. This study, therefore, contributes towards a recently increasing emphasis on discerning major patterns of ontogeny among snakes (e.g. [6]).

While many of the ontogenetic changes undergone by *T. radix* are consistent with patterns of ontogeny noted in other taxa, certain peculiarities raise interesting questions regarding the evolution of important diagnostic snake characters. Developmental changes resulting in macrostomy in the adult *T. radix* do not conform to either of the two main independent ontogenetic patterns recognized for this feeding mechanism in other macrostomatan snakes. The presence of this novel ontogenetic pathway in both *T. radix* and an unrelated genus, *Homalopsis* [6], indicates that the evolution of the macrostomatan condition may be more complex than previously anticipated. Our observations of suspensorium development in *T. radix* also enable novel interpretations of paedomorphosis in the evolution and ontogeny of the scolecophidian skull. Furthermore, the absence of a CCF in *T. radix*—an unexpected plesiomorphic character state previously unrecognized within Colubroidea—challenges current assumptions which consider the CCF to be universally present among all lineages within this clade [21] and provides evidence for paedomorphosis in the evolution of this trait.

Altogether, our findings indicate the continued importance of anatomical description and re-assessment in evolutionary biology, as even a taxon as well known as the garter snake displays unexpected features that challenge previous paradigms of snake cranial development and cranial trait evolution.

Data accessibility. The data supporting this study are available via the Dryad Digital Repository: https://doi.org/10.5061/dryad.n50st0n [15].

Authors' contributions. C.R.C.S. conducted the micro-CT scanning, segmentations and descriptions and drafted the manuscript. M.W.C. and T.R.S. conceived and supervised the project. M.R.D. developed the protocol for and conducted the micro-CT scans. All authors contributed to writing and discussions and gave their final approval for publication.

Competing interests. We declare that we have no competing interests.

Funding. This study was supported by a Natural Sciences and Engineering Research Council of Canada Discovery Grant (23458) and a Chair's Research Allowance to M.W.C.

Acknowledgements. We thank Aaron LeBlanc and Ilaria Paparella for their help in conducting the micro-CT scans. We also thank Alison Murray and Braden Barr for providing access to specimens from the University of Alberta Museum of Zoology, and for their assistance with the UAMZ database. We thank Dr Alex Deufel and another, anonymous reviewer for their comments, which greatly improved the quality of this manuscript.

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
