## [Reviewer comments · Royal Society Open Science]

Review History

RSOS-182228.R0 (Original submission)

Review form: Reviewer 1

Is the manuscript scientifically sound in its present form?

No

Are the interpretations and conclusions justified by the results?

No

Is the language acceptable?

Yes

Is it clear how to access all supporting data?

Not Applicable

Do you have any ethical concerns with this paper?

No

Have you any concerns about statistical analyses in this paper?

No

Recommendation?

Major revision is needed (please make suggestions in comments)

Comments to the Author(s)

Strong et al. present a nice overview of the ontogenetic development of the skull in a modern colubrid snake, *Thamnophis sirtalis*. I found it intriguing that despite the fact that this is a very well known snake species so much can still be found when closely observing its bones and their development. Things like the contact between quadrate and prootic in the snake embryo and the lack of a crista circumfenestralis in the adult are certainly novel reports and are worth publishing. Having said this, I also found several issues with this manuscript, which preclude publication in its current state.

Here is a summary of the major points:

- 1) Some of the terminology used is inadequate, like use of terms such as “orbitosphenoidal plate” or “incisive process”.
- 2) There is confusion between fusion of elements and simple sutures.
- 3) The manuscript would be greatly improved if the lower jaws were rendered separately in lateral and medial view. This way the authors could properly illustrate all the elements they describe, including angular and splenial, and would not have the lower jaws obscuring most of the anatomy of the skull in ventral view (e.g. palatines, parabasisphenoid, and basioccipital cannot be adequately seen because of this). Moreover, the authors took the time to segment all the individual skull bones from the three developmental stages, but then simply figure the whole skulls without showing the internal details of any of the elements. If the authors only wanted to provide colorized versions of the skulls, then Adobe Photoshop could have provided the very same results in a fraction of the time. I suggest that the authors take advantage of the fact that they have digitally segmented individual elements in 3D and provide rendered images to illustrate internal anatomical details that at present are discussed but not figured anywhere (at least as supplementary figures).
- 4) I noticed an issue with tooth counts. The authors appear to have counted only those teeth that are actually present and ankylosed to the dentigerous elements, when instead they should have provided counts of tooth positions. This resulted in very inconsistent tooth counts (i.e. very low numbers in the juvenile compared to the adult), when in fact the numbers of tooth positions are expected to be very close, if not identical (at least for dentaries and maxillae). Moreover, tooth counts could be affected by intraspecific variability (to a small extent), but the authors provide counts for only 3 of their eight specimens. This way it is not possible to tell whether different tooth counts could be caused by simple random intraspecific variation rather than by ontogenetic change.
- 5) There is confusion revolving around the concept of paedomorphosis and on how it is detected. The authors assume that if a trait is retained from an embryonic or juvenile stage into the adult stage of a given species then the adult of that species is paedomorphic. This is incorrect. See details below (e.g. Page 21, line 10). This misconception affects their interpretation of the lack of a CCF in *Thamnophis* as a plesiomorphic feature, and also their interpretation of the

scolecophidian skull as the result of paedomorphosis. Because the authors cannot demonstrate that the lack of a CCF or the contact between quadrate and prootic are plesiomorphic embryonic/juvenile traits in snakes, then they cannot reach the conclusion that *Thamnophis* and scolecophidians are paedomorphic for these traits. The authors can only raise the possibility (i.e. formulate a hypothesis that requires further data to be supported).

6) Figure 7 is quite a problematic figure. It deceptively suggests that the authors have way more information on the distribution of the various modes of jaw suspension than they actually have. I have extensively discussed the issues relating to this figure in one of my comments below.

Below I provide detailed comments on all major and minor issues as they present themselves in the manuscript:

Page 1, line 53: than in most other (insert "most").

Page 2, line 17: replace "many" with "some" (there aren't really that many studies).

Page 6, line 6: "increased integration of the premaxilla with the rest of the snout". I would argue that this is not true. The images show that the premaxilla is very loosely connected to the rest of the snout in all three specimens/stages, and the only contact is between premaxilla and anterior tip of the septomaxilla.

Page 6, line 6: the authors mistakenly use the term "incisive process" to refer to the premaxillary process of the vomer (the "incisive process" is a process on the ventral side of the premaxilla of some squamates other than snakes). Please replace "incisive process" with "premaxillary process" here and in any following instance (lines 17, 26).

Page 6, line 13: "posterior premaxillary notch which will eventually accommodate the septomaxilla and vomer".
The notch does not accommodate any part of those bones.

Page 6, line 22: "premaxilla's nasal process expands both dorsoventrally and anteroposteriorly". I believe this is incorrect, it expands mediolaterally and anteroposteriorly. It is quite clear that the relative height of the nasal process remains the same if compared to septomaxilla, nasal, and prefrontal.

Page 6, lines 24-31: "Extension of the anterior processes of the septomaxilla and incisive processes of the vomer relative to the rest of these bones creates stronger contact - and thus improved integration - between the premaxilla and the rest of the snout (Fig. 3C)."
This is incorrect, see previous comment on line 6.

Page 6, lines 35-36: "the dorsal laminae are only present as a cluster of unfused ossifications". This is inaccurate, there is no cluster of unfused ossifications, which seems to imply the presence of multiple centers of ossification, but only a weakly ossified surface that when rendered digitally appears full of holes, most likely as a rendering artifact.

Page 6, lines 38-39: "dorsal laminae distinctly present and fused to the vertical laminae".

This is inaccurate. The dorsal laminae are not individual elements separate from the vertical laminae, they are part of the same element (the nasal), which is in the process of being mineralised. It follows that there can be no fusion between parts of the same element.

Page 7, line 13: sutured (not "fused"). The frontals never fuse in snakes.

Page 7, line 22: The frontals are not completely sutured to each other (don't write "fuse").

Page 7, line 24: please remove "at which point a deep sagittal sulcus is present." (that is not a "sulcus", is the suture between the bones).

Page 7, line 33: I would suggest using the term "optic fenestra" (not "foramen", foramina are small round holes, fenestrae are large gaps, not necessarily round).

Page 7, line 36: Please replace the term "orbitosphenoidal plate" with "descending flange of the frontal". The only literature reference I could find where the term "orbitosphenoid plate" was used in the description of a snake skull dates back to 1831 (and also refers to something different). Please replace "orbitosphenoidal plate" with "descending flange of the frontal" also in the following instances (same page lines 40, 52; and page 8, lines 3, 8).

Page 7, line 43: parabasisphenoid (not "basisphenoid"; the bone referred to results from fusion of parasphenoid and basisphenoid). Please replace "basisphenoid" with "parabasisphenoid" also in the following instances.

Page 8, line 8: emargination (not "notch").

Page 8, line 19: Add "(Fig. 1)" after "juvenile stages".

Page 9, line 6: replace "expanded" with "bent" or "recurved".

Page 9, line 22: "The palatine tooth row bears 15 teeth in the embryo, 14 teeth in the juvenile, and 16 teeth in the adult".

The palatines are hidden by the lower jaws, so it is impossible to verify the tooth counts from the figures provided. In fact, the lower jaws hide a great deal of the ventral anatomy of the skulls; it would be better to have them rendered separately next to the skulls, this way the authors could also illustrate the medial view of the lower jaws, which at present is not shown.

Based on what argued below for the tooth counts on dentaries and maxillae (Page 15, lines 31 & 52), I would suggest the authors to double-check these counts as well, in order to make sure that they actually provide counts of tooth positions and not just of preserved teeth.

Finally, the numbers presented here, if correct, do not appear to follow an ontogenetic trend (15-14-16). I suspect that there may be some intraspecific variation in there as well. In order to see if this is the case, the authors should provide tooth counts for all specimens that they CT scanned (3 embryos, 2 juveniles, and 3 adults). With only one specimen per ontogenetic stage it is hard to discriminate between what is actually ontogenetic change and what is simply random intraspecific variation. The same holds for the tooth counts in dentaries and maxillae. Moreover, the authors should also provide the counts for left and right counterparts of each bone, in case the numbers are different.

Page 9, lines 31-35: "The braincase undergoes a general dorsoventral flattening throughout ontogeny, from an initially globular appearance in the embryo to a dorsoventrally compressed and anteroposteriorly elongated form in the adult (Fig. 3A-C)."

The compression of the braincase in the adult is mostly anterior, the posterior half is as deep in the adult as it is in the juvenile (the length/max-depth ratio of the skull is about 2.1 in both). Please be more precise in your description.

Page 9, line 42: insert "posterior to" before "lateral aperture" (the jugular foramen is not posteroventral to the lateral aperture, it is posterior to it).

Page 10, line 19: "The jugular foramen also develops internal struts that subdivides it into three openings internally".

"Subdivide" (not "subdivides", the word "struts" is plural), and you could mention which nerves go through those foramina, they are well known in the snake literature.

Page 10, line 36: "stapedial footplate and LARST exposed in lateral view (Fig. 5A-C)."

No stapedial footplate is shown in figure 5, you could add a reference to figure 3 to illustrate your point.

Page 10, lines 42-47: "The nuchal crest that runs anteromedial-posterolateral above the fenestra ovalis is minimally developed in the embryo, resulting in a globular process above each fenestra ovalis".

Please add a figure reference and label the nuchal crest in it for those people unfamiliar with the term.

Page 10, line 49: I would replace "this process" with "it".

Page 11, line 6: Delete "Similarly" (there is no similitude here).

Page 11, line 36: "a narrowing of the large ventral notch of the supraoccipital". Between the otic capsules? If so, please specify, it is unclear which notch the authors are referring to.

Page 11, lines 40-42: "Internally, the supraoccipital undergoes a general elaboration of internal crests and subdivisions." This sounds extremely vague and superficial. What kind of elaboration? Which crests? Which subdivisions? Please be precise in your description or remove this sentence.

Page 11, line 52: "contact with the prootic is stronger than in the juvenile but still not complete."

Do the authors mean: still not complete internally? Externally it looks complete in the figures.

Page 12, line 3: insert "dorsally" after "concave".

Page 12, line 12 and following: What is the point of segmenting all the individual bones if then internal anatomical details like these are not properly figured and labeled? I would recommend adding some additional figures, if not to the main text at least as supplementary information.

Page 12, line 22: insert "complex" after "dorsalis" (the cid is a muscle complex, not a single muscle). Moreover, a better reference for the meaning of cid would be:

Rieppel, O. (1979). The evolution of the basicranium in the Henophidia (Reptilia: Serpentes). *Zoological Journal of the Linnean Society*, 66(4), 411-431.

The Rieppel (1979) paper the authors are referencing is on scolecophidians, and while it mentions the cid, it does not explain its meaning.

Page 12, line 35: "parabasisphenoid" instead of "basisphenoid".

Page 12, line 35 and following: "The junction between the parasphenoid process and the body of the basisphenoid also becomes elaborated, with minor lateral projections in the embryo and juvenile becoming enlarged into prominent triangular projections in the adult. These projections contribute to an increase in sutural contact with the anteroventral parietal."

All these details are not visible because hidden by the lower jaws (see also comment above on Page 9, line 22).

Page 12, line 52: “parabasisphenoid” instead of “basisphenoid”.

Page 13, line 10: delete “dorsal and” (there are no muscle crests on the dorsal surface of these bones).

Page 13, line 19: “a slight dorsal longitudinal crest on the juvenile occipital condyle is extended into a prominent crest running from the centre to the posterior margin of the dorsal basioccipital.” I don’t quite understand what the authors are describing here, and there is no figure to illustrate this feature. A prominent dorsal longitudinal crest running along the dorsal surface of a snake basioccipital would be a very unusual, if not unique, feature. If this feature is real it should be figured.

Page 13, line 33: “dorsal cephalic condyle”, because there is no “ventral cephalic condyle” the word “dorsal” is redundant, please remove.

Page 13, line 52: same as above, there is no “dorsal mandibular condyle”, so the word “ventral” is redundant and can be removed.

Page 13, line 49: “The quadrate also undergoes a progressive deepening of its ventral mandibular condyle”. What do the authors mean by “progressive deepening”? it’s not clear, a dorsoventral expansion is not apparent from the figures, perhaps a mediolateral expansion?

Page 15, line 6 and following: Here the authors again describe features that are not illustrated. This could be easily remedied by rendering the lower jaws separately in lateral and medial views.

Page 15, line 31: “The dentary bears 18 teeth in the embryo, 18 teeth in the juvenile, and 28 teeth in the adult.”

These tooth counts cannot be correct. I can count at least 25 tooth positions in the juvenile illustrated in fig. 3. The authors should count tooth positions, not just fully erupted teeth that are ankylosed to the margin of the dentary. As far as I know, tooth counts on the marginal tooth-bearing elements are very conservative in snake species and across ontogeny, I would be really surprised if *Thamnophis* would show such differences in tooth counts.

Page 15, line 52: “The maxilla bears 16 teeth in the embryo and 18 teeth in the juvenile and adult.” Same problem as above. I can count at least 23 tooth positions in the embryo and a similar number in the juvenile. Please check these numbers again.

Page 16, lines 8-15. Angular and splenial are not figured nor labeled anywhere. Again, the manuscript would be much improved if the lower jaws were rendered separately, so that the ventral aspect of the skull was not obscured, and all elements of the lower jaws could be properly illustrated.

Page 16, lines 19-26: aren’t these statoliths present in the adults as well? I would be surprised if this was not the case.

Page 17, line 8: “pterygoid and dentary tooth rows, which increase from 18 to 28 and from 22 to 26 teeth, respectively”.

Based on the likely erroneous tooth counts provided by the authors for dentary and maxilla, I would suggest to check these numbers again.

Page 17, line 47: “the quadrate, though also elongated, remains upright.”

This is inaccurate. In some booids, such as *Calabaria*, the quadrate is not elongated and remains

upright; however, when the quadrate is elongated in the adult, then it does not remain upright; in fact, a longer quadrate becomes tilted sideways, so that the condyles are approximately in the same vertical transverse plane, but are not aligned vertically one above the other (i.e. the condyles appear to be one above the other only in lateral view). Obviously, a longer quadrate has to be displaced either laterally or posteriorly, or it will create a gap between upper and lower jaws.

Page 18, line 6: I am not sure why the authors think that is the “typical” caenophidian mode. There has been no systematic survey of jaw attachment modes in snakes as far as I am aware, and the references listed here only examined a limited number of species.

Page 19, line 3: “the suspension of the lower jaw shifts from a prootic-supratemporal articulation of the quadrate to a supratemporal-only articulation (Fig. 3 and 6).”

If accurate, this is a very interesting observation. A contact between quadrate and otic capsule is thought to be absent in snake embryos (see for example Kamal and Hammouda (1965:291), who studied the colubroid *Psammophis*: “It is of importance to notice that the quadrate during ontogeny is quite apart from the neurocranium and bears no connection with the auditory capsule. This is correlated with the absence of the otic process and the subsequent loss of the crista parotica commonly present in Lacertilia. Only in the fully formed chondrocranium, after the quadrate has reached its definitive position, its dorso-medial margin is attached to the hind end of the auditory capsule by the supratemporal (tabular) bone.”

Kamal, A. M., & Hammouda, H. G. (1965). The development of the skull of *Psammophis sibilans*. II. The fully formed chondrocranium. *Journal of Morphology*, 116(2), 247-295.

Because of this, some readers may question the authors’ observation. The fact that the quadrate overlaps the prootic in lateral view doesn’t mean that they are in contact, let alone in articulation. Furthermore, in Fig. 6 the tilted perspective seems to suggest that the quadrate is only in articulation with the supratemporal in all three developmental stages, and has no contact with the prootic (but could be an artifact of perspective). And finally, if an articulation between quadrate and prootic was present, then one may expect an articulatory facet on the prootic in fig. 5, but such facet is not apparent. I would suggest the authors to provide as supplementary data 3D surface files (in ply or stl format) of all three embryos that have been CT-scanned, in order to unequivocally show that an articulation between quadrate and prootic is indeed consistently present.

Page 19, lines 6-17: Here the authors speculate about the fact that the quadrate in earlier stages of embryonic *Thamnophis* is likely in articulation with the braincase. As I mentioned above, this goes against what observed by Kamal and Hammouda (1956), and should therefore be discussed in more detail, pointing out the contradiction with that study.

Page 19, line 29: remove “though there are exceptions” (redundant, you already said “in almost all”).

Page 19, line 33: “The data present in the *T. radix* embryos and juveniles reveal that the suspensorium of scolecophidians can be interpreted as heterchronic.”

I recommend inserting the sentence: “, if representative of the plesiomorphic embryonic snake condition,” just after “juveniles”. Paedomorphosis is inferred after observing plesiomorphic juvenile features in adults, thus adult scolecophidians can be considered paedomorphs only if the plesiomorphic embryonic condition for snakes matches what described in *Thamnophis* rather than what described in other snakes (such as *Psammophis*, the snake described by Kamal and Hammouda [1965]).

Page 19, line 47: replace “juvenile” with “early embryonic”. Such anterior orientation in snakes is

only visible in developmental stages occurring before those investigated here by the authors; see Kamal & Hammouda (1965) and Boughner et al. (2007).

Boughner, J. C., Buchtová, M., Fu, K., Diewert, V., Hallgrímsson, B., & Richman, J. M. (2007). Embryonic development of *Python sebae*-I: Staging criteria and macroscopic skeletal morphogenesis of the head and limbs. *Zoology*, 110(3), 212-230.

Page 21, line 10: remove “thus” (“thus” implies that the following sentence is the only possible consequence of what is stated before, but in fact the authors later present another scenario where *T. radix* is still within colubroidea but is not necessarily the only colubroid without a CCF).

Page 21, line 10 and following: “Given that the CCF is absent throughout the embryonic and juvenile forms (Fig. 5), the continued absence of this feature in the adult snake is the result of paedomorphosis, i.e., retention of an embryonic or juvenile feature in an adult individual.”

This is a misinterpretation of the concept of paedomorphosis. Paedomorphosis is not inferred by the retention of a feature from juvenile to adult, but by the retention in an adult of what is generally assumed to be a juvenile ancestral feature (i.e. the plesiomorphic juvenile condition). In order to say that the absence of a CCF is a juvenile feature, the authors should first show that (at least closely related) juvenile snakes typically lack it (not just *Thamnophis*). This would identify the plesiomorphic condition (ideally with support from a phylogeny). Then, and only then, they could argue that the absence of a CCF in adult *Thamnophis* is a paedomorphic feature. Otherwise anything could be a paedomorphic feature, even the presence of two eyes.

To quote from McNamara (1986:5):

“If a descendant passes through fewer stages of ontogenetic development than its ancestor, the descendant adult form will have morphological characteristics which occurred in juveniles of the ancestor. This phenomenon has been termed paedomorphosis.”

Or using Klingenberg’s (1998:83) words:

“The morphological outcomes of changes in rates and timing of development are paedomorphosis or peramorphosis; they are identified by comparisons of ancestors and descendants in relation to the ancestral ontogeny. A descendant is paedomorphic if its later ontogenetic stages retain characteristics from earlier stages of an ancestor”.

In summary, paedomorphosis requires a phylogenetic context, and the ontogenetic trajectory of an individual species is not sufficient to come to any conclusion regarding heterochrony.

McNamara, K. J. (1986). A guide to the nomenclature of heterochrony. *Journal of Paleontology*, 60(1), 4-13.

Klingenberg, C. P. (1998). Heterochrony and allometry: the analysis of evolutionary change in ontogeny. *Biological Reviews*, 73(1), 79-123.

Page 22, line 19: “A well-studied example of absence of the CCF occurs in *Acrochordus*, the sister group to the Colubroidea (Fig. 7).”

This is the first reference to figure 7, where a summary of the distribution of the CCF and type of macrostomy is presented. However, the authors do not tell anywhere where they have obtained their data. Their material and methods section does not list representatives of all these families and subfamilies of snakes, and to my knowledge there is no published study containing an exhaustive sample of all these taxa and covering their type of jaw suspension. It seems that the authors here are trying to overgeneralize statements presented in the literature, extending

observations that were made on some specific booids and colubroids to the whole snake clade, with specifics that go down to each family and subfamily where no data currently exist. The authors referenced a study (their ref. 6) that provided descriptions of the types of macrostomy present in some booids and some colubroids. However, fig. 7 in this manuscript seems to extrapolate the data presented in that work to ALL booids and ALL colubroids, specifying even the condition for every subfamily, but without providing any sources for this additional information. A few examples:

- 1) How do the authors know that all Dipsadinae show type B macrostomy? Did they personally observe representatives of all species from the subfamily Dipsadinae? If so, why there is no mention of that in their material and methods?
 - 2) The authors themselves seem to recognize the possibility that type C macrostomy may be more widespread than currently known, making the evolution of the macrostomatan condition more complex than previously anticipated (page 25, lines 6-12), and yet they provide a misleading figure where this is not the case, and type C macrostomy is limited to only two exceptions in a deceptively very informative image (see also the comment on the caption for Fig. 6).
- In conclusion, unless the authors really possess data on the type of suspensorium for all snakes, I would recommend them to remove such level of detail from the figure, which is highly misleading about what we actually know about the distribution of types of macrostomy within specific snake lineages. The authors should simplify this figure and make it very clear that it only represents a summary of their personal hypothesis of the distribution of these traits within snakes. They should be more straightforward about what represents actual data and what is only assumed to happen in the various lineages.

Page 22, line 33 and following: “However, this basal placement of *Acrochordus* based on CCF morphology was later rejected, due largely to the strong phylogenetic evidence placing *Acrochordus* at a far more derived position as the sister to *Colubroidea* [33]; again, this rejection parallels our rejection of scenario one in *Thamnophis*. This later analysis re-characterized the absence of the CCF as a paedomorphic feature, making *Acrochordus* unique among snakes in retaining this embryonic state – similar to our scenario two – though lacked ontogenetic data supporting this conclusion [33].”

To be honest, Rieppel 1980 (ref. 33 herein) did not discuss the CCF of *Acrochordus*, but the undivided fissura metotica of this taxon (there is no mention of the CCF at all in that paper). In fact, the basal placement of *Acrochordus* by McDowell (1979) and others was mostly based on its undivided fissura metotica (a well documented embryonic feature within squamata), and only partially on its lack of a CCF (which according to some is still present in this snake but in rudimentary form, see authors’ ref. 27).

Page 23, line 26: “From our data, the persistent absence of the CCF throughout all three ontogenetic stages unequivocally demonstrates this condition in *Thamnophis* to be a case of paedomorphosis.”

As I pointed out above (comment on page 21, line 10), this is an incorrect statement based on a misinterpretation of what paedomorphosis is and how it is detected. Absence of a feature in all developmental stages of the same given species does not provide evidence of paedomorphosis. The authors need to identify the plesiomorphic embryonic/juvenile snake condition via a comparison with other embryonic/juvenile snakes.

Page 23, line 43: replace “the adult condition matches the embryonic condition” with “the adult condition matches the plesiomorphic snake embryonic condition”.

Page 25, line 3: insert “main” after “two” (exceptions have already been noted in the literature, and have also been mentioned in this manuscript).

Page 25, line 8: insert a reference after *Homalopsis* (this taxon was not the object of this study).

Page 25, line 12: “Our observations of suspensorium development in *T. radix* also provide novel evidence for paedomorphosis in the evolution and ontogeny of the scolecephidian skull.”

The condition observed in *Thamnophis* does not provide any evidence for paedomorphosis in scolecephidians, unless it is shown to represent the general (plesiomorphic) embryonic snake condition. What it does is open the possibility to new interpretations on how the scolecephidian skull may have come to be if the contact between quadrate and prootic indeed turns out to be a plesiomorphic embryonic snake trait (i.e. pending further data).

Page 25, line 27 and following: “Furthermore, the absence of a crista circumfenestralis in *T. radix* [...] provides evidence for paedomorphosis in the evolution of this trait.”

As discussed above, the absence of a crista circumfenestralis in embryonic and adult *Thamnophis* does not demonstrate that paedomorphosis is the cause of this absence in the latter. *Thamnophis* alone does not tell us anything about which evolutionary developmental process is responsible for the absence of the CCF in adult snakes. The authors first need to verify that absence of a CCF is a plesiomorphic condition in embryonic or juvenile snakes, if that is the case, then they can conclude that its absence in adult *Thamnophis* can be interpreted as paedomorphic.

Page 33, line 6: as discussed above, in booids the quadrate does not always remain upright (see comment on page 17, line 47)

Reference 25 is likely Kamal (1969) (and not 19696).

Table 1:

Frontal & Adult: Completely sutured with frontal (not “completely fused along frontal suture”).

Parietal & Embryo: “Almost entirely unfused along sagittal suture.” There is no sagittal suture to speak of, only an unossified central region of the parietal.

Parietal & Juvenile: parietal roof ossified (not “sutured”).

Dentary & Embryo/Juvenile/Adult: To my knowledge, tooth counts on marginal dentigerous elements (dentary and maxilla) are very conservative in snakes within a given species and throughout ontogeny. Please recheck tooth positions, and make sure you don't simply count visible teeth, but tooth positions. Tooth counts from multiple specimens (and from both sides of the skull) would be welcome to make sure that there is no intraspecific variability added to what may vary ontogenetically.

Figure 6: I wonder if showing the angle between quadrate and compound in a tilted perspective of the skull is appropriate (the angles appear smaller than they actually are). I think that showing these angles in lateral view would be much more effective. The figure caption contains a sentence that I find misleading: “this pathway of development is unique to *T. radix* and one other, unrelated genus within the Caenophidia, indicating independent evolution of this pattern”. The sentence assumes that only *Thamnophis* and another caenophidian have a type C macrostomy (see fig. 7), while in fact those are simply the only two snakes where presence of this type of jaw suspension has been noted so far in the literature. There has been no systematic review of jaw suspension in all snakes, therefore making such generalizations is unjustified. The authors should rephrase by writing that “this pathway of development has so far been observed only in ...” (do not write that it is unique to two species, we simply do not know that).

Figure 7: see comment above (Page 22, line 19).

Review form: Reviewer 2 (Alexandra Deufel)

Is the manuscript scientifically sound in its present form?

Yes

Are the interpretations and conclusions justified by the results?

Yes

Is the language acceptable?

Yes

Is it clear how to access all supporting data?

Yes

Do you have any ethical concerns with this paper?

No

Have you any concerns about statistical analyses in this paper?

No

Recommendation?

Accept with minor revision (please list in comments)

Comments to the Author(s)

This is a well-written paper and was a pleasure to read! I rarely review manuscripts that do not require many grammatical and stylistic changes, so, thank you.

Here are my comments:

Materials and Methods section: Briefly explain why you only scanned 8 specimens of this widely available species (cost, local availability?). Were the two juveniles from the same mother or were they unrelated? It is not fatal to your conclusions if the juveniles are related, but it would be good to mention.

You scanned the heads of alcohol preserved specimens, not skulls, so please explain briefly that you digitally removed soft tissues and rendered the skulls for analysis.

Results section 3.1 (page 6, line 15): I think that the premaxillary notch is seen better in the ventral view and you should refer to Fig. 4, not Fig. 3.

Results section 3.5. (page 10, lines 29ff): It would be good to add an illustration of the crista circumfenestralis in a different colubrid to allow direct comparison with your specimens. If that's impossible, insert a citation where the reader could find such an illustration.

Discussion section 4.1 (bottom of page 16): Briefly mention diets of juveniles vs adult *T. radix*. Since you are talking about ontogenetic shifts in diet and gape, prey type is important.

Discussion section 4.3 (page 22): Explain why you didn't examine other *Thamnophis* to see presence or absence of CCF. This is one of the big points of your paper, you should tell the reader why you didn't check other species. *Thamnophis* are readily available in collections.

Table: Define LARST in table heading or inside table. In the row about the frontal, column 3 (adult), it should read "Complete contact with parietal", not with frontal.

These are all the suggestions I have. I'm looking forward to seeing this in print.
Cheers,
Alex Deufel

Decision letter (RSOS-182228.R0)

04-Apr-2019

Dear Ms Strong,

The editors assigned to your paper ("Cranial ontogeny of *Thamnophis radix* (Serpentes: Colubroidea) with a re-evaluation of current paradigms of snake skull evolution") have now received comments from reviewers. We would like you to revise your paper in accordance with the referee and Associate Editor suggestions which can be found below (not including confidential reports to the Editor). Please note this decision does not guarantee eventual acceptance.

Please submit a copy of your revised paper before 27-Apr-2019. Please note that the revision deadline will expire at 00.00am on this date. If we do not hear from you within this time then it will be assumed that the paper has been withdrawn. In exceptional circumstances, extensions may be possible if agreed with the Editorial Office in advance. We do not allow multiple rounds of revision so we urge you to make every effort to fully address all of the comments at this stage. If deemed necessary by the Editors, your manuscript will be sent back to one or more of the original reviewers for assessment. If the original reviewers are not available, we may invite new reviewers.

- Data accessibility

It is a condition of publication that all supporting data are made available either as supplementary information or preferably in a suitable permanent repository. The data

accessibility section should state where the article's supporting data can be accessed. This section should also include details, where possible of where to access other relevant research materials such as statistical tools, protocols, software etc can be accessed. If the data have been deposited in an external repository this section should list the database, accession number and link to the DOI for all data from the article that have been made publicly available. Data sets that have been deposited in an external repository and have a DOI should also be appropriately cited in the manuscript and included in the reference list.

If you wish to submit your supporting data or code to Dryad (<http://datadryad.org/>), or modify your current submission to dryad, please use the following link:
<http://datadryad.org/submit?journalID=RSOS&manu=RSOS-182228>

- **Competing interests**

- **Authors' contributions**

- **Acknowledgements**

- **Funding statement**

Kind regards,

Andrew Dunn

on behalf of Prof Kevin Padian (Subject Editor)

Associate Editor's comments:

The reviewers of your paper are broadly positively inclined towards it; however, there a number of matters that need to be addressed before the journal can consider it further -- the extensive feedback from the reviewers should help here. Please ensure that you fully address the concerns of the reviewers, and incorporate their suggestions (or provide a fully reasoned scientific rebuttal to them) before submitting your revision for consideration. Be aware that, in general, we are unable to consider multiple rounds of major revision. Best of luck and thanks for submitting to the journal.

Comments to Author:

Reviewers' Comments to Author:

Reviewer: 1

Comments to the Author(s)

Strong et al. present a nice overview of the ontogenetic development of the skull in a modern colubrid snake, *Thamnophis sirtalis*. I found it intriguing that despite the fact that this is a very well known snake species so much can still be found when closely observing its bones and their development. Things like the contact between quadrate and prootic in the snake embryo and the lack of a crista circumfenestralis in the adult are certainly novel reports and are worth publishing. Having said this, I also found several issues with this manuscript, which preclude publication in its current state.

Here is a summary of the major points:

- 1) Some of the terminology used is inadequate, like use of terms such as "orbitosphenoidal plate" or "incisive process".
- 2) There is confusion between fusion of elements and simple sutures.
- 3) The manuscript would be greatly improved if the lower jaws were rendered separately in lateral and medial view. This way the authors could properly illustrate all the elements they describe, including angular and splenial, and would not have the lower jaws obscuring most of the anatomy of the skull in ventral view (e.g. palatines, parabasisphenoid, and basioccipital cannot be adequately seen because of this). Moreover, the authors took the time to segment all the individual skull bones from the three developmental stages, but then simply figure the whole skulls without showing the internal details of any of the elements. If the authors only wanted to provide colorized versions of the skulls, then Adobe Photoshop could have provided the very same results in a fraction of the time. I suggest that the authors take advantage of the fact that they have digitally segmented individual elements in 3D and provide rendered images to illustrate internal anatomical details that at present are discussed but not figured anywhere (at least as supplementary figures).
- 4) I noticed an issue with tooth counts. The authors appear to have counted only those teeth that are actually present and ankylosed to the dentigerous elements, when instead they should have provided counts of tooth positions. This resulted in very inconsistent tooth counts (i.e. very low numbers in the juvenile compared to the adult), when in fact the numbers of tooth positions are expected to be very close, if not identical (at least for dentaries and maxillae). Moreover, tooth counts could be affected by intraspecific variability (to a small extent), but the authors provide counts for only 3 of their eight specimens. This way it is not possible to tell whether different tooth

counts could be caused by simple random intraspecific variation rather than by ontogenetic change.

5) There is confusion revolving around the concept of paedomorphosis and on how it is detected. The authors assume that if a trait is retained from an embryonic or juvenile stage into the adult stage of a given species then the adult of that species is paedomorphic. This is incorrect. See details below (e.g. Page 21, line 10). This misconception affects their interpretation of the lack of a CCF in *Thamnophis* as a plesiomorphic feature, and also their interpretation of the scolecophidian skull as the result of paedomorphosis. Because the authors cannot demonstrate that the lack of a CCF or the contact between quadrate and prootic are plesiomorphic embryonic/juvenile traits in snakes, then they cannot reach the conclusion that *Thamnophis* and scolecophidians are paedomorphic for these traits. The authors can only raise the possibility (i.e. formulate a hypothesis that requires further data to be supported).

6) Figure 7 is quite a problematic figure. It deceptively suggests that the authors have way more information on the distribution of the various modes of jaw suspension than they actually have. I have extensively discussed the issues relating to this figure in one of my comments below.

Below I provide detailed comments on all major and minor issues as they present themselves in the manuscript:

Page 1, line 53: than in most other (insert "most").

Page 2, line 17: replace "many" with "some" (there aren't really that many studies).

Page 6, line 6: "increased integration of the premaxilla with the rest of the snout". I would argue that this is not true. The images show that the premaxilla is very loosely connected to the rest of the snout in all three specimens/stages, and the only contact is between premaxilla and anterior tip of the septomaxilla.

Page 6, line 6: the authors mistakenly use the term "incisive process" to refer to the premaxillary process of the vomer (the "incisive process" is a process on the ventral side of the premaxilla of some squamates other than snakes). Please replace "incisive process" with "premaxillary process" here and in any following instance (lines 17, 26).

Page 6, line 13: "posterior premaxillary notch which will eventually accommodate the septomaxilla and vomer".

The notch does not accommodate any part of those bones.

Page 6, line 22: "premaxilla's nasal process expands both dorsoventrally and anteroposteriorly". I believe this is incorrect, it expands mediolaterally and anteroposteriorly. It is quite clear that the relative height of the nasal process remains the same if compared to septomaxilla, nasal, and prefrontal.

Page 6, lines 24-31: "Extension of the anterior processes of the septomaxilla and incisive processes of the vomer relative to the rest of these bones creates stronger contact – and thus improved integration – between the premaxilla and the rest of the snout (Fig. 3C)."

This is incorrect, see previous comment on line 6.

Page 6, lines 35-36: "the dorsal laminae are only present as a cluster of unfused ossifications".

This is inaccurate, there is no cluster of unfused ossifications, which seems to imply the presence of multiple centers of ossification, but only a weakly ossified surface that when rendered digitally appears full of holes, most likely as a rendering artifact.

Page 6, lines 38-39: “dorsal laminae distinctly present and fused to the vertical laminae”.

This is inaccurate. The dorsal laminae are not individual elements separate from the vertical laminae, they are part of the same element (the nasal), which is in the process of being mineralised. It follows that there can be no fusion between parts of the same element.

Page 7, line 13: sutured (not “fused”). The frontals never fuse in snakes.

Page 7, line 22: The frontals are not completely sutured to each other (don’t write “fuse”).

Page 7, line 24: please remove “at which point a deep sagittal sulcus is present.” (that is not a “sulcus”, is the suture between the bones).

Page 7, line 33: I would suggest using the term “optic fenestra” (not “foramen”, foramina are small round holes, fenestrae are large gaps, not necessarily round).

Page 7, line 36: Please replace the term “orbitosphenoidal plate” with “descending flange of the frontal”. The only literature reference I could find where the term “orbitosphenoid plate” was used in the description of a snake skull dates back to 1831 (and also refers to something different). Please replace “orbitosphenoidal plate” with “descending flange of the frontal” also in the following instances (same page lines 40, 52; and page 8, lines 3, 8).

Page 7, line 43: parabasisphenoid (not “basisphenoid”; the bone referred to results from fusion of parasphenoid and basisphenoid). Please replace “basisphenoid” with “parabasisphenoid” also in the following instances.

Page 8, line 8: emargination (not “notch”).

Page 8, line 19: Add “(Fig. 1)” after “juvenile stages”.

Page 9, line 6: replace “expanded” with “bent” or “recurved”.

Page 9, line 22: “The palatine tooth row bears 15 teeth in the embryo, 14 teeth in the juvenile, and 16 teeth in the adult”.

The palatines are hidden by the lower jaws, so it is impossible to verify the tooth counts from the figures provided. In fact, the lower jaws hide a great deal of the ventral anatomy of the skulls; it would be better to have them rendered separately next to the skulls, this way the authors could also illustrate the medial view of the lower jaws, which at present is not shown.

Based on what argued below for the tooth counts on dentaries and maxillae (Page 15, lines 31 & 52), I would suggest the authors to double-check these counts as well, in order to make sure that they actually provide counts of tooth positions and not just of preserved teeth.

Finally, the numbers presented here, if correct, do not appear to follow an ontogenetic trend (15-14-16). I suspect that there may be some intraspecific variation in there as well. In order to see if this is the case, the authors should provide tooth counts for all specimens that they CT scanned (3 embryos, 2 juveniles, and 3 adults). With only one specimen per ontogenetic stage it is hard to discriminate between what is actually ontogenetic change and what is simply random intraspecific variation. The same holds for the tooth counts in dentaries and maxillae. Moreover, the authors should also provide the counts for left and right counterparts of each bone, in case the numbers are different.

Page 9, lines 31-35: “The braincase undergoes a general dorsoventral flattening throughout

ontogeny, from an initially globular appearance in the embryo to a dorsoventrally compressed and anteroposteriorly elongated form in the adult (Fig. 3A-C)."

The compression of the braincase in the adult is mostly anterior, the posterior half is as deep in the adult as it is in the juvenile (the length/max-depth ratio of the skull is about 2.1 in both). Please be more precise in your description.

Page 9, line 42: insert "posterior to" before "lateral aperture" (the jugular foramen is not posteroventral to the lateral aperture, it is posterior to it).

Page 10, line 19: "The jugular foramen also develops internal struts that subdivides it into three openings internally".

"Subdivide" (not "subdivides", the word "struts" is plural), and you could mention which nerves go through those foramina, they are well known in the snake literature.

Page 10, line 36: "stapedial footplate and LARST exposed in lateral view (Fig. 5A-C)."

No stapedial footplate is shown in figure 5, you could add a reference to figure 3 to illustrate your point.

Page 10, lines 42-47: "The nuchal crest that runs anteromedial-posterolateral above the fenestra ovalis is minimally developed in the embryo, resulting in a globular process above each fenestra ovalis".

Please add a figure reference and label the nuchal crest in it for those people unfamiliar with the term.

Page 10, line 49: I would replace "this process" with "it".

Page 11, line 6: Delete "Similarly" (there is no similitude here).

Page 11, line 36: "a narrowing of the large ventral notch of the supraoccipital". Between the otic capsules? If so, please specify, it is unclear which notch the authors are referring to.

Page 11, lines 40-42: "Internally, the supraoccipital undergoes a general elaboration of internal crests and subdivisions." This sounds extremely vague and superficial. What kind of elaboration? Which crests? Which subdivisions? Please be precise in your description or remove this sentence.

Page 11, line 52: "contact with the prootic is stronger than in the juvenile but still not complete."

Do the authors mean: still not complete internally? Externally it looks complete in the figures.

Page 12, line 3: insert "dorsally" after "concave".

Page 12, line 12 and following: What is the point of segmenting all the individual bones if then internal anatomical details like these are not properly figured and labeled? I would recommend adding some additional figures, if not to the main text at least as supplementary information.

Page 12, line 22: insert "complex" after "dorsalis" (the cid is a muscle complex, not a single muscle). Moreover, a better reference for the meaning of cid would be:

Rieppel, O. (1979). The evolution of the basicranium in the Henophidia (Reptilia: Serpentes). *Zoological Journal of the Linnean Society*, 66(4), 411-431.

The Rieppel (1979) paper the authors are referencing is on scolecophidians, and while it mentions the cid, it does not explain its meaning.

Page 12, line 35: “parabasisphenoid” instead of “basisphenoid”.

Page 12, line 35 and following: “The junction between the parasphenoid process and the body of the basisphenoid also becomes elaborated, with minor lateral projections in the embryo and juvenile becoming enlarged into prominent triangular projections in the adult. These projections contribute to an increase in sutural contact with the anteroventral parietal.”

All these details are not visible because hidden by the lower jaws (see also comment above on Page 9, line 22).

Page 12, line 52: “parabasisphenoid” instead of “basisphenoid”.

Page 13, line 10: delete “dorsal and” (there are no muscle crests on the dorsal surface of these bones).

Page 13, line 19: “a slight dorsal longitudinal crest on the juvenile occipital condyle is extended into a prominent crest running from the centre to the posterior margin of the dorsal basioccipital.” I don’t quite understand what the authors are describing here, and there is no figure to illustrate this feature. A prominent dorsal longitudinal crest running along the dorsal surface of a snake basioccipital would be a very unusual, if not unique, feature. If this feature is real it should be figured.

Page 13, line 33: “dorsal cephalic condyle”, because there is no “ventral cephalic condyle” the word “dorsal” is redundant, please remove.

Page 13, line 52: same as above, there is no “dorsal mandibular condyle”, so the word “ventral” is redundant and can be removed.

Page 13, line 49: “The quadrate also undergoes a progressive deepening of its ventral mandibular condyle”. What do the authors mean by “progressive deepening”? it’s not clear, a dorsoventral expansion is not apparent from the figures, perhaps a mediolateral expansion?

Page 15, line 6 and following: Here the authors again describe features that are not illustrated. This could be easily remedied by rendering the lower jaws separately in lateral and medial views.

Page 15, line 31: “The dentary bears 18 teeth in the embryo, 18 teeth in the juvenile, and 28 teeth in the adult.”

These tooth counts cannot be correct. I can count at least 25 tooth positions in the juvenile illustrated in fig. 3. The authors should count tooth positions, not just fully erupted teeth that are ankylosed to the margin of the dentary. As far as I know, tooth counts on the marginal tooth-bearing elements are very conservative in snake species and across ontogeny, I would be really surprised if *Thamnophis* would show such differences in tooth counts.

Page 15, line 52: “The maxilla bears 16 teeth in the embryo and 18 teeth in the juvenile and adult.” Same problem as above. I can count at least 23 tooth positions in the embryo and a similar number in the juvenile. Please check these numbers again.

Page 16, lines 8-15. Angular and splenial are not figured nor labeled anywhere. Again, the manuscript would be much improved if the lower jaws were rendered separately, so that the ventral aspect of the skull was not obscured, and all elements of the lower jaws could be properly illustrated.

Page 16, lines 19-26: aren't these statolith masses present in the adults as well? I would be surprised if this was not the case.

Page 17, line 8: "pterygoid and dentary tooth rows, which increase from 18 to 28 and from 22 to 26 teeth, respectively".

Based on the likely erroneous tooth counts provided by the authors for dentary and maxilla, I would suggest to check these numbers again.

Page 17, line 47: "the quadrate, though also elongated, remains upright."

This is inaccurate. In some booids, such as *Calabaria*, the quadrate is not elongated and remains upright; however, when the quadrate is elongated in the adult, then it does not remain upright; in fact, a longer quadrate becomes tilted sideways, so that the condyles are approximately in the same vertical transverse plane, but are not aligned vertically one above the other (i.e. the condyles appear to be one above the other only in lateral view). Obviously, a longer quadrate has to be displaced either laterally or posteriorly, or it will create a gap between upper and lower jaws.

Page 18, line 6: I am not sure why the authors think that is the "typical" caenophidian mode. There has been no systematic survey of jaw attachment modes in snakes as far as I am aware, and the references listed here only examined a limited number of species.

Page 19, line 3: "the suspension of the lower jaw shifts from a prootic-supratemporal articulation of the quadrate to a supratemporal-only articulation (Fig. 3 and 6)."

If accurate, this is a very interesting observation. A contact between quadrate and otic capsule is thought to be absent in snake embryos (see for example Kamal and Hammouda (1965:291), who studied the colubroid *Psammophis*: "It is of importance to notice that the quadrate during ontogeny is quite apart from the neurocranium and bears no connection with the auditory capsule. This is correlated with the absence of the otic process and the subsequent loss of the crista parotica commonly present in Lacertilia. Only in the fully formed chondrocranium, after the quadrate has reached its definitive position, its dorso-medial margin is attached to the hind end of the auditory capsule by the supratemporal (tabular) bone."

Kamal, A. M., & Hammouda, H. G. (1965). The development of the skull of *Psammophis sibilans*. II. The fully formed chondrocranium. *Journal of Morphology*, 116(2), 247-295.

Because of this, some readers may question the authors' observation. The fact that the quadrate overlaps the prootic in lateral view doesn't mean that they are in contact, let alone in articulation. Furthermore, in Fig. 6 the tilted perspective seems to suggest that the quadrate is only in articulation with the supratemporal in all three developmental stages, and has no contact with the prootic (but could be an artifact of perspective). And finally, if an articulation between quadrate and prootic was present, then one may expect an articulatory facet on the prootic in fig. 5, but such facet is not apparent. I would suggest the authors to provide as supplementary data 3D surface files (in ply or stl format) of all three embryos that have been CT-scanned, in order to unequivocally show that an articulation between quadrate and prootic is indeed consistently present.

Page 19, lines 6-17: Here the authors speculate about the fact that the quadrate in earlier stages of embryonic *Thamnophis* is likely in articulation with the braincase. As I mentioned above, this goes against what observed by Kamal and Hammouda (1956), and should therefore be discussed in more detail, pointing out the contradiction with that study.

Page 19, line 29: remove "though there are exceptions" (redundant, you already said "in almost all").

Page 19, line 33: “The data present in the *T. radix* embryos and juveniles reveal that the suspensorium of scolecocephalians can be interpreted as heterochronic.”

I recommend inserting the sentence: “, if representative of the plesiomorphic embryonic snake condition,” just after “juveniles”. Paedomorphosis is inferred after observing plesiomorphic juvenile features in adults, thus adult scolecocephalians can be considered paedomorphs only if the plesiomorphic embryonic condition for snakes matches what described in *Thamnophis* rather than what described in other snakes (such as *Psammophis*, the snake described by Kamal and Hammouda [1965]).

Page 19, line 47: replace “juvenile” with “early embryonic”. Such anterior orientation in snakes is only visible in developmental stages occurring before those investigated here by the authors; see Kamal & Hammouda (1965) and Boughner et al. (2007).

Boughner, J. C., Buchtová, M., Fu, K., Diewert, V., Hallgrímsson, B., & Richman, J. M. (2007). Embryonic development of *Python sebae*-I: Staging criteria and macroscopic skeletal morphogenesis of the head and limbs. *Zoology*, 110(3), 212-230.

Page 21, line 10: remove “thus” (“thus” implies that the following sentence is the only possible consequence of what is stated before, but in fact the authors later present another scenario where *T. radix* is still within colubroidea but is not necessarily the only colubroid without a CCF).

Page 21, line 10 and following: “Given that the CCF is absent throughout the embryonic and juvenile forms (Fig. 5), the continued absence of this feature in the adult snake is the result of paedomorphosis, i.e., retention of an embryonic or juvenile feature in an adult individual.”

This is a misinterpretation of the concept of paedomorphosis. Paedomorphosis is not inferred by the retention of a feature from juvenile to adult, but by the retention in an adult of what is generally assumed to be a juvenile ancestral feature (i.e. the plesiomorphic juvenile condition). In order to say that the absence of a CCF is a juvenile feature, the authors should first show that (at least closely related) juvenile snakes typically lack it (not just *Thamnophis*). This would identify the plesiomorphic condition (ideally with support from a phylogeny). Then, and only then, they could argue that the absence of a CCF in adult *Thamnophis* is a paedomorphic feature. Otherwise anything could be a paedomorphic feature, even the presence of two eyes.

To quote from McNamara (1986:5):

“If a descendant passes through fewer stages of ontogenetic development than its ancestor, the descendant adult form will have morphological characteristics which occurred in juveniles of the ancestor. This phenomenon has been termed paedomorphosis.”

Or using Klingenberg’s (1998:83) words:

“The morphological outcomes of changes in rates and timing of development are paedomorphosis or peramorphosis; they are identified by comparisons of ancestors and descendants in relation to the ancestral ontogeny. A descendant is paedomorphic if its later ontogenetic stages retain characteristics from earlier stages of an ancestor”.

In summary, paedomorphosis requires a phylogenetic context, and the ontogenetic trajectory of an individual species is not sufficient to come to any conclusion regarding heterochrony.

McNamara, K. J. (1986). A guide to the nomenclature of heterochrony. *Journal of Paleontology*, 60(1), 4-13.

Klingenberg, C. P. (1998). Heterochrony and allometry: the analysis of evolutionary change in ontogeny. *Biological Reviews*, 73(1), 79-123.

Page 22, line 19: "A well-studied example of absence of the CCF occurs in *Acrochordus*, the sister group to the *Colubroidea* (Fig. 7)."

This is the first reference to figure 7, where a summary of the distribution of the CCF and type of macrostomy is presented. However, the authors do not tell anywhere where they have obtained their data. Their material and methods section does not list representatives of all these families and subfamilies of snakes, and to my knowledge there is no published study containing an exhaustive sample of all these taxa and covering their type of jaw suspension. It seems that the authors here are trying to overgeneralize statements presented in the literature, extending observations that were made on some specific booids and colubroids to the whole snake clade, with specifics that go down to each family and subfamily where no data currently exist. The authors referenced a study (their ref. 6) that provided descriptions of the types of macrostomy present in some booids and some colubroids. However, fig. 7 in this manuscript seems to extrapolate the data presented in that work to ALL booids and ALL colubroids, specifying even the condition for every subfamily, but without providing any sources for this additional information. A few examples:

- 1) How do the authors know that all Dipsadinae show type B macrostomy? Did they personally observe representatives of all species from the subfamily Dipsadinae? If so, why there is no mention of that in their material and methods?
 - 2) The authors themselves seem to recognize the possibility that type C macrostomy may be more widespread than currently known, making the evolution of the macrostomatan condition more complex than previously anticipated (page 25, lines 6-12), and yet they provide a misleading figure where this is not the case, and type C macrostomy is limited to only two exceptions in a deceptively very informative image (see also the comment on the caption for Fig. 6).
- In conclusion, unless the authors really possess data on the type of suspensorium for all snakes, I would recommend them to remove such level of detail from the figure, which is highly misleading about what we actually know about the distribution of types of macrostomy within specific snake lineages. The authors should simplify this figure and make it very clear that it only represents a summary of their personal hypothesis of the distribution of these traits within snakes. They should be more straightforward about what represents actual data and what is only assumed to happen in the various lineages.

Page 22, line 33 and following: "However, this basal placement of *Acrochordus* based on CCF morphology was later rejected, due largely to the strong phylogenetic evidence placing *Acrochordus* at a far more derived position as the sister to *Colubroidea* [33]; again, this rejection parallels our rejection of scenario one in *Thamnophis*. This later analysis re-characterized the absence of the CCF as a paedomorphic feature, making *Acrochordus* unique among snakes in retaining this embryonic state - similar to our scenario two - though lacked ontogenetic data supporting this conclusion [33]."

To be honest, Rieppel 1980 (ref. 33 herein) did not discuss the CCF of *Acrochordus*, but the undivided fissura metotica of this taxon (there is no mention of the CCF at all in that paper). In fact, the basal placement of *Acrochordus* by McDowell (1979) and others was mostly based on its undivided fissura metotica (a well documented embryonic feature within squamata), and only partially on its lack of a CCF (which according to some is still present in this snake but in rudimentary form, see authors' ref. 27).

Page 23, line 26: "From our data, the persistent absence of the CCF throughout all three ontogenetic stages unequivocally demonstrates this condition in *Thamnophis* to be a case of paedomorphosis."

As I pointed out above (comment on page 21, line 10), this is an incorrect statement based on a misinterpretation of what paedomorphosis is and how it is detected. Absence of a feature in all developmental stages of the same given species does not provide evidence of paedomorphosis. The authors need to identify the plesiomorphic embryonic/juvenile snake condition via a comparison with other embryonic/juvenile snakes.

Page 23, line 43: replace “the adult condition matches the embryonic condition” with “the adult condition matches the plesiomorphic snake embryonic condition”.

Page 25, line 3: insert “main” after “two” (exceptions have already been noted in the literature, and have also been mentioned in this manuscript).

Page 25, line 8: insert a reference after *Homalopsis* (this taxon was not the object of this study).

Page 25, line 12: “Our observations of suspensorium development in *T. radix* also provide novel evidence for paedomorphosis in the evolution and ontogeny of the scolecophidian skull.” The condition observed in *Thamnophis* does not provide any evidence for paedomorphosis in scolecophidians, unless it is shown to represent the general (plesiomorphic) embryonic snake condition. What it does is open the possibility to new interpretations on how the scolecophidian skull may have come to be if the contact between quadrate and prootic indeed turns out to be a plesiomorphic embryonic snake trait (i.e. pending further data).

Page 25, line 27 and following: “Furthermore, the absence of a crista circumfenestralis in *T. radix* [...] provides evidence for paedomorphosis in the evolution of this trait.”

As discussed above, the absence of a crista circumfenestralis in embryonic and adult *Thamnophis* does not demonstrate that paedomorphosis is the cause of this absence in the latter. *Thamnophis* alone does not tell us anything about which evolutionary developmental process is responsible for the absence of the CCF in adult snakes. The authors first need to verify that absence of a CCF is a plesiomorphic condition in embryonic or juvenile snakes, if that is the case, then they can conclude that its absence in adult *Thamnophis* can be interpreted as paedomorphic.

Page 33, line 6: as discussed above, in booids the quadrate does not always remain upright (see comment on page 17, line 47)

Reference 25 is likely Kamal (1969) (and not 19696).

Table 1:

Frontal & Adult: Completely sutured with frontal (not “completely fused along frontal suture”).

Parietal & Embryo: “Almost entirely unfused along sagittal suture.” There is no sagittal suture to speak of, only an unossified central region of the parietal.

Parietal & Juvenile: parietal roof ossified (not “sutured”).

Dentary & Embryo/Juvenile/ Adult: To my knowledge, tooth counts on marginal dentigerous elements (dentary and maxilla) are very conservative in snakes within a given species and throughout ontogeny. Please recheck tooth positions, and make sure you don't simply count visible teeth, but tooth positions. Tooth counts from multiple specimens (and from both sides of the skull) would be welcome to make sure that there is no intraspecific variability added to what may vary ontogenetically.

Figure 6: I wonder if showing the angle between quadrate and compound in a tilted perspective of the skull is appropriate (the angles appear smaller than they actually are). I think that showing these angles in lateral view would be much more effective. The figure caption contains a sentence that I find misleading: “this pathway of development is unique to *T. radix* and one other, unrelated genus within the Caenophidia, indicating independent evolution of this pattern”. The

sentence assumes that only *Thamnophis* and another caenophidian have a type C macrostomy (see fig. 7), while in fact those are simply the only two snakes where presence of this type of jaw suspension has been noted so far in the literature. There has been no systematic review of jaw suspension in all snakes, therefore making such generalizations is unjustified. The authors should rephrase by writing that “this pathway of development has so far been observed only in ...” (do not write that it is unique to two species, we simply do not know that).

Figure 7: see comment above (Page 22, line 19).

Reviewer: 2

Comments to the Author(s)

This is a well-written paper and was a pleasure to read! I rarely review manuscripts that do not require many grammatical and stylistic changes, so, thank you.

Here are my comments:

Materials and Methods section: Briefly explain why you only scanned 8 specimens of this widely available species (cost, local availability?). Were the two juveniles from the same mother or were they unrelated? It is not fatal to your conclusions if the juveniles are related, but it would be good to mention.

You scanned the heads of alcohol preserved specimens, not skulls, so please explain briefly that you digitally removed soft tissues and rendered the skulls for analysis.

Results section 3.1 (page 6, line 15): I think that the premaxillary notch is seen better in the ventral view and you should refer to Fig. 4, not Fig. 3.

Results section 3.5. (page 10, lines 29ff): It would be good to add an illustration of the crista circumfenestralis in a different colubrid to allow direct comparison with your specimens. If that's impossible, insert a citation where the reader could find such an illustration.

Discussion section 4.1 (bottom of page 16): Briefly mention diets of juveniles vs adult *T. radix*. Since you are talking about ontogenetic shifts in diet and gape, prey type is important.

Discussion section 4.3 (page 22): Explain why you didn't examine other *Thamnophis* to see presence or absence of CCF. This is one of the big points of your paper, you should tell the reader why you didn't check other species. *Thamnophis* are readily available in collections.

Table: Define LARST in table heading or inside table. In the row about the frontal, column 3 (adult), it should read "Complete contact with parietal", not with frontal.

These are all the suggestions I have. I'm looking forward to seeing this in print.

Cheers,

Alex Deufel

Author's Response to Decision Letter for (RSOS-182228.R0)

See Appendix A.

RSOS-182228.R1 (Revision)

Review form: Reviewer 1

Is the manuscript scientifically sound in its present form?

Yes

Are the interpretations and conclusions justified by the results?

Yes

Is the language acceptable?

Yes

Do you have any ethical concerns with this paper?

No

Recommendation?

Accept as is

Comments to the Author(s)

I must confess that I am a bit disappointed by the tone of the authors' replies, and especially that after spending a considerable amount of time writing what I would consider a thorough and helpful review, my comments were considered "unrealistic, demanding, and often pedantic". I can guarantee the authors that it was not my intention "to create the impression that their manuscript was flawed empirically, conceptually, and grammatically." A review is meant to provide comments and suggestions on how to improve a manuscript, and my belief is that it needs to be rigorous when pointing out flaws, no matter how small.

In any case, the manuscript has now been significantly improved, and I have only a few additional comments.

1) My original comment on Page 6, line 13 was: "posterior premaxillary notch which will eventually accommodate the septomaxilla and vomer". The notch does not accommodate any part of those bones.

To which the authors replied: *The Reviewer is in error – the premaxilla of all squamates, *Thamnophis* included, articulates with both the vomer and septomaxilla. The articulation is clearly articulated and the notch we are discussing is visible in our supplementary files (S4-S6), and in numerous illustrations provided in such works as Cundall and Irish (2008:pg 404, fig. 2.25). Perhaps we are referring to different notches, but as mentioned, we illustrate what we mean by this.*

I take it that the posterior notch in question is the notch between the two vomerine processes of the premaxilla, if so, it does not accommodate neither the vomer nor the septomaxilla. While the vomer and septomaxilla do articulate with the premaxilla in most squamates, these bones do not enter that notch (quite obvious in Fig. 4). If the authors are referring to a different notch then please elaborate and be more precise in your description.

2) My original comment on Page 12, line 12 was: What is the point of segmenting all the individual bones if then internal anatomical details like these are not properly figured and labeled? I would recommend adding some additional figures, if not to the main text at least as supplementary information.

To which the authors replied: *We have added a figure showing an internal view of the braincase, with pertinent features labelled (Fig. 6D-F). Had the Reviewer consulted the supplementary data he/she would have noted we supplied the requested internal anatomy as STL files.*

I must admit that I missed the supplementary data in my first round of review, but in my defense there was no mention of such data in the main text (the link to the dryad repository was only presented at the end of the manuscript and could be easily missed). In any case, anatomical features are not labeled in the STL files, therefore the new figures certainly help illustrate what the authors are writing about.

3) Concerning Fig. 7, my original comment was: In conclusion, unless the authors really possess data on the type of suspensorium for all snakes, I would recommend them to remove such level of detail from the figure, which is highly misleading about what we actually know about the distribution of types of macrostomy within specific snake lineages. The authors should simplify this figure and make it very clear that it only represents a summary of their personal hypothesis of the distribution of these traits within snakes. They should be more straightforward about what represents actual data and what is only assumed to happen in the various lineages.

To which the authors replied: *Strongly disagreed. The reviewer cannot be serious on this comment. Again, establishing the nonsense empirical constraint of having ALL data on ALL snakes does not make this requirement so, it only enforces the Reviewer's misguided approach to this component of our manuscript. A case in point, no phylogenetic hypothesis ever published, using genes or morphology, has ever once sampled all taxa and all genes and all morphology – never once; according to the Reviewer's expectations of our study, not a single phylogenetic or comparative study should ever have been published in the history of evolutionary biology. Yet these hypotheses of sister-group relationships and inferred phylogeny exist and are used, clearly by the Reviewer (if in fact the reviewer is not guilty of generating phylogenies from incomplete and imperfect data sampling) for second, third and fourth order hypothesis constructions. It is ridiculous to make such statements and expect accommodation of these extensive, misguided, and condescending criticisms.*

Obviously I would not expect anyone to sample all species of snakes to draw some general conclusions as those presented in this manuscript, mine was a rhetorical point. The issue was that the authors' Fig. 7 did not make it explicit which taxa had been actually sampled and for which ones anatomical conditions had been extrapolated. A naïve reader (or just someone unfamiliar with the snake literature) looking at their figure may have mistakenly concluded that all of those snake families and subfamilies had been sampled, at least partially if not thoroughly. I am pleased to see that the authors have modified their Fig. 7 so that readers can now see which families/subfamilies have actually been sampled, and also made it explicit that the figure just represents their personal hypothesis of the evolution of macrostomy.

4) My original comment on Page 22, line 33 was: To be honest, Rieppel 1980 (ref. 33 herein) did not discuss the CCF of Acrochordus, but the undivided fissura metotica of this taxon (there is no mention of the CCF at all in that paper).

To which the authors replied: *Note that, while Rieppel (1980) focussed on the undivided metotic fissure as paedomorphic, his later paper with Zaher [ref. 18 herein] included the CCF in this hypothesis of paedomorphosis: "[Acrochordus] appears to retain a number of surprisingly plesiomorphic features in its cranial anatomy. [...] The most unusual features in the skull of Acrochordus are the apparent lack of the*

crista circumfenestralis and the undivided metotic fissure (McDowell, 1979), characteristics that have been explained as a probably consequence of paedomorphosis.” (Rieppel and Zaher 2001: pg 252). As such, our discussion of Rieppel (1980) similarly mentions the overall otico-occipital region of Acrochordus as paedomorphic, in line with Rieppel and Zaher’s (2001) discussion of this hypothesis.

Then the authors should have referenced Rieppel and Zaher (2001) and not Rieppel (1980). However, the authors should know that Rieppel and Zaher (2001) also misquoted Rieppel (1980). In fact, the complete quote from Rieppel and Zaher (2001:252) would be “In light of these phylogenetic relationships, the most unusual features in the skull of Acrochordus are the apparent lack of the crista circumfenestralis and the undivided metotic fissure (McDowell, 1979), characteristics that have been explained as a probable consequence of paedomorphosis (Rieppel, 1980).” The citation of Rieppel (1980) at the end of the quote was conveniently left out by the authors in their reply. I suspect that this is a case where a misquote is reiterated by someone (the authors) who simply trusted another paper (in this case Rieppel and Zaher, 2001) without actually verifying the accuracy of that quote by reading the original source (in this case Rieppel, 1980). Thus, a situation is created where two articles (Rieppel and Zaher, 2001 and this manuscript) make reference to the work of Rieppel (1980) as if it said something about the CCF of Acrochordus as being a paedomorphic feature, while in truth that work didn’t mention the CCF at all. I understand that it is common practice to trust what is written in the primary literature and take the references therein as accurate (especially when one of the authors is on both the referenced and the citing paper!). However, the authors should have checked the original paper by Rieppel to make sure that their statements are indeed accurate and avoid perpetuating a case of poor scholarship.

Despite all this, I am happy with the way the authors have modified the problematic paragraph so that a reference to Rieppel (1980) implying that he wrote about the CCF of Acrochordus is now avoided:

We have modified this paragraph as follows: “The absence of the CCF in Acrochordus was therefore considered consistent with a basal placement of Acrochordus among snakes, similar to the first scenario proposed herein for Thamnophis [36]. However, this basal placement of Acrochordus was later rejected, due largely to the strong phylogenetic evidence placing Acrochordus at a far more deeply nested position as the sister to Colubroidea [37]; again, this rejection parallels our rejection of scenario one in Thamnophis (i.e., lack of a CCF does not inherently indicate basal phylogenetic status). This later analysis re-characterized the otico-occipital region of the skull of Acrochordus – focussing on the persistent lack of division of the metotic fissure – as paedomorphic, making Acrochordus unique among snakes in retaining this embryonic morphology – similar to our scenario two – though lacked ontogenetic data supporting this conclusion [18,37].”

In summary, I would recommend the authors to be more explicit about which “posterior premaxillary notch” (point 1 above) they are referring to, because at the moment the only posterior premaxillary notch I can think of is that between the vomerine processes of the premaxilla. This notch does not accommodate the septomaxilla or the vomer (quite evident in their Fig 4). If the notch they are describing is another notch, then they need to be clearer in their description. Apart from this minor issue I am happy with the manuscript as it is.

Decision letter (RSOS-182228.R1)

04-Jul-2019

Dear Ms Strong:

On behalf of the Editors, I am pleased to inform you that your Manuscript RSOS-182228.R1 entitled "Cranial ontogeny of *Thamnophis radix* (Serpentes: Colubroidea) with a re-evaluation of current paradigms of snake skull evolution" has been accepted for publication in Royal Society Open Science subject to minor revision in accordance with the referee suggestions. Please find the referees' comments at the end of this email.

The reviewers and Subject Editor have recommended publication, but also suggest some minor revisions to your manuscript. Therefore, I invite you to respond to the comments and revise your manuscript.

- Ethics statement

- Data accessibility

If you wish to submit your supporting data or code to Dryad (<http://datadryad.org/>), or modify your current submission to dryad, please use the following link:
<http://datadryad.org/submit?journalID=RSOS&manu=RSOS-182228.R1>

- Competing interests

- Authors' contributions

- Acknowledgements

- Funding statement

Because the schedule for publication is very tight, it is a condition of publication that you submit the revised version of your manuscript before 13-Jul-2019. Please note that the revision deadline will expire at 00.00am on this date. If you do not think you will be able to meet this date please let me know immediately.

Supplementary files will be published alongside the paper on the journal website and posted on

the online figshare repository (<https://figshare.com>). The heading and legend provided for each supplementary file during the submission process will be used to create the figshare page, so please ensure these are accurate and informative so that your files can be found in searches. Files on figshare will be made available approximately one week before the accompanying article so that the supplementary material can be attributed a unique DOI.

on behalf of Kevin Padian (Subject Editor)
openscience@royalsociety.org

Associate Editor Comments to Author:

A few remaining comments have been made by the reviewer (who is broadly happy with the paper now), which we would like you to take into consideration and modify the manuscript accordingly. We'll look forward to receiving the revision shortly.

Reviewer comments to Author:
Reviewer: 1

Comments to the Author(s)

I must confess that I am a bit disappointed by the tone of the authors' replies, and especially that after spending a considerable amount of time writing what I would consider a thorough and helpful review, my comments were considered "unrealistic, demanding, and often pedantic". I can guarantee the authors that it was not my intention "to create the impression that their manuscript was flawed empirically, conceptually, and grammatically." A review is meant to provide comments and suggestions on how to improve a manuscript, and my belief is that it needs to be rigorous when pointing out flaws, no matter how small.

In any case, the manuscript has now been significantly improved, and I have only a few additional comments.

1) My original comment on Page 6, line 13 was: "posterior premaxillary notch which will eventually accommodate the septomaxilla and vomer". The notch does not accommodate any part of those bones.

To which the authors replied: *The Reviewer is in error – the premaxilla of all squamates, *Thamnophis* included, articulates with both the vomer and septomaxilla. The articulation is clearly articulated and the notch we are discussing is visible in our supplementary files (S4-S6), and in numerous illustrations provided in such works as Cundall and Irish (2008:pg 404, fig. 2.25). Perhaps we are referring to different notches, but as mentioned, we illustrate what we mean by this.*

I take it that the posterior notch in question is the notch between the two vomerine processes of

the premaxilla, if so, it does not accommodate neither the vomer nor the septomaxilla. While the vomer and septomaxilla do articulate with the premaxilla in most squamates, these bones do not enter that notch (quite obvious in Fig. 4). If the authors are referring to a different notch then please elaborate and be more precise in your description.

2) My original comment on Page 12, line 12 was: What is the point of segmenting all the individual bones if then internal anatomical details like these are not properly figured and labeled? I would recommend adding some additional figures, if not to the main text at least as supplementary information.

To which the authors replied: *We have added a figure showing an internal view of the braincase, with pertinent features labelled (Fig. 6D-F). Had the Reviewer consulted the supplementary data he/she would have noted we supplied the requested internal anatomy as STL files.*

I must admit that I missed the supplementary data in my first round of review, but in my defense there was no mention of such data in the main text (the link to the dryad repository was only presented at the end of the manuscript and could be easily missed). In any case, anatomical features are not labeled in the STL files, therefore the new figures certainly help illustrate what the authors are writing about.

3) Concerning Fig. 7, my original comment was: In conclusion, unless the authors really possess data on the type of suspensorium for all snakes, I would recommend them to remove such level of detail from the figure, which is highly misleading about what we actually know about the distribution of types of macrostomy within specific snake lineages. The authors should simplify this figure and make it very clear that it only represents a summary of their personal hypothesis of the distribution of these traits within snakes. They should be more straightforward about what represents actual data and what is only assumed to happen in the various lineages.

To which the authors replied: *Strongly disagreed. The reviewer cannot be serious on this comment. Again, establishing the nonsense empirical constraint of having ALL data on ALL snakes does not make this requirement so, it only enforces the Reviewer's misguided approach to this component of our manuscript. A case in point, no phylogenetic hypothesis ever published, using genes or morphology, has ever once sampled all taxa and all genes and all morphology – never once; according to the Reviewer's expectations of our study, not a single phylogenetic or comparative study should ever have been published in the history of evolutionary biology. Yet these hypotheses of sister-group relationships and inferred phylogeny exist and are used, clearly by the Reviewer (if in fact the reviewer is not guilty of generating phylogenies from incomplete and imperfect data sampling) for second, third and fourth order hypothesis constructions. It is ridiculous to make such statements and expect accommodation of these extensive, misguided, and condescending criticisms.*

Obviously I would not expect anyone to sample all species of snakes to draw some general conclusions as those presented in this manuscript, mine was a rhetorical point. The issue was that the authors' Fig. 7 did not make it explicit which taxa had been actually sampled and for which ones anatomical conditions had been extrapolated. A naïve reader (or just someone unfamiliar with the snake literature) looking at their figure may have mistakenly concluded that all of those snake families and subfamilies had been sampled, at least partially if not thoroughly. I am pleased to see that the authors have modified their Fig. 7 so that readers can now see which families/subfamilies have actually been sampled, and also made it explicit that the figure just represents their personal hypothesis of the evolution of macrostomy.

4) My original comment on Page 22, line 33 was: To be honest, Rieppel 1980 (ref. 33 herein) did not discuss the CCF of Acrochordus, but the undivided fissura metotica of this taxon (there is no mention of the CCF at all in that paper).

To which the authors replied: *Note that, while Rieppel (1980) focussed on the undivided metotic fissure as paedomorphic, his later paper with Zaher [ref. 18 herein] included the CCF in this hypothesis of paedomorphosis: “[Acrochordus] appears to retain a number of surprisingly plesiomorphic features in its cranial anatomy. [...] The most unusual features in the skull of Acrochordus are the apparent lack of the crista circumfenestralis and the undivided metotic fissure (McDowell, 1979), characteristics that have been explained as a probably consequence of paedomorphosis.” (Rieppel and Zaher 2001: pg 252). As such, our discussion of Rieppel (1980) similarly mentions the overall otico-occipital region of Acrochordus as paedomorphic, in line with Rieppel and Zaher’s (2001) discussion of this hypothesis.*

Then the authors should have referenced Rieppel and Zaher (2001) and not Rieppel (1980). However, the authors should know that Rieppel and Zaher (2001) also misquoted Rieppel (1980). In fact, the complete quote from Rieppel and Zaher (2001:252) would be “In light of these phylogenetic relationships, the most unusual features in the skull of Acrochordus are the apparent lack of the crista circumfenestralis and the undivided metotic fissure (McDowell, 1979), characteristics that have been explained as a probable consequence of paedomorphosis (Rieppel, 1980).” The citation of Rieppel (1980) at the end of the quote was conveniently left out by the authors in their reply. I suspect that this is a case where a misquote is reiterated by someone (the authors) who simply trusted another paper (in this case Rieppel and Zaher, 2001) without actually verifying the accuracy of that quote by reading the original source (in this case Rieppel, 1980). Thus, a situation is created where two articles (Rieppel and Zaher, 2001 and this manuscript) make reference to the work of Rieppel (1980) as if it said something about the CCF of Acrochordus as being a paedomorphic feature, while in truth that work didn’t mention the CCF at all. I understand that it is common practice to trust what is written in the primary literature and take the references therein as accurate (especially when one of the authors is on both the referenced and the citing paper!). However, the authors should have checked the original paper by Rieppel to make sure that their statements are indeed accurate and avoid perpetuating a case of poor scholarship.

Despite all this, I am happy with the way the authors have modified the problematic paragraph so that a reference to Rieppel (1980) implying that he wrote about the CCF of Acrochordus is now avoided:

*We have modified this paragraph as follows: “The absence of the CCF in Acrochordus was therefore considered consistent with a basal placement of Acrochordus among snakes, similar to the first scenario proposed herein for *Thamnophis* [36]. However, this basal placement of Acrochordus was later rejected, due largely to the strong phylogenetic evidence placing Acrochordus at a far more deeply nested position as the sister to Colubroidea [37]; again, this rejection parallels our rejection of scenario one in *Thamnophis* (i.e., lack of a CCF does not inherently indicate basal phylogenetic status). This later analysis re-characterized the otico-occipital region of the skull of Acrochordus – focussing on the persistent lack of division of the metotic fissure – as paedomorphic, making Acrochordus unique among snakes in retaining this embryonic morphology – similar to our scenario two – though lacked ontogenetic data supporting this conclusion [18,37].”*

In summary, I would recommend the authors to be more explicit about which “posterior premaxillary notch” (point 1 above) they are referring to, because at the moment the only posterior premaxillary notch I can think of is that between the vomerine processes of the premaxilla. This notch does not accommodate the septomaxilla or the vomer (quite evident in their Fig 4). If the notch they are describing is another notch, then they need to be clearer in their description. Apart from this minor issue I am happy with the manuscript as it is.

Author's Response to Decision Letter for (RSOS-182228.R1)

See Appendix B.

Decision letter (RSOS-182228.R2)

17-Jul-2019

Dear Ms Strong,

I am pleased to inform you that your manuscript entitled "Cranial ontogeny of *Thamnophis radix* (Serpentes: Colubroidea) with a re-evaluation of current paradigms of snake skull evolution" is now accepted for publication in Royal Society Open Science.

on behalf of Kevin Padian (Subject Editor)
openscience@royalsociety.org

Appendix A

To the Editor:

Prior to responding to Reviewer 1, we feel it is important to respectfully convey our concerns regarding aspects of this review. Reviewer 1 did provide many useful comments and criticisms of our manuscript, which we greatly appreciate and which improved the document; however, we also feel that in many places the review is unrealistic and demanding, and often pedantic to the point where it seems the Reviewer wanted merely to create the impression that our manuscript was flawed empirically, conceptually, and grammatically. As is often the case with such reviews, Reviewer 1 is also mistaken on many points of grammar, anatomy, and interpretation. We have accommodated this Reviewer's comments to the best of our ability, though we have also rebutted numerous points where necessary with fully reasoned arguments.

As evidence of this broad problem in Review 1, we point to the nature and tone of the second review to illustrate our point.

Nevertheless, as mentioned previously and in the spirit of the peer review process, we have made corrections and thus improvements to the manuscript, rebutting only those comments/criticisms which we considered incorrect.

Reviewers' Comments to Author:

Reviewer: 1

Comments to the Author(s)

Strong et al. present a nice overview of the ontogenetic development of the skull in a modern colubrid snake, *Thamnophis sirtalis*. I found it intriguing that despite the fact that this is a very well known snake species so much can still be found when closely observing its bones and their development. Things like the contact between quadrate and prootic in the snake embryo and the lack of a crista circumfenestralis in the adult are certainly novel reports and are worth publishing. Having said this, I also found several issues with this manuscript, which preclude publication in its current state.

We must point out immediately that we are not presenting information on *Thamnophis sirtalis*, but rather *Thamnophis radix* (the Plains Garter Snake). We compare our materials to a skeletonized specimen of *T. sirtalis* but this manuscript details new data from *T. radix*.

Here is a summary of the major points:

1) Some of the terminology used is inadequate, like use of terms such as "orbitosphenoidal plate" or "incisive process".

Agreed. This terminology has been changed, in accordance with the reviewer's suggestions – see specific comments below.

2) There is confusion between fusion of elements and simple sutures.

Agreed. This has also been revised – see specific comments below.

3) The manuscript would be greatly improved if the lower jaws were rendered separately in lateral and medial view. This way the authors could properly illustrate all the elements they describe, including angular and splenial, and would not have the lower jaws obscuring most of the anatomy of the skull in ventral view (e.g. palatines, parabasisphenoid, and basioccipital cannot be adequately seen because of this).

Agreed. We've added a figure (new Fig. 5) showing the skull without the lower jaws in ventral view, with a separate depiction of the lower jaws in dorsal and medial view (lateral and ventral views of the jaws are visible in Fig. 3 and 4, respectively). We decided to add an extra figure instead of simply modifying Fig. 4 so that a ventral view of the complete articulated skull is still included.

Moreover, the authors took the time to segment all the individual skull bones from the three developmental stages, but then simply figure the whole skulls without showing the internal details of any of the elements. If the authors only wanted to provide colorized versions of the skulls, then Adobe Photoshop could have provided the very same results in a fraction of the time. I suggest that the authors take advantage of the fact that they have digitally segmented individual elements in 3D and provide rendered images to illustrate internal anatomical details that at present are discussed but not figured anywhere (at least as supplementary figures).

Disagreed. Surface mesh (.stl) files of each bone from the featured skulls were included in the supplementary data available to the reviewers (<https://datadryad.org/review?doi=doi:10.5061/dryad.n50st0n>). We also included videos of the skulls that the reader could easily access to obtain this anatomical data. It is clear to us that Reviewer #1 did not refer to our supplementary data files as submitted. However, to ensure that readers are aware of these files, we have added references to the supplementary data in the figure captions. We have also added a figure showing an internal view of the braincase (Fig. 6D-F) in order to illustrate pertinent internal anatomical features.

4) I noticed an issue with tooth counts. The authors appear to have counted only those teeth that are actually present and ankylosed to the dentigerous elements, when instead they should have provided counts of tooth positions. This resulted in very inconsistent tooth counts (i.e. very low numbers in the juvenile compared to the adult), when in fact the numbers of tooth positions are expected to be very close, if not identical (at least for dentaries and maxillae). Moreover, tooth counts could be affected by intraspecific variability (to a small extent), but the authors provide counts for only 3 of their eight specimens. This way it is not possible to tell whether different tooth counts could be caused by simple random intraspecific variation rather than by ontogenetic change.

Agreed. While we do not consider this to be a major issue with the manuscript as written, we have modified the tooth counts to reflect our best estimate regarding number of tooth positions, along with a count of fully ankylosed teeth.

5) There is confusion revolving around the concept of paedomorphosis and on how it is detected. The authors assume that if a trait is retained from an embryonic or juvenile stage into the adult stage of a given species then the adult of that species is paedomorphic. This is incorrect. See details below (e.g. Page 21, line 10). This misconception affects their interpretation of the lack of a CCF in *Thamnophis* as a plesiomorphic feature, and also their interpretation of the scolecophidian skull as the result of paedomorphosis. Because the authors cannot demonstrate that the lack of a CCF or the contact between quadrate and prootic are plesiomorphic embryonic/juvenile traits in snakes, then they cannot

reach the conclusion that *Thamnophis* and scolecophidians are paedomorphic for these traits. The authors can only raise the possibility (i.e. formulate a hypothesis that requires further data to be supported).

We have addressed these points in detail below and modified the manuscript text accordingly – see relevant comments (especially page 19 line 33 and page 21 line 10). It is true that an inference of paedomorphosis in an evolutionary context depends on a comparison with a sister taxon or more. However, we believe nobody challenges the notion that the presence of a fully developed CCF is widespread among colubroids, and that ontogenetically, to get to a fully developed CCF condition an individual must pass through a stage where the CCF is still not fully developed. Even in the absence of an explicit phylogenetic mapping of this particular character among species closely related to *Thamnophis*, it is safe to assume that the condition in *Thamnophis* would necessarily be a paedomorphic one compared to almost any other colubroid. We remind the Reviewer we are making a comparison at very broad phylogenetic levels, comparing *Thamnophis* to the generalized condition among colubroids and other families. This is not a low taxonomic level comparison or analysis (i.e., genus level or species level). However, to try to meet this criticism we have provided the above explanation explicitly in the text and acknowledge that further studies within the genus *Thamnophis* are necessary to indicate if this pattern is specific to *T. radix* (therefore a paedomorphic event within the genus), or if it is widespread among other species of the genus (implying that the entire genus is paedomorphic relative to others).

6) Figure 7 is quite a problematic figure. It deceptively suggests that the authors have way more information on the distribution of the various modes of jaw suspension than they actually have. I have extensively discussed the issues relating to this figure in one of my comments below.

Partially disagreed. We have addressed the reviewer’s concerns regarding this figure and have modified Fig. 7 and its figure caption accordingly – see our response to the comment for page 22 line 19.

Below I provide detailed comments on all major and minor issues as they present themselves in the manuscript:

Page 1, line 53: than in most other (insert “most”).

Done. We have also made another revision to the abstract, modifying page 1 line 44 to specify that these are the first complete micro-CT segmentations of any non-adult snake skull (at the time of initial submission, these were the first complete segmentations of any snake skull, but since then a paper on scolecophidian skull anatomy included a completely segmented adult skull).

Chretien, J, Wang-Claypool CY, Glaw F, and Scherz MD. (2019). The bizarre skull of *Xenotyphlops* sheds light on synapomorphies of Typhlopoidea. *Journal of Anatomy* (234):637-655. doi:10.1111/joa.12952

Page 2, line 17: replace “many” with “some” (there aren’t really that many studies).

Disagreed. Making this change would alter the meaning of this sentence. We have modified the wording to: “Of these few studies on snake postnatal cranial ontogeny, many have focused on changes in gape size and diet...”. As the reviewer indicates, there haven’t been many studies on this

topic; however, what we're conveying in this sentence is that, of the few studies that have been done, they mostly focus on gape size and diet. Our addition of the word "few" clarifies this point.

Page 6, line 6: "increased integration of the premaxilla with the rest of the snout". I would argue that this is not true. The images show that the premaxilla is very loosely connected to the rest of the snout in all three specimens/stages, and the only contact is between premaxilla and anterior tip of the septomaxilla.

We have no response to this point other to disagree with the Reviewer. We invite the reviewer to access the online supplementary videos and STL files to better observe the differences between the two ontogenetic stages.

Page 6, line 6: the authors mistakenly use the term "incisive process" to refer to the premaxillary process of the vomer (the "incisive process" is a process on the ventral side of the premaxilla of some squamates other than snakes). Please replace "incisive process" with "premaxillary process" here and in any following instance (lines 17, 26).

Done.

Page 6, line 13: "posterior premaxillary notch which will eventually accommodate the septomaxilla and vomer". The notch does not accommodate any part of those bones.

The Reviewer is in error – the premaxilla of all squamates, *Thamnophis* included, articulates with both the vomer and septomaxilla. The articulation is clearly articulated and the notch we are discussing is visible in our supplementary files (S4-S6), and in numerous illustrations provided in such works as Cundall and Irish (2008:pg 404, fig. 2.25). Perhaps we are referring to different notches, but as mentioned, we illustrate what we mean by this.

Page 6, line 22: "premaxilla's nasal process expands both dorsoventrally and anteroposteriorly". I believe this is incorrect, it expands mediolaterally and anteroposteriorly. It is quite clear that the relative height of the nasal process remains the same if compared to septomaxilla, nasal, and prefrontal.

Disagreed. We made an absolute statement about the nasal process, through the ontogenetic stages we observed, becoming longer and deeper. The Reviewer is attempting to diminish this simple observation by taking our absolute statement of the nasal process in exclusion of all other elements and making it a relative statement by comparing it to other snout elements, something we did not do. This is a trivial distinction and represents the Reviewer's approach to describing anatomy and ontogeny and not our own. We therefore have made no changes to our straightforward observation.

Page 6, lines 24-31: "Extension of the anterior processes of the septomaxilla and incisive processes of the vomer relative to the rest of these bones creates stronger contact – and thus improved integration – between the premaxilla and the rest of the snout (Fig. 3C)."
This is incorrect, see previous comment on line 6.

As per line 6, we have changed our use of "incisive process".

Page 6, lines 35-36: "the dorsal laminae are only present as a cluster of unfused ossifications".

This is inaccurate, there is no cluster of unfused ossifications, which seems to imply the presence of multiple centers of ossification, but only a weakly ossified surface that when rendered digitally appears full of holes, most likely as a rendering artifact.

We have changed the wording to: “the dorsal laminae are present only as a very weakly ossified surface”. Table 1 has been updated accordingly.

Page 6, lines 38-39: “dorsal laminae distinctly present and fused to the vertical laminae”. This is inaccurate. The dorsal laminae are not individual elements separate from the vertical laminae, they are part of the same element (the nasal), which is in the process of being mineralised. It follows that there can be no fusion between parts of the same element.

We have changed the wording to: “The juvenile nasal (Fig. 2B) shows improved ossification, with the dorsal laminae in particular showing increased mineralization, thus approaching the adult form.”

Page 7, line 13: sutured (not “fused”). The frontals never fuse in snakes.

Done.

Page 7, line 22: The frontals are not completely sutured to each other (don’t write “fuse”).

We have changed this sentence to: “The frontals are not completely sutured to each other until the adult stage.”

Page 7, line 24: please remove “at which point a deep sagittal sulcus is present.” (that is not a “sulcus”, is the suture between the bones).

Done.

Page 7, line 33: I would suggest using the term “optic fenestra” (not “foramen”, foramina are small round holes, fenestrae are large gaps, not necessarily round).

Again, we feel this criticism is made in the absence of substance. We point to Cundall and Irish (2008:351) who use “OF” throughout their classic work in all illustrations of snake skulls and describe that abbreviation as “optic foramen or fenestra”. The terminology for the frontal was based on Bullock and Tanner (1966) and Cundall and Irish (2008), among many, who refer to this opening as the “optic foramen”.

Bullock RE, Tanner WW. 1966. A comparative osteological study of two species of Colubridae (*Pituophis* and *Thamnophis*). Brigham Young University Science Bulletin – Biological Series, 8(3), 1-29.

Cundall, D., and Irish, F. 2008. The snake skull. In Gans, C, Gaunt, A.S, and Adler, K. (eds.) Biology of the Reptilia, the Skull of Lepidosauria, Vol. 20, Morphology H. Society for the Study of Amphibians and Reptiles, University Heights, Ohio, pp. 340-692.

Page 7, line 36: Please replace the term “orbitosphenoidal plate” with “descending flange of the frontal”. The only literature reference I could find where the term “orbitosphenoid plate” was used in the description of a snake skull dates back to 1831 (and also refers to something different). Please

replace “orbitosphenoidal plate” with “descending flange of the frontal” also in the following instances (same page lines 40, 52; and page 8, lines 3, 8).

As above, this terminology was based on Bullock and Tanner (1966). However, we also recognize that there is no defined orbitosphenoid in snakes, though the ventral process underlying the optic foramen is often referred to as the orbitosphenoid process of the frontal. Recognizing inconsistencies and consistencies in terminology, we have modified our term in this case as per the reviewer’s suggestion. The label of this structure on Fig. 3 has also been changed accordingly.

Page 7, line 43: parabasisphenoid (not “basisphenoid”; the bone referred to results from fusion of parasphenoid and basisphenoid). Please replace “basisphenoid” with “parabasisphenoid” also in the following instances.

Done. Labels on Figures 3-6 have also been updated accordingly.

Page 8, line 8: emargination (not “notch”).

Trivial, but done.

Page 8, line 19: Add “(Fig. 1)” after “juvenile stages”.

Done.

Page 9, line 6: replace “expanded” with “bent” or “recurved”.

Again, trivial, but done.

Page 9, line 22: “The palatine tooth row bears 15 teeth in the embryo, 14 teeth in the juvenile, and 16 teeth in the adult”.

The palatines are hidden by the lower jaws, so it is impossible to verify the tooth counts from the figures provided. In fact, the lower jaws hide a great deal of the ventral anatomy of the skulls; it would be better to have them rendered separately next to the skulls, this way the authors could also illustrate the medial view of the lower jaws, which at present is not shown.

Based on what argued below for the tooth counts on dentaries and maxillae (Page 15, lines 31 & 52), I would suggest the authors to double-check these counts as well, in order to make sure that they actually provide counts of tooth positions and not just of preserved teeth.

As per our above comment, the Reviewer did not refer to our surface mesh (.stl) files of each bone as made available in the supplementary data. However, we have added a figure (new Fig. 5) rendering the lower jaws separately from the rest of the skull and displaying medial and dorsal views of the lower jaws, to assist our readers assuming they might also not refer to the supplementary data. We have also modified all tooth counts to indicate both ankylosed teeth and likely number of tooth positions.

Finally, the numbers presented here, if correct, do not appear to follow an ontogenetic trend (15-14-16). I suspect that there may be some intraspecific variation in there as well. In order to see if this is the case, the authors should provide tooth counts for all specimens that they CT scanned (3 embryos, 2 juveniles,

and 3 adults). With only one specimen per ontogenetic stage it is hard to discriminate between what is actually ontogenetic change and what is simply random intraspecific variation. The same holds for the tooth counts in dentaries and maxillae. Moreover, the authors should also provide the counts for left and right counterparts of each bone, in case the numbers are different.

Ultimately, we did not speculate on the meaning of these tooth counts nor are they critical to any conclusions drawn in this paper; this observation was a simple one noting numbers of teeth, nothing more. However, to acknowledge the reviewer's suggestion of intraspecific variation, we have removed references to tooth counts from our discussion and table. Beyond this modification, a detailed analysis of the dentition of *Thamnophis* is outside the scope of this project; we also feel that the extremely detailed counts that the reviewer is requesting would not meaningfully increase the impact of our results. We therefore have done nothing further.

Page 9, lines 31-35: "The braincase undergoes a general dorsoventral flattening throughout ontogeny, from an initially globular appearance in the embryo to a dorsoventrally compressed and anteroposteriorly elongated form in the adult (Fig. 3AC)."

The compression of the braincase in the adult is mostly anterior, the posterior half is as deep in the adult as it is in the juvenile (the length/max-depth ratio of the skull is about 2.1 in both). Please be more precise in your description.

This has been changed to: "This dorsoventral compression is most pronounced in the anterior half of the braincase, as the posterior half maintains a relatively constant depth throughout ontogeny."

Page 9, line 42: insert "posterior to" before "lateral aperture" (the jugular foramen is not posteroventral to the lateral aperture, it is posterior to it).

Done.

Page 10, line 19: "The jugular foramen also develops internal struts that subdivides it into three openings internally".

"Subdivide" (not "subdivides", the word "struts" is plural), and you could mention which nerves go through those foramina, they are well known in the snake literature.

For clarity, we have changed the wording to "...develops internal struts subdividing it into..."

Page 10, line 36: "stapedial footplate and LARST exposed in lateral view (Fig. 5A-C)."

No stapedial footplate is shown in figure 5, you could add a reference to figure 3 to illustrate your point.

Done. We have also added a reference to Fig. 3 on pg 20, lines 43-45 in the original manuscript ("...leaving the stapedial footplate, fenestra ovalis, and LARST completely exposed in lateral view (Fig. 3A-C, 5A-C)") and on pg 21, lines 10-12 in the original manuscript (in the revised manuscript: "We therefore suggest that, in the phylogenetic context of colubroids, the absence of a CCF in adults of *T. radix* (Fig. 3 and 6)..."), as these sentences have the same issue raised here.

Page 10, lines 42-47: "The nuchal crest that runs anteromedial-posterolateral above the fenestra ovalis is minimally developed in the embryo, resulting in a globular process above each fenestra ovalis".

Please add a figure reference and label the nuchal crest in it for those people unfamiliar with the term.

Done.

Page 10, line 49: I would replace “this process” with “it”.

Done.

Page 11, line 6: Delete “Similarly” (there is no similitude here).

We have retained the use of this word, as we are referring to the similar elaboration of crests that occurs on the prootic’s dorsal and internal surfaces.

Page 11, line 36: “a narrowing of the large ventral notch of the supraoccipital”. Between the otic capsules? If so, please specify, it is unclear which notch the authors are referring to.

We consider this request to be quite redundant: the entire supraoccipital bone forms in the dermatocranium between the otic capsules; the brain is also between the right and left otic capsules, as is the basioccipital, a good portion of the parietal, etc, etc. However, to ensure clarity, we have specified what we mean in the text.

Page 11, lines 40-42: “Internally, the supraoccipital undergoes a general elaboration of internal crests and subdivisions.”

This sounds extremely vague and superficial. What kind of elaboration? Which crests? Which subdivisions? Please be precise in your description or remove this sentence.

We’ve removed this sentence.

Page 11, line 52: “contact with the prootic is stronger than in the juvenile but still not complete.”
Do the authors mean: still not complete internally? Externally it looks complete in the figures.

Yes, this sentence refers to the internal contact. We have changed it to: “The supraoccipital’s internal contact with the otoccipital and prootic is weak in the embryo and juvenile braincases compared to the adult condition. Within the adult otic capsule, internal contact between the supraoccipital and otoccipital is almost complete, while internal contact with the prootic is stronger than in the juvenile but still incomplete”.

Page 12, line 3: insert “dorsally” after “concave”.

Done.

Page 12, line 12 and following: What is the point of segmenting all the individual bones if then internal anatomical details like these are not properly figured and labeled? I would recommend adding some additional figures, if not to the main text at least as supplementary information.

We have added a figure showing an internal view of the braincase, with pertinent features labelled (Fig. 6D-F). Had the Reviewer consulted the supplementary data he/she would have noted we supplied the requested internal anatomy as STL files.

Page 12, line 22: insert “complex” after “dorsalis” (the cid is a muscle complex, not a single muscle). Moreover, a better reference for the meaning of cid would be:

Rieppel, O. (1979). The evolution of the basicranium in the Henophidia (Reptilia: Serpentes). *Zoological Journal of the Linnean Society*, 66(4), 411-431.

The Rieppel (1979) paper the authors are referencing is on scolecophidians, and while it mentions the cid, it does not explain its meaning.

We have added “complex” to the CID and modified the sentence.

Page 12, line 35: “parabasisphenoid” instead of “basisphenoid”.

Done.

Page 12, line 35 and following: “The junction between the parasphenoid process and the body of the basisphenoid also becomes elaborated, with minor lateral projections in the embryo and juvenile becoming enlarged into prominent triangular projections in the adult. These projections contribute to an increase in sutural contact with the anteroventral parietal.”

All these details are not visible because hidden by the lower jaws (see also comment above on Page 9, line 22).

We have added a figure of the ventral skull without the lower jaws (new Fig. 5), so that these features are visible. Again, all views were illustrated in the STL files in the supplementary data.

Page 12, line 52: “parabasisphenoid” instead of “basisphenoid”.

Done.

Page 13, line 10: delete “dorsal and” (there are no muscle crests on the dorsal surface of these bones).

Disagreed. Both the basioccipital and basisphenoid possess a ventral floor or base to the element, but also lateral walls that are dorsal to the floor or base of each element. There are indeed crests for the attachment of musculature that are dorsally placed to those present on the ventral surface of each element. Were this not so, neither element would be visible in lateral view when observing the skull of a snake. However, to clarify, we have changed ‘dorsal’ to ‘lateral’.

Page 13, line 19: “a slight dorsal longitudinal crest on the juvenile occipital condyle is extended into a prominent crest running from the centre to the posterior margin of the dorsal basioccipital.”

I don’t quite understand what the authors are describing here, and there is no figure to illustrate this feature. A prominent dorsal longitudinal crest running along the dorsal surface of a snake basioccipital would be a very unusual, if not unique, feature. If this feature is real it should be figured.

This feature is visible in our modified Fig. 6, showing an internal view of the braincase. While the reviewer may disagree with our use of the adjective “prominent,” our depiction of this feature allows readers to draw their own conclusions.

Page 13, line 33: “dorsal cephalic condyle”, because there is no “ventral cephalic condyle” the word “dorsal” is redundant, please remove.

Done.

Page 13, line 52: same as above, there is no “dorsal mandibular condyle”, so the word “ventral” is redundant and can be removed.

Done.

Page 13, line 49: “The quadrate also undergoes a progressive deepening of its ventral mandibular condyle”.

What do the authors mean by “progressive deepening”? it’s not clear, a dorsoventral expansion is not apparent from the figures, perhaps a mediolateral expansion?

We mean that the notch of the forked mandibular condyle becomes deeper. This change is visible in the supplementary files.

Page 15, line 6 and following: Here the authors again describe features that are not illustrated. This could be easily remedied by rendering the lower jaws separately in lateral and medial views.

Again, had the Reviewer consulted the supplementary data, this would have been clear. However, for future readers we have added this view to the figures of the manuscript (see Fig. 5) and have referred to this new figure in the text.

Page 15, line 31: “The dentary bears 18 teeth in the embryo, 18 teeth in the juvenile, and 28 teeth in the adult.”

These tooth counts cannot be correct. I can count at least 25 tooth positions in the juvenile illustrated in fig. 3. The authors should count tooth positions, not just fully erupted teeth that are ankylosed to the margin of the dentary. As far as I know, tooth counts on the marginal tooth-bearing elements are very conservative in snake species and across ontogeny, I would be really surprised if *Thamnophis* would show such differences in tooth counts.

As noted previously, we have modified the tooth counts to reflect counts of tooth positions and ankylosed teeth.

Page 15, line 52: “The maxilla bears 16 teeth in the embryo and 18 teeth in the juvenile and adult.” Same problem as above. I can count at least 23 tooth positions in the embryo and a similar number in the juvenile. Please check these numbers again.

Same as above.

Page 16, lines 8-15. Angular and splenial are not figured nor labeled anywhere. Again, the manuscript would be much improved if the lower jaws were rendered separately, so that the ventral aspect of the skull was not obscured, and all elements of the lower jaws could be properly illustrated.

Done – see Fig. 5. The angular and splenial are also available as STL files in the supplementary data.

Page 16, lines 19-26: aren't these statolith masses present in the adults as well? I would be surprised if this was not the case.

No, there were no statolith masses in the adult. Had there been, we would have reported them. Polachowski and Werneburg (2013) [ref. 2 herein] also report statolith masses only being present in some specimens.

Page 17, line 8: "pterygoid and dentary tooth rows, which increase from 18 to 28 and from 22 to 26 teeth, respectively".

Based on the likely erroneous tooth counts provided by the authors for dentary and maxilla, I would suggest to check these numbers again.

See above. We have modified this sentence to: "The aforementioned increase in gape and lengthening of the jaw apparatus occurs in conjunction with a general increase in length and recurvature of the teeth throughout ontogeny, thus improving the snake's ability to grasp onto progressively larger and stronger prey."

Page 17, line 47: "the quadrate, though also elongated, remains upright."

This is inaccurate. In some booids, such as Calabaria, the quadrate is not elongated and remains upright; however, when the quadrate is elongated in the adult, then it does not remain upright; in fact, a longer quadrate becomes tilted sideways, so that the condyles are approximately in the same vertical transverse plane, but are not aligned vertically one above the other (i.e. the condyles appear to be one above the other only in lateral view). Obviously, a longer quadrate has to be displaced either laterally or posteriorly, or it will create a gap between upper and lower jaws.

This lack of anteroposterior rotation of the quadrate is what we were referring to (i.e., the condition the reviewer discusses with the quadrate staying in the same vertical transverse plane). To clarify this, we have changed this to: "The quadrate, though also elongated, is laterally displaced at the mandibular condyle but remains in the same transverse plane (i.e., perpendicular to the skull in lateral view)."

Page 18, line 6: I am not sure why the authors think that is the "typical" caenophidian mode. There has been no systematic survey of jaw attachment modes in snakes as far as I am aware, and the references listed here only examined a limited number of species.

While Palci *et al.* (2016) only discussed 2 caenophidian species in detail in their analysis of snake post-natal ontogeny, they refer to personal observations of this ontogenetic pathway in several other caenophidians, including representatives of the Colubridae, Elapidae, Homalopsidae, and Viperidae [ref. 6 herein]. As such, based on the current literature, this pathway is widespread within heretofore sampled caenophidians, hence our referral to it as the typical caenophidian mode. However, to clarify this point, we have modified this sentence to: "the typical caenophidian mode of jaw development as proposed by Palci *et al.* (2016) [6]."

Page 19, line 3: "the suspension of the lower jaw shifts from a prootic-supratemporal articulation of the quadrate to a supratemporal-only articulation (Fig. 3 and 6)."

If accurate, this is a very interesting observation. A contact between quadrate and otic capsule is thought to be absent in snake embryos (see for example Kamal and Hammouda (1965:291), who studied the colubroid *Psammophis*: "It is of importance to notice that the quadrate during ontogeny is quite

apart from the neurocranium and bears no connection with the auditory capsule. This is correlated with the absence of the otic process and the subsequent loss of the crista parotica commonly present in Lacertilia. Only in the fully formed chondrocranium, after the quadrate has reached its definitive position, its dorso-medial margin is attached to the hind end of the auditory capsule by the supratemporal (tabular) bone.”

Kamal, A. M., & Hammouda, H. G. (1965). The development of the skull of *Psammophis sibilans*. II. The fully formed chondrocranium. *Journal of Morphology*, 116(2), 247-295.

Because of this, some readers may question the authors' observation. The fact that the quadrate overlaps the prootic in lateral view doesn't mean that they are in contact, let alone in articulation. Furthermore, in Fig. 6 the tilted perspective seems to suggest that the quadrate is only in articulation with the supratemporal in all three developmental stages, and has no contact with the prootic (but could be an artifact of perspective). And finally, if an articulation between quadrate and prootic was present, then one may expect an articulatory facet on the prootic in fig. 5, but such facet is not apparent. I would suggest the authors to provide as supplementary data 3D surface files (in ply or stl format) of all three embryos that have been CT-scanned, in order to unequivocally show that an articulation between quadrate and prootic is indeed consistently present.

We are familiar with Kamal and Hammouda (1965) and note that their techniques of observation were coarse (enlarged tracings of chondrocranial anatomy in embryos, and thin sectioning following decalcification in late stage embryos) compared to the refined, detailed and non-invasive data we obtained with microCT scans. That the articulation/contact between the quadrate and prootic was not evident to them comes as no surprise as a result. Additionally, we must question the Reviewer's assertion of the need for a facet to be found on the prootic for the quadrate. The articulation between the quadrate and prootic in scolecophidians, such as it is, bears no facet on the prootic. Why would an embryo-neonate develop a facet in the absence of active or any movement at the joint?

Nonetheless, to address this point, we note that we uploaded STL files in the supplementary data which provide a more versatile view of the quadrate's dorsal articulation than is possible in the figures, and which show that the quadrate is indeed in strong contact with the prootic and supratemporal in the embryonic snake. As the Reviewer mentions, in Fig. 6 (now Fig. 7) the apparent lack of articulation is a consequence of perspective; we have therefore added a reference to the supplementary data in the figure caption to point readers toward the STL files we've provided. We feel that our current supplementary files provide ample detail to sufficiently support our observation, and do not see it necessary to upload STL files of all of our specimens.

Page 19, lines 6-17: Here the authors speculate about the fact that the quadrate in earlier stages of embryonic *Thamnophis* is likely in articulation with the braincase. As I mentioned above, this goes against what observed by Kamal and Hammouda (1956), and should therefore be discussed in more detail, pointing out the contradiction with that study.

As suggested, we have expanded our discussion of this observed quadrate-neurocranium contact. See the first paragraph of 4.2 – Suspensorium: “Previous studies have observed a lack of contact between the quadrate and otic capsule in embryonic snakes, with the quadrate only ever articulating dorsally with the supratemporal (e.g., [28]). However, the new data reported here on the suspensorium (see 3.6) indicate that, through the observed ontogenetic stages, the suspension of the lower jaw shifts from a prootic-supratemporal articulation of the quadrate to a supratemporal-only articulation (Fig. 3

and 7). This quadrate-prootic contact – which is most prominent in the embryo – thus revises our current knowledge of this aspect of braincase development and indicates heretofore unrecognized variation in the ontogeny of these structures.”

28. Kamal AM, Hammouda HG. 1965 The development of the skull of *Psammophis sibilans* II. The fully formed chondrocranium. *J. Morphol.* 116, 247-295.

Page 19, line 29: remove “though there are exceptions” (redundant, you already said “in almost all”).

Done.

Page 19, line 33: “The data present in the *T. radix* embryos and juveniles reveal that the suspensorium of scolecophidians can be interpreted as heterochronic.”

I recommend inserting the sentence: “, if representative of the plesiomorphic embryonic snake condition,” just after “juveniles”. Paedomorphosis is inferred after observing plesiomorphic juvenile features in adults, thus adult scolecophidians can be considered paedomorphs only if the plesiomorphic embryonic condition for snakes matches what described in *Thamnophis* rather than what described in other snakes (such as *Psammophis*, the snake described by Kamal and Hammouda [1965]).

We have expanded our discussion to acknowledge this point. See the second paragraph of 4.2 – Suspensorium, where we have added the following explanation:

“If the plesiomorphic snake condition for the development of the quadrate is as observed in *T. radix*, with the quadrate changing from a prootic-supratemporal articulation in embryos and juveniles to a supratemporal-only articulation in adults – rather than as described by Kamal and Hammouda (1965), in which the quadrate does not contact the chondrocranium – this condition in scolecophidians can be interpreted as heterochronic. More specifically, the data present in *T. radix* embryos and juveniles would reveal the scolecophidian skull to be paedomorphic, as adult scolecophidians retain the putative plesiomorphic embryonic condition in which the quadrate articulates with only the chondrocranium.”

Page 19, line 47: replace “juvenile” with “early embryonic”. Such anterior orientation in snakes is only visible in developmental stages occurring before those investigated here by the authors; see Kamal & Hammouda (1965) and Boughner et al. (2007).

Boughner, J. C., Buchtová, M., Fu, K., Diewert, V., Hallgrímsson, B., & Richman, J. M. (2007). Embryonic development of *Python sebae*—I: Staging criteria and macroscopic skeletal morphogenesis of the head and limbs. *Zoology*, 110(3), 212-230.

Done.

Page 21, line 10: remove “thus” (“thus” implies that the following sentence is the only possible consequence of what is stated before, but in fact the authors later present another scenario where *T. radix* is still within colubroidea but is not necessarily the only colubroid without a CCF).

Disagreed and incorrect. “Thus” suggests no finality with respect to the subject of the sentence and its predicate and dependent clauses. It is a “conclusion” word to be sure, but comments on an outcome related to a specific possibility, not all possibilities. A later sentence can raise a counterpoint, i.e., an

alternative hypothesis, and not be in conflict with the use of “thus” in a previous sentence. Knowledge claims are relative, not absolute, and so we use “thus” and retain its use in the sentence as noted.

Page 21, line 10 and following: “Given that the CCF is absent throughout the embryonic and juvenile forms (Fig. 5), the continued absence of this feature in the adult snake is the result of paedomorphosis, i.e., retention of an embryonic or juvenile feature in an adult individual.”

This is a misinterpretation of the concept of paedomorphosis. Paedomorphosis is not inferred by the retention of a feature from juvenile to adult, but by the retention in an adult of what is generally assumed to be a juvenile ancestral feature (i.e. the plesiomorphic juvenile condition). In order to say that the absence of a CCF is a juvenile feature, the authors should first show that (at least closely related) juvenile snakes typically lack it (not just *Thamnophis*). This would identify the plesiomorphic condition (ideally with support from a phylogeny). Then, and only then, they could argue that the absence of a CCF in adult *Thamnophis* is a paedomorphic feature. Otherwise anything could be a paedomorphic feature, even the presence of two eyes.

To quote from McNamara (1986:5):

“If a descendant passes through fewer stages of ontogenetic development than its ancestor, the descendant adult form will have morphological characteristics which occurred in juveniles of the ancestor. This phenomenon has been termed paedomorphosis.”

Or using Klingenberg’s (1998:83) words:

“The morphological outcomes of changes in rates and timing of development are paedomorphosis or peramorphosis; they are identified by comparisons of ancestors and descendants in relation to the ancestral ontogeny. A descendant is paedomorphic if its later ontogenetic stages retain characteristics from earlier stages of an ancestor”.

In summary, paedomorphosis requires a phylogenetic context, and the ontogenetic trajectory of an individual species is not sufficient to come to any conclusion regarding heterochrony.

McNamara, K. J. (1986). A guide to the nomenclature of heterochrony. *Journal of Paleontology*, 60(1), 4-13.

Klingenberg, C. P. (1998). Heterochrony and allometry: the analysis of evolutionary change in ontogeny. *Biological Reviews*, 73(1), 79-123.

We have responded this comment in the general remarks at the beginning of the review. The Reviewer has in several places in this review expended a great deal of effort in suggesting we do not understand paedomorphosis. We have read the Reviewer’s lengthy ‘instructional’ text, replete with quotes from McNamara and Klingenberg, and have noted that our understanding matches that of the suggested mentors. To clarify what we had considered common knowledge, we have modified this sentence so as to accommodate the Reviewer’s discussion:

“We therefore suggest that, in the phylogenetic context of colubroids, the continued absence of a CCF in adults of *T. radix* (Fig. 3 and 6) can be seen as the derived condition, thus representing a

paedomorphic pattern when compared to other colubroids, i.e., retention of a plesiomorphic embryonic or juvenile feature in an adult individual.”

If the Reviewer thought we were comparing within a species to generate statements on paedomorphic patterns in the absence of comparisons, constantly explicit or not, to other colubroids specifically or other snakes and lizards generally, then the Reviewer missed the context of the paper – a context and point not missed by Reviewer #2. We have made several modifications throughout the paper to include a statement on states of juveniles in more basal/ancestral clades.

This modified sentence is part of a larger justification we have added for our interpretation of paedomorphosis in *T. radix* (see the 3rd paragraph of 4.3.):

“Although we did not (nor did any other previous study) assess this character at a species level across all species of colubroids (a seemingly impossible task at the moment), the current consensus in the literature is that presence of the CCF in adults is widespread among colubroids (e.g., [18,20,33]). In order to achieve this adult condition, the organism must necessarily pass through an earlier ontogenetic stage in which the CCF is absent or undeveloped, as is characteristic of any endochondral ossification. As such, it is safe to infer that, for colubroids, the most frequent condition is to lack a CCF in the embryonic (and possibly also juvenile) stage and to possess a CCF in the adult stage. We therefore suggest that, in the phylogenetic context of colubroids, the continued absence of a CCF in adults of *T. radix* (Fig. 3 and 6) can be seen as the derived condition, thus representing a paedomorphic pattern when compared to other colubroids, i.e., retention of a plesiomorphic embryonic or juvenile feature in an adult individual.”

While an explicit and thoroughly-sampled phylogenetic mapping of the CCF throughout ontogeny would be the ideal way to determine the plesiomorphic condition for colubroids, such an undertaking is beyond the scope of this paper. However, even without this explicit phylogenetic mapping, we can reconstruct absence of the CCF as the plesiomorphic juvenile condition for colubroids following the line of logic presented above. This supports our conclusions and addresses the issues raised by the reviewer regarding the need for phylogenetic context in interpretations of paedomorphosis.

18. Rieppel O, Zaher H. 2001 The development of the skull in *Acrochordus granulatus* (Schneider) (Reptilia: Serpentes), with special consideration of the otico-occipital complex. *J. Morphol.* 249, 252–266.

20. Palci A, Caldwell MW. 2014 The Upper Cretaceous snake *Dinilysia patagonica* Smith-Woodward, 1901, and the crista circumfenestralis of snakes. *J. Morphol.* 275, 1187–1200. (doi: 10.1002/jmor.20297)

33. Zaher H, Rieppel O. 1999 The phylogenetic relationships of *Pachyrhachis problematicus*, and the evolution of limblessness in snakes (Lepidosauria, Squamata). *C. R. Acad. Sci. - Ser. IIA - Sci. Terre plan./Earth Plan. Sci.* 329, 831-837.

Page 22, line 19: “A well-studied example of absence of the CCF occurs in *Acrochordus*, the sister group to the Colubroidea (Fig. 7).”

This is the first reference to figure 7, where a summary of the distribution of the CCF and type of macrostomy is presented. However, the authors do not tell anywhere where they have obtained their data. Their material and methods section does not list representatives of all these families and

subfamilies of snakes, and to my knowledge there is no published study containing an exhaustive sample of all these taxa and covering their type of jaw suspension.

This hypothesis of macrostomy is based on the conclusions of Palci *et al.* (2016) [ref. 6 herein], as mentioned in our discussion of macrostomy evolution in the main manuscript text (see 4.1, where we first refer to Fig. 7). Distribution of the CCF is from Palci and Caldwell (2014), as mentioned in our discussion in 4.3 [ref. 20 herein]. References to these studies have been added to the figure caption to clarify these sources of data.

Palci A, Lee MSY, Hutchinson MN. 2016 Patterns of postnatal ontogeny of the skull and lower jaw of snakes as revealed by micro-CT scan data and three-dimensional geometric morphometrics. *J. Anat.* 229, 723-754. (doi: 10.1111/joa.12509)

Palci A, Caldwell MW. 2014 The Upper Cretaceous snake *Dinilysia patagonica* Smith-Woodward, 1901, and the crista circumfenestralis of snakes. *J. Morphol.* 275, 1187–1200. (doi: 10.1002/jmor.20297)

It seems that the authors here are trying to overgeneralize statements presented in the literature, extending observations that were made on some specific booids and colubroids to the whole snake clade, with specifics that go down to each family and subfamily where no data currently exist. The authors referenced a study (their ref. 6) that provided descriptions of the types of macrostomy present in some booids and some colubroids. However, fig. 7 in this manuscript seems to extrapolate the data presented in that work to ALL booids and ALL colubroids, specifying even the condition for every subfamily, but without providing any sources for this additional information.

Strongly disagreed. We suggest the reviewer more closely examine the Discussion section (especially pg 749) of the paper whose data we used to illustrate Figure 7. While Palci *et al.* (2016) [ref. 6 herein] focus their discussion on 5 snake species, they refer to personal observations of SEVERAL other booid and caenophidian species which also display the respective hypothesized booid/caenophidian pathways, thus supporting these developmental trajectories as being widespread within these respective groups (see Palci *et al.* (2016), pg 749). Our depiction of these pathways in Fig. 7 is simply a diagrammatic representation of the evolution of macrostomy exactly as proposed by Palci *et al.* (2016) – including extrapolation of these observations to the overall Booidea and Caenophidia – with the addition of *Thamnophis*. As such, this figure and its associated extrapolation of these ontogenetic trajectories is well-rooted in published studies of snake macrostomy.

Furthermore, our approach of taking the known distribution of character states and extrapolating it to fill in “gaps” in the phylogeny is a common and well-accepted approach to ancestral state reconstruction used in almost every single study using phylogenetic comparative methods. For example, see Palci *et al.* (2014) – Fig. 6: not all species of Elapidae, for instance, were sampled in that study, but, based on the elapids that were sampled, the condition of the CCF could be generalized to that family in their phylogeny. This same principle of extrapolation based on current data is what we have employed in Fig. 7, and is what is widely accepted in the paleontological and phylogenetic literature.

Our inclusion of colubrid subfamilies is not intended as an attempt to overgeneralize this hypothesis; instead, we split Colubridae into its constituent subfamilies to: 1) provide a more specific phylogenetic context/placement for *Thamnophis*, as it is the focus of this paper; and 2) visually highlight the

diversity of Colubridae, so as to emphasize the status of *Thamnophis* as the only colubrid currently recognized as lacking a CCF.

That being said, we agree with the reviewer that this figure ultimately represents a hypothesis, and that further sampling of macrostomatan taxa is required to more definitively support or falsify it (though we note that thus far there is no evidence available in the literature nor provided by the reviewer to falsify this hypothesis, as initially proposed by Palci *et al.* 2016 and further supported by our findings). As such, we have modified Figure 7 (now Figure 8) so that families/subfamilies that were directly sampled (either by Palci *et al.* (2016) or, in the case of *Thamnophis*, by us) are in green text; this provides transparency so that it is obvious which taxa have been directly observed to conform to this hypothesis, vs taxa whose status we are extrapolating. We have also modified the figure caption to read: "Phylogeny of Serpentes, highlighting the distribution of presence of the crista circumfenestralis (CCF) and the hypothesized evolution of macrostomy (modified from the evolutionary pathways proposed by [6])..." in order to emphasize that this is ultimately a hypothesis.

Palci A, Caldwell MW. 2014 The Upper Cretaceous snake *Dinilysia patagonica* Smith-Woodward, 1901, and the crista circumfenestralis of snakes. *J. Morphol.* 275, 1187–1200. (doi: 10.1002/jmor.20297)

6. Palci A, Lee MSY, Hutchinson MN. 2016 Patterns of postnatal ontogeny of the skull and lower jaw of snakes as revealed by micro-CT scan data and three-dimensional geometric morphometrics. *J. Anat.* 229, 723-754. (doi: 10.1111/joa.12509)

A few examples:

1) How do the authors know that all Dipsadinae show type B macrostomy? Did they personally observe representatives of all species from the subfamily Dipsadinae? If so, why there is no mention of that in their material and methods?

This is how phylogenetic bracketing works and as such generates hypotheses for investigation. The rhetorical question concerning the empirical requirement to view all members of Dipsadinae is obviously impossible to be done for a single study.

Phylogenetic bracketing allows the extrapolation of type B macrostomy to Dipsadinae based on the current hypothesis of type B macrostomy as the typical caenophidian condition, as presented by Palci *et al.* (2016). This extrapolation is the typical approach to ancestral state reconstruction (see comment above), i.e., reconstruction of a condition within a bracketed clade. Our coloration of taxon names (see figure caption for Fig. 8) provides transparency regarding which taxa have been observed and which are extrapolated.

2) The authors themselves seem to recognize the possibility that type C macrostomy may be more widespread than currently known, making the evolution of the macrostomatan condition more complex than previously anticipated (page 25, lines 6-12), and yet they provide a misleading figure where this is not the case, and type C macrostomy is limited to only two exceptions in a deceptively very informative image (see also the comment on the caption for Fig. 6).

Strongly disagreed. This pattern is known from only two species, which we indicate in the figure. We fail to see how this is misleading or deceptive. To clarify that this distribution of type C macrostomy is based on current observations of snake taxa, we have modified the figure caption to include the

following: “(C) *Thamnophis* and *Homalopsis* – both caenophidians – achieve macrostomy via both rotation of the quadrate and elongation of the supratemporal and quadrate. This pathway may be more widespread than previously recognized, but has currently only been observed in these two genera.”

Type A and B macrostomy have been observed in several booid and caenophidian species, respectively [see Palci *et al.* (2016:749)], allowing our extrapolation of these pathways to booids and caenophidians in general (= the most frequently observed pattern). In contrast, type C macrostomy has only been observed in *Thamnophis* and *Homalopsis*; therefore, while it may turn out to be more widespread than currently recognized, it would not be accurate to definitively generalize this pathway to other taxa given current observations and current hypotheses of macrostomy evolution. Instead, it is most parsimonious to depict type B macrostomy as the typical caenophidian pathway, with type C macrostomy as the exception, based on current data.

In conclusion, unless the authors really possess data on the type of suspensorium for all snakes, I would recommend them to remove such level of detail from the figure, which is highly misleading about what we actually know about the distribution of types of macrostomy within specific snake lineages. The authors should simplify this figure and make it very clear that it only represents a summary of their personal hypothesis of the distribution of these traits within snakes. They should be more straightforward about what represents actual data and what is only assumed to happen in the various lineages.

Strongly disagreed. The reviewer cannot be serious on this comment. Again, establishing the nonsense empirical constraint of having ALL data on ALL snakes does not make this requirement so, it only enforces the Reviewer’s misguided approach to this component of our manuscript. A case in point, no phylogenetic hypothesis ever published, using genes or morphology, has ever once sampled all taxa and all genes and all morphology – never once; according to the Reviewer’s expectations of our study, not a single phylogenetic or comparative study should ever have been published in the history of evolutionary biology. Yet these hypotheses of sister-group relationships and inferred phylogeny exist and are used, clearly by the Reviewer (if in fact the reviewer is not guilty of generating phylogenies from incomplete and imperfect data sampling) for second, third and fourth order hypothesis constructions. It is ridiculous to make such statements and expect accommodation of these extensive, misguided, and condescending criticisms.

In summary of the points discussed above, we have modified the figure caption to make it explicit that this is a hypothesis of macrostomy evolution, and have modified the figure itself to make it explicit which taxa have been directly observed and which have been extrapolated. We have also included references in the figure caption for our data sources regarding distribution of the CCF and of macrostomy-related ontogenetic pathways. These changes make this figure more transparent, as requested by the reviewer. We have also provided justification above for dividing Colubridae into its constituent subfamilies. Overall, this approach to ancestral state reconstruction – as discussed above – is well-accepted in the literature.

Page 22, line 33 and following: “However, this basal placement of *Acrochordus* based on CCF morphology was later rejected, due largely to the strong phylogenetic evidence placing *Acrochordus* at a far more derived position as the sister to Colubroidea [33]; again, this rejection parallels our rejection of scenario one in *Thamnophis*. This later analysis re-characterized the absence of the CCF as a

paedomorphic feature, making *Acrochordus* unique among snakes in retaining this embryonic state – similar to our scenario two – though lacked ontogenetic data supporting this conclusion [33].”

To be honest, Rieppel 1980 (ref. 33 herein) did not discuss the CCF of *Acrochordus*, but the undivided fissura metotica of this taxon (there is no mention of the CCF at all in that paper). In fact, the basal placement of *Acrochordus* by McDowell (1979) and others was mostly based on its undivided fissura metotica (a well documented embryonic feature within squamata), and only partially on its lack of a CCF (which according to some is still present in this snake but in rudimentary form, see authors’ ref. 27).

We have modified this paragraph as follows: “The absence of the CCF in *Acrochordus* was therefore considered consistent with a basal placement of *Acrochordus* among snakes, similar to the first scenario proposed herein for *Thamnophis* [36]. However, this basal placement of *Acrochordus* was later rejected, due largely to the strong phylogenetic evidence placing *Acrochordus* at a far more deeply nested position as the sister to Colubroidea [37]; again, this rejection parallels our rejection of scenario one in *Thamnophis* (i.e., lack of a CCF does not inherently indicate basal phylogenetic status). This later analysis re-characterized the otico-occipital region of the skull of *Acrochordus* – focussing on the persistent lack of division of the metotic fissure – as paedomorphic, making *Acrochordus* unique among snakes in retaining this embryonic morphology – similar to our scenario two – though lacked ontogenetic data supporting this conclusion [18,37].”

These changes address the reviewer’s comments by: 1) removing our previous emphasis on the CCF as direct evidence for a basal placement of *Acrochordus*, as suggested by the reviewer; and 2) modifying our discussion of Rieppel (1980) to emphasize the metotic fissure rather than the CCF.

Note that, while Rieppel (1980) focussed on the undivided metotic fissure as paedomorphic, his later paper with Zaher [ref. 18 herein] included the CCF in this hypothesis of paedomorphosis:

“[*Acrochordus*] appears to retain a number of surprisingly plesiomorphic features in its cranial anatomy. [...] The most unusual features in the skull of *Acrochordus* are the apparent lack of the crista circumfenestralis and the undivided metotic fissure (McDowell, 1979), characteristics that have been explained as a probably consequence of paedomorphosis.” (Rieppel and Zaher 2001: pg 252)

As such, our discussion of Rieppel (1980) similarly mentions the overall otico-occipital region of *Acrochordus* as paedomorphic, in line with Rieppel and Zaher’s (2001) discussion of this hypothesis.

Finally, the reviewer mentions that, according to some authors, a rudimentary form of the CCF is still present in *Acrochordus*; however, a more recent large-scale sampling of the CCF among colubroids [ref. 20 herein] considered it to be absent in *Acrochordus*, which is the interpretation we have presented herein (see Fig. 7).

18. Rieppel O, Zaher H. 2001 The development of the skull in *Acrochordus granulatus* (Schneider) (Reptilia: Serpentes), with special consideration of the otico-occipital complex. *J. Morphol.* 249, 252–266.

20. Palci A, Caldwell MW. 2014 The Upper Cretaceous snake *Dinilysia patagonica* Smith-Woodward, 1901, and the crista circumfenestralis of snakes. *J. Morphol.* 275, 1187–1200. (doi: 10.1002/jmor.20297)

36. McDowell SB. 1979 A catalogue of the snakes of New Guinea and the Solomons, with special reference to those in the Bernice P. Bishop Museum. Part III. Boinae and Acrochordoidea (Reptilia, Serpentes). *J. Herpetol.* 13, 1-92.

37. Rieppel O. 1980 The perilymphatic system of the skull of *Typhlops* and *Acrochordus*, with comments on the origin of snakes. *J. Herpetol.* 14, 105-108.

Page 23, line 26: "From our data, the persistent absence of the CCF throughout all three ontogenetic stages unequivocally demonstrates this condition in *Thamnophis* to be a case of paedomorphosis." As I pointed out above (comment on page 21, line 10), this is an incorrect statement based on a misinterpretation of what paedomorphosis is and how it is detected. Absence of a feature in all developmental stages of the same given species does not provide evidence of paedomorphosis. The authors need to identify the plesiomorphic embryonic/juvenile snake condition via a comparison with other embryonic/juvenile snakes.

See our response to the comment on page 21, line 10. We have also modified this sentence to: "**From our data, the persistent absence of the CCF throughout all three ontogenetic stages (a condition shared with numerous non-ophidian lizards and with other more phylogenetically basal snakes such as *Acrochordus* [18]) demonstrates this condition in *Thamnophis* to be a case of paedomorphosis.**"

18. Rieppel O, Zaher H. 2001 The development of the skull in *Acrochordus granulatus* (Schneider) (Reptilia: Serpentes), with special consideration of the otico-occipital complex. *J. Morphol.* 249, 252–266.

Page 23, line 43: replace "the adult condition matches the embryonic condition" with "the adult condition matches the plesiomorphic snake embryonic condition".

Done.

Page 25, line 3: insert "main" after "two" (exceptions have already been noted in the literature, and have also been mentioned in this manuscript).

Done.

Page 25, line 8: insert a reference after *Homalopsis* (this taxon was not the object of this study).

Done.

Page 25, line 12: "Our observations of suspensorium development in *T. radix* also provide novel evidence for paedomorphosis in the evolution and ontogeny of the scolecophidian skull." The condition observed in *Thamnophis* does not provide any evidence for paedomorphosis in scolecophidians, unless it is shown to represent the general (plesiomorphic) embryonic snake condition. What it does is open the possibility to new interpretations on how the scolecophidian skull may have come to be if the contact between quadrate and prootic indeed turns out to be a plesiomorphic embryonic snake trait (i.e. pending further data).

We have changed this sentence to: “Our observations of suspensorium development in *T. radix* also enable novel interpretations of paedomorphosis in the evolution and ontogeny of the scolecophidian skull.”

(Our response to the comment on page 19, line 33, as well as our updated section 4.2, fully address the comments made here by the reviewer.)

Page 25, line 27 and following: “Furthermore, the absence of a crista circumfenestralis in *T. radix* [...] provides evidence for paedomorphosis in the evolution of this trait.”

As discussed above, the absence of a crista circumfenestralis in embryonic and adult *Thamnophis* does not demonstrate that paedomorphosis is the cause of this absence in the latter. *Thamnophis* alone does not tell us anything about which evolutionary developmental process is responsible for the absence of the CCF in adult snakes. The authors first need to verify that absence of a CCF is a plesiomorphic condition in embryonic or juvenile snakes, if that is the case, then they can conclude that its absence in adult *Thamnophis* can be interpreted as paedomorphic.

See our response and associated revisions for the comments regarding page 21, line 10.

Page 33, line 6: as discussed above, in booids the quadrate does not always remain upright (see comment on page page 17, line 47)

We have changed this sentence to: “(A) Booids (boas, pythons, and relatives) achieve macrostomy via elongation of the supratemporal and quadrate, with the quadrate becoming ventrolaterally deflected but remaining in the same transverse plane.”

Reference 25 is likely Kamal (1969) (and not 19696).

Done – corrected to Kamal (1966).

Table 1:

Frontal & Adult: Completely sutured with frontal (not “completely fused along frontal suture”).

Changed to: “Frontals completely sutured”

Parietal & Embryo: “Almost entirely unfused along sagittal suture.” There is no sagittal suture to speak of, only an unossified central region of the parietal.

Changed to: “Central parietal roof unossified”

Parietal & Juvenile: parietal roof ossified (not “sutured”).

Done.

Dentary & Embryo/Juvenile/Adult: To my knowledge, tooth counts on marginal dentigerous elements (dentary and maxilla) are very conservative in snakes within a given species and throughout ontogeny. Please recheck tooth positions, and make sure you don't simply count visible teeth, but tooth positions. Tooth counts from multiple specimens (and from both sides of the skull) would be welcome to make sure that there is no intraspecific variability added to what may vary ontogenetically.

See our response to the comment on page 9, line 22.

Figure 6: I wonder if showing the angle between quadrate and compound in a tilted perspective of the skull is appropriate (the angles appear smaller than they actually are). I think that showing these angles in lateral view would be much more effective.

We used a tilted perspective for this figure to provide a more direct view of the suspensorium articulation than is visible in lateral or dorsal view. To more accurately display the change in angle of the quadrate, we have moved the angles to Fig. 3, and have added the following sentence to the Fig. 3 caption: “Note the posterior rotation of the quadrate from the juvenile to adult stages.”

The figure caption contains a sentence that I find misleading: “this pathway of development is unique to *T. radix* and one other, unrelated genus within the Caenophidia, indicating independent evolution of this pattern”. The sentence assumes that only *Thamnophis* and another caenophidian have a type C macrostomy (see fig. 7), while in fact those are simply the only two snakes where presence of this type of jaw suspension has been noted so far in the literature. There has been no systematic review of jaw suspension in all snakes, therefore making such generalizations is unjustified. The authors should rephrase by writing that “this pathway of development has so far been observed only in ...” (do not write that it is unique to two species, we simply do not know that).

Done.

Figure 7: see comment above (Page 22, line 19).

See our response to the aforementioned comment.

Reviewer: 2

Comments to the Author(s)

This is a well-written paper and was a pleasure to read! I rarely review manuscripts that do not require many grammatical and stylistic changes, so, thank you.

Here are my comments:

Materials and Methods section: Briefly explain why you only scanned 8 specimens of this widely available species (cost, local availability?). Were the two juveniles from the same mother or were they unrelated? It is not fatal to your conclusions if the juveniles are related, but it would be good to mention.

We’ve expanded the first paragraph of the Materials and Methods section to address these points.

Regarding relatedness among sampled individuals, we added the following explanation: “Specific information regarding the parentage of the embryonic and juvenile snakes was unavailable. However, given that all of the embryos were accessioned under the same specimen number, it is likely that they are from the same mother. The same line of reasoning also applies to the two juvenile snakes.”

Regarding our sample size, we added the following explanation: “Our sample size was limited to eight individuals, as the only other embryonic or juvenile *Thamnophis* specimens in the UAMZ collections were from other species and therefore would have introduced interspecific variation as a confounding factor in this ontogenetic analysis. This sample size is consistent with that used in other micro-CT-related studies of snake morphology (e.g., see [6,15]).”

6. Palci A, Lee MSY, Hutchinson MN. 2016 Patterns of postnatal ontogeny of the skull and lower jaw of snakes as revealed by micro-CT scan data and three-dimensional geometric morphometrics. *J. Anat.* 229, 723-754. (doi: 10.1111/joa.12509)

15. Chretien J, Wang-Claypool CY, Glaw F, Scher MD. 2019 The bizarre skull of *Xenotyphlops* sheds light on synapomorphies of Typhlopoidea. *J. Anat.*, (doi: 10.1111/joa.12952)

You scanned the heads of alcohol preserved specimens, not skulls, so please explain briefly that you digitally removed soft tissues and rendered the skulls for analysis.

We’ve modified the end of the 2nd paragraph of the Materials and Methods section to explain this (“The resulting image files were visualized in Dragonfly 1.0 [...] with the Threshold tool being used to digitally remove the surrounding soft tissue from the skull”).

Results section 3.1 (page 6, line 15): I think that the premaxillary notch is seen better in the ventral view and you should refer to Fig. 4, not Fig. 3.

We’ve added references to Fig. 4 and our new Fig. 5 in this sentence, and in the other figure references throughout that paragraph.

Results section 3.5. (page 10, lines 29ff): It would be good to add an illustration of the crista circumfenestralis in a different colubrid to allow direct comparison with your specimens. If that's impossible, insert a citation where the reader could find such an illustration.

We’ve added a reference to Palci and Caldwell (2014), which provides detailed illustrations of the CCF in a range of snake taxa for comparison.

Palci A, Caldwell MW. 2014 The Upper Cretaceous snake *Dinilysia patagonica* Smith-Woodward, 1901, and the crista circumfenestralis of snakes. *J. Morphol.* 275, 1187–1200. (doi: 10.1002/jmor.20297)

Discussion section 4.1 (bottom of page 16): Briefly mention diets of juveniles vs adult *T. radix*. Since you are talking about ontogenetic shifts in diet and gape, prey type is important.

Done – we’ve added a sentence specifying this shift in diet: “Specifically, *T. radix* juveniles eat small prey items such as annelids and small anurans, whereas adults consume a range of prey, including larger organisms such as small mammals [23]”.

23. Tuttle KN, Gregory PT. 2009 Food habits of the Plains Garter Snake (*Thamnophis radix*) at the northern limit of its range. *J. Herpetol.* 43, 65-73. (doi: 10.1670/07-298R1.1)

Discussion section 4.3 (page 22): Explain why you didn't examine other *Thamnophis* to see presence or absence of CCF. This is one of the big points of your paper, you should tell the reader why you didn't check other species. *Thamnophis* are readily available in collections.

There are a few reasons for this: mainly, since we were focussing specifically on the ontogeny of *T. radix*, we focussed our sampling and CT-scanning efforts on individuals of that species. For logistical reasons, we weren't able to CT-scan alcohol-preserved specimens of other species. We did compare our results to a skeletonized specimen of *T. sirtalis parietalis* (which we've modified the manuscript to mention), but limited access to other skeletonized specimens also contributed to our limited comparisons. Of course, as you note, this observation is a key point in our paper, so we've also modified the manuscript to highlight denser sampling and comparisons as a key avenue of future research. See our addition to the end of section 4.3:

"Preliminary comparison to *T. sirtalis parietalis* reveals this taxon to also lack a CCF (CS, pers. obs.), suggesting that this absence may be widespread within the genus *Thamnophis* as a whole. Since our study focusses specifically on the ontogeny of *T. radix*, our sampling efforts were in turn focussed on this species. However, denser sampling of other colubroid species – including other species of *Thamnophis* – is a key avenue of future research to further investigate the possibilities we have raised herein."

Table: Define LARST in table heading or inside table. In the row about the frontal, column 3 (adult), it should read "Complete contact with parietal", not with frontal.

We've added a definition for LARST in the table caption, and have also corrected "frontal" to "parietal."

These are all the suggestions I have. I'm looking forward to seeing this in print.

Appendix B

Reviewer comments to Author:

Reviewer: 1

Comments to the Author(s)

I must confess that I am a bit disappointed by the tone of the authors' replies, and especially that after spending a considerable amount of time writing what I would consider a thorough and helpful review, my comments were considered "unrealistic, demanding, and often pedantic". I can guarantee the authors that it was not my intention "to create the impression that their manuscript was flawed empirically, conceptually, and grammatically." A review is meant to provide comments and suggestions on how to improve a manuscript, and my belief is that it needs to be rigorous when pointing out flaws, no matter how small.

In any case, the manuscript has now been significantly improved, and I have only a few additional comments.

As stated in our initial response to this review, we are grateful for the thoroughness of the provided comments and criticisms. As you state (and we agree), the manuscript was much improved in light of both reviewers' comments. The above quote was not in reference to the entire review from Reviewer 1, but only to those select aspects of this review which extensively characterized our paper as "highly misleading" and "deceptive" (e.g., the original Fig. 7) and which heavily disputed our understanding of certain methodologies and concepts (e.g., phylogenetic bracketing and paedomorphosis, respectively). We emphasize that we do respect the value of such a rigorous review and the detail that went into writing it, and are pleased to see that we have sufficiently addressed all of the points raised therein.

1) My original comment on Page 6, line 13 was: "posterior premaxillary notch which will eventually accommodate the septomaxilla and vomer". The notch does not accommodate any part of those bones.

To which the authors replied: *The Reviewer is in error – the premaxilla of all squamates, *Thamnophis* included, articulates with both the vomer and septomaxilla. The articulation is clearly articulated and the notch we are discussing is visible in our supplementary files (S4-S6), and in numerous illustrations provided in such works as Cundall and Irish (2008:pg 404, fig. 2.25). Perhaps we are referring to different notches, but as mentioned, we illustrate what we mean by this.*

I take it that the posterior notch in question is the notch between the two vomerine processes of the premaxilla, if so, it does not accommodate neither the vomer nor the septomaxilla. While the vomer and septomaxilla do articulate with the premaxilla in most squamates, these bones do not enter that notch (quite obvious in Fig. 4). If the authors are referring to a different notch then please elaborate and be more precise in your description.

We are in fact referring to the notch between the vomerine processes ventrally and the nasal process dorsally of the premaxilla. To clarify this, we have modified the description to read:

“As such, the space between the nasal process and vomerine processes of the premaxilla, which will eventually accommodate the septomaxilla and vomer, is shallow and articulates only with the septomaxilla (Fig. 3A, 4A, 5A). In the juvenile, the vomer’s premaxillary processes become elongated relative to the rest of the vomer, now extending to the vomerine processes of the premaxilla (Fig. 3B, 4B, 5B). From the juvenile to adult stages, the premaxilla’s nasal process expands both dorsoventrally and anteroposteriorly, thus making the space between the nasal and vomerine processes narrower dorsoventrally and deeper anteroposteriorly.”

2) My original comment on Page 12, line 12 was: What is the point of segmenting all the individual bones if then internal anatomical details like these are not properly figured and labeled? I would recommend adding some additional figures, if not to the main text at least as supplementary information.

To which the authors replied: *We have added a figure showing an internal view of the braincase, with pertinent features labelled (Fig. 6D-F). Had the Reviewer consulted the supplementary data he/she would have noted we supplied the requested internal anatomy as STL files.*

I must admit that I missed the supplementary data in my first round of review, but in my defense there was no mention of such data in the main text (the link to the dryad repository was only presented at the end of the manuscript and could be easily missed). In any case, anatomical features are not labeled in the STL files, therefore the new figures certainly help illustrate what the authors are writing about.

We are glad that our new figures increase the clarity of our descriptions. We hope that the added references to the supplementary files in the figure captions (added in the first round of reviews) will prevent the above issue for future readers. We also note that STL files are 3-D vector files which cannot carry individual text labels for anatomical components as regular in-text figures can. Yet, these files are aimed for the specialist reader interested in snake anatomy, and by themselves provide the most detailed visualization to this day of individual anatomical components of the skull. In combination with any anatomical map of the snake skull (e.g., Cundall and Irish 2008), any dedicated student of squamate anatomy will be able to use these files to understand individual components of the snake skull in great detail.

3) Concerning Fig. 7, my original comment was: In conclusion, unless the authors really possess data on the type of suspensorium for all snakes, I would recommend them to remove such level of detail from the figure, which is highly misleading about what we actually know about the distribution of types of macrostomy within specific snake lineages. The authors should simplify this figure and make it very clear that it only represents a summary of their personal hypothesis of the distribution of these traits within snakes. They should be more straightforward about what represents actual data and what is only assumed to happen in the various lineages.

To which the authors replied: *Strongly disagreed. The reviewer cannot be serious on this comment. Again, establishing the nonsense empirical constraint of having ALL data on ALL snakes does not make this requirement so, it only enforces the Reviewer’s misguided approach to this component of our*

manuscript. A case in point, no phylogenetic hypothesis ever published, using genes or morphology, has ever once sampled all taxa and all genes and all morphology – never once; according to the Reviewer’s expectations of our study, not a single phylogenetic or comparative study should ever have been published in the history of evolutionary biology. Yet these hypotheses of sister-group relationships and inferred phylogeny exist and are used, clearly by the Reviewer (if in fact the reviewer is not guilty of generating phylogenies from incomplete and imperfect data sampling) for second, third and fourth order hypothesis constructions. It is ridiculous to make such statements and expect accommodation of these extensive, misguided, and condescending criticisms.

Obviously I would not expect anyone to sample all species of snakes to draw some general conclusions as those presented in this manuscript, mine was a rhetorical point. The issue was that the authors’ Fig. 7 did not make it explicit which taxa had been actually sampled and for which ones anatomical conditions had been extrapolated. A naïve reader (or just someone unfamiliar with the snake literature) looking at their figure may have mistakenly concluded that all of those snake families and subfamilies had been sampled, at least partially if not thoroughly. I am pleased to see that the authors have modified their Fig. 7 so that readers can now see which families/subfamilies have actually been sampled, and also made it explicit that the figure just represents their personal hypothesis of the evolution of macrostomy.

Again, we’re glad to have sufficiently addressed and clarified the points raised regarding this figure.

4) My original comment on Page 22, line 33 was: To be honest, Rieppel 1980 (ref. 33 herein) did not discuss the CCF of Acrochordus, but the undivided fissura metotica of this taxon (there is no mention of the CCF at all in that paper).

To which the authors replied: Note that, while Rieppel (1980) focussed on the undivided metotic fissure as paedomorphic, his later paper with Zaher [ref. 18 herein] included the CCF in this hypothesis of paedomorphosis: “[Acrochordus] appears to retain a number of surprisingly plesiomorphic features in its cranial anatomy. [...] The most unusual features in the skull of Acrochordus are the apparent lack of the crista circumfenestralis and the undivided metotic fissure (McDowell, 1979), characteristics that have been explained as a probable consequence of paedomorphosis.” (Rieppel and Zaher 2001: pg 252). As such, our discussion of Rieppel (1980) similarly mentions the overall otico-occipital region of Acrochordus as paedomorphic, in line with Rieppel and Zaher’s (2001) discussion of this hypothesis.

Then the authors should have referenced Rieppel and Zaher (2001) and not Rieppel (1980). However, the authors should know that Rieppel and Zaher (2001) also misquoted Rieppel (1980). In fact, the complete quote from Rieppel and Zaher (2001:252) would be “In light of these phylogenetic relationships, the most unusual features in the skull of Acrochordus are the apparent lack of the crista circumfenestralis and the undivided metotic fissure (McDowell, 1979), characteristics that have been explained as a probable consequence of paedomorphosis (Rieppel, 1980).” The citation of Rieppel (1980) at the end of the quote was conveniently left out by the authors in their reply. I suspect that this is a case where a misquote is reiterated by someone (the authors) who simply trusted another paper (in this case Rieppel and Zaher, 2001) without actually verifying the accuracy of that quote by reading the original source (in this case Rieppel, 1980). Thus, a situation is created where two articles (Rieppel and Zaher, 2001 and this manuscript) make reference to the work of Rieppel (1980) as if it said something

about the CCF of *Acrochordus* as being a pedomorphic feature, while in truth that work didn't mention the CCF at all. I understand that it is common practice to trust what is written in the primary literature and take the references therein as accurate (especially when one of the authors is on both the referenced and the citing paper!). However, the authors should have checked the original paper by Rieppel to make sure that their statements are indeed accurate and avoid perpetuating a case of poor scholarship.

Despite all this, I am happy with the way the authors have modified the problematic paragraph so that a reference to Rieppel (1980) implying that he wrote about the CCF of *Acrochordus* is now avoided:

*We have modified this paragraph as follows: "The absence of the CCF in *Acrochordus* was therefore considered consistent with a basal placement of *Acrochordus* among snakes, similar to the first scenario proposed herein for *Thamnophis* [36]. However, this basal placement of *Acrochordus* was later rejected, due largely to the strong phylogenetic evidence placing *Acrochordus* at a far more deeply nested position as the sister to *Colubroidea* [37]; again, this rejection parallels our rejection of scenario one in *Thamnophis* (i.e., lack of a CCF does not inherently indicate basal phylogenetic status). This later analysis re-characterized the otico-occipital region of the skull of *Acrochordus* – focussing on the persistent lack of division of the metotic fissure – as pedomorphic, making *Acrochordus* unique among snakes in retaining this embryonic morphology – similar to our scenario two – though lacked ontogenetic data supporting this conclusion [18,37]."*

We're glad that our revisions have addressed your concerns regarding this paragraph. To clarify what we'd originally written, as you've discussed above: While Rieppel and Zaher (2001) technically misquoted Rieppel (1980), we interpreted these papers in light of the fact that the metotic fissure becomes subdivided into the jugular foramen and the lateral aperture of the recessus scalae tympani, separated by the crista tuberalis (one of the components of the CCF). If the metotic fissure is undivided as in *Acrochordus*, this necessarily means that the crista tuberalis is absent and therefore that the CCF must also be absent. This is a logical conclusion based on the anatomical definition of the CCF. Therefore, while Rieppel (1980) did not explicitly refer to the CCF, the absence of this structure is a necessary implication of the undivided metotic fissure present in *Acrochordus*. Furthermore, if the metotic fissure in *Acrochordus* is undivided due to pedomorphosis, then by extension the crista tuberalis – and thus the CCF – are also absent due to pedomorphosis. We therefore view Rieppel and Zaher's (2001) discussion of the previous paper and their added mention of the CCF as simply an explicit reference to an implicit point raised by Rieppel (1980). In this manner, Rieppel and Zaher correctly cited Rieppel (1980), and so did we. We did in fact read both of these papers (rather than blindly trusting what Rieppel and Zaher had written, as per the erroneous claims offered here by Reviewer 1), but we interpreted Rieppel and Zaher (2001) as having followed the same line of logic presented above and therefore did not take issue with discussing Rieppel (1980) in relation to both the CCF and metotic fissure. We agree with the reviewer regarding the importance of verifying the primary literature and hope that this elaboration has clarified our thinking in writing our original paragraph.

In summary, I would recommend the authors to be more explicit about which "posterior premaxillary notch" (point 1 above) they are referring to, because at the moment the only posterior premaxillary

notch I can think of is that between the vomerine processes of the premaxilla. This notch does not accommodate the septomaxilla or the vomer (quite evident in their Fig 4). If the notch they are describing is another notch, then they need to be clearer in their description. Apart from this minor issue I am happy with the manuscript as it is.

As discussed above, we've clarified our description. Again, we thank the reviewers for their comments throughout the review of this manuscript.